# Anthropogenic climate and land-use change drive short- and long-term biodiversity shifts across taxa

Teresa Montràs-Janer [1], Andrew J. Suggitt [2], Richard Fox [3], Mari Jönsson [4], Blaise Martay[5], David B. Roy [6], Kevin J. Walker [7] & Alistair G. Auffret [1] ✉

Climate change and habitat loss present serious threats to nature. Yet, due to a lack of historical land-use data, the potential for land-use change and baseline land-use conditions to interact with a changing climate to affect biodiversity remains largely unknown. Here, we use historical land use, climate data and species observation data to investigate the patterns and causes of biodiversity change in Great Britain. We show that anthropogenic climate change and land conversion have broadly led to increased richness, biotic homogenization and warmer-adapted communities of British birds, butterflies and plants over the long term (50+ years) and short term (20 years). Biodiversity change was found to be largely determined by baseline environmental conditions of land use and climate, especially over shorter timescales, suggesting that biodiversity change in recent periods could reflect an inertia derived from past environmental changes. Climate–land-use interactions were mostly related to long-term change in species richness and beta diversity across taxa. Semi-natural grasslands (in a broad sense, including meadows, pastures, lowland and upland heathlands and open wetlands) were associated with lower rates of biodiversity change, while their contribution to national-level biodiversity doubled over the long term. Our findings highlight the need to protect and restore natural and semi-natural habitats, alongside a fuller consideration of individual species' requirements beyond simple measures of species richness in biodiversity management and policy.

One of the main concerns of the ongoing biodiversity crisis is that future losses are predicted to reduce the resilience of ecosystems to further change[1,2]. Already, shifts in land use and a changing climate are considered to be the most important drivers of global biodiversity loss and the reorganization of ecological communities over time[3–5], both across taxa[1,6] and spatiotemporal scales[7–9]. Since the early twentieth century, agriculture and forestry have intensified across Europe[10,11] and these changes have been broadly associated with declining biodiversity and the homogenization of species assemblages across taxonomic groups[12–15]. Over the same period, the global climate has warmed by

[1]Department of Ecology, Swedish University of Agricultural Sciences, Uppsala, Sweden. [2]Department of Geography and Environmental Sciences, Northumbria University, Newcastle, UK. [3]Butterfly Conservation, East Lulworth, UK. [4]Swedish Species Information Centre, Swedish University of Agricultural Sciences, Uppsala, Sweden. [5]British Trust for Ornithology, Thetford, UK. [6]UK Centre for Ecology & Hydrology, Wallingford, UK. [7]Botanical Society of Britain and Ireland, Harrogate, UK. ✉e-mail: alistair.auffret@slu.se

~1 °C (ref. [16]), which has been associated with shifts in species' ranges to higher latitudes[6,17,18]. However, climate and land-use change do not occur in isolation. Habitat availability is a prerequisite for successful climate-driven range expansions[19,20], while landscapes that have already been subjected to habitat destruction can both inhibit colonization by warm-adapted species and reduce persistence in cold-adapted species[21,22]. Also, the type of change in land use that occurs in each location affects the level of climate warming that organisms experience (land use affecting local, microclimatic conditions) and thus may influence the trajectories of biological change[23–26]. Furthermore, studies of climate change effects have shown that baseline environmental conditions (conditions at the beginning of the investigated time period of change) can determine both community change and modern patterns of biodiversity[27,28]. Yet, due to a lack of historical land-use data covering the timescales over which anthropogenic climate change and its effects have manifested, investigations of climate–land-use interactions are often limited to space-for-time substitution (comparison of sites across a spatial gradient[29]) or the use of modern anthropogenic land cover as a measure of landscape change[21,22]. Thus, the potential for land-use change to interact with a changing climate to affect biodiversity, as well as any effect of baseline environmental conditions remains largely unquantified, particularly over longer timescales (decades or more).

Here, we compile a comprehensive national-scale dataset of land use, climate (average mean annual temperature and average total annual precipitation) and species observations at the 10 km grid cell level, to investigate baseline and interacting effects of land-use and climate change on biodiversity changes in birds, butterflies and plants over two different temporal scales (Fig. 1): 50+ years (long term, 1960s to 2010s) and 20 years (short term, 1990s to 2010s) in Great Britain. First, we identify whether there has been an increase or decrease in taxonomic richness and biotic homogenization and community adaptation to warmer climates (via the community temperature index (CTI), an indicator of the relative occupancy of warm- and cold-adapted species within a community[6]). Second, we investigate how changes in these three metrics relate to concurrent changes in climate and land use, in terms of their individual effects, their interactions and the effects of baseline conditions. We also consider the roles of baseline biodiversity conditions and microclimatic heterogeneity (variation in microclimate temperature), which can influence biodiversity responses to larger-scale environmental changes[1,2,24]. Third, we uncover how climate and land use are associated with local (grid cell) contribution to national-level beta diversity over time, which can aid conservation prioritization and planning both by identifying particular locations that contribute to large-scale biodiversity, as well as finding environmental attributes shared by the most valuable sites[30].

We focused on local-assemblage responses at the 10 km square grid cell resolution, rather than species-specific responses. For each taxon and temporal scale, we only included in the analysis the grid cells that were recorded in both time periods (1960s and 2010s for the long term; 1990s and 2010s for the short term). The species data refer to atlas and monitoring programmes with uneven and unknown recorder effort across time and space. To deal with spatiotemporal variation in recorder effort and to prevent our models from producing biased results, we first estimated for each taxon and time period (1960s, 1990s and 2010s), the recorder effort in each focal grid cell, using the Frescalo approach[31]. Then, we proceeded as follows. (1) To reveal differences in biodiversity between time periods (1960s and 2010s for the long term; 1990s and 2010s for the short term), we fitted for each taxon, biodiversity metric and temporal scale, one generalized linear mixed-effects model (GLMM), with a natural logarithm link function. Species richness, beta diversity (measured as the mean community distance of each focal grid cell in relation to the surrounding eight grid cells) and CTI were modelled as a function of estimated recorder effort, together with a categorical variable with two categories representing time period, that is, 1960s and 2010s for the long term and 1990s and

2010s for the short term. The parameter estimate of this categorical variable therefore describes any increase or decrease in biodiversity and any shift in CTI between time periods. Estimated recorder effort was included in the model as a variable on the logarithmic scale for the richness models and on the natural scale for the beta diversity model. Because CTI strongly depends on the identity of the species within the community, which cannot be captured by the recorder effort per se, we dealt with uneven recorder effort by applying a species-richness grid cell threshold cutoff (Methods). (2) To investigate baseline and interacting effects of land-use and climate change on biodiversity change, we fitted for each taxon, biodiversity metric and temporal scale, one linear mixed-effects model. The observed change in each biodiversity metric was modelled as a function of climate and land-use change, climate–land-use change interactions, baseline conditions of climate, land use, biodiversity and microclimatic heterogeneity. Estimated recorder effort at the initial time period (1960s for the long term; 1990s for the short term) and change of estimated recorder effort were both included as variables in the models, in their natural scale. For the CTI models, we again applied the same species richness grid cell threshold cutoff and did not include recorder effort at the initial time period, only the change in recorder effort. (3) To identify environmental characteristics associated with grid cell contribution to national biodiversity, we first calculated for each taxon and time period, the relative contribution of each grid cell to national-level beta diversity (local contribution to beta diversity, LCBD)[32]. LCDB is a comparative indicator of the ecological uniqueness of a site in terms of its contribution to beta diversity across all sites. Then, for each taxon and time period, we fitted one GLMM with a natural logarithm link function; LCBD was modelled as function of land use, climate and microclimatic heterogeneity, with estimated recorder effort included as a variable on the natural scale. All models in the study were run using the integrated nested Laplace approximation for Bayesian inference[33], including controls for spatial autocorrelation. To deal with confounding collinearity effects, we made use of sequential regression analysis[34–36].

## Results

### Trends of community change

Despite considerable variation between grid cells, our models (equations (1) to (3); Methods) show that communities of birds, butterflies and plants experienced an overall increase in species richness over both the long (50+ years) and short (20 years) temporal scales (Fig. 2). This was coupled with a trend of biotic homogenization, as measured by a decrease in beta diversity over time, with the exception of birds, for which beta diversity increased over the long term (1960s to 2010; Fig. 2). All three taxa exhibited an increase in CTI over the long term, indicating an increased representation of warm-associated species. However, butterfly communities showed a surprising decrease in CTI over the short term (1990s to 2010; Fig. 2), despite an overall warming climate. This result was counterintuitive, yet resistant to further examination of the data (Methods) and such a pattern has also been observed in other systems[37].

### Individual and interacting anthropogenic drivers

Despite variation across taxa, biodiversity metrics and temporal scales, some broad patterns emerged regarding the effects of anthropogenic climate and land-use changes on biodiversity and community change in birds, butterflies and plants during the last 60 years. Our models (equation (4); Methods) show that increases in species richness and biotic homogenization were generally associated with ongoing trends of increased temperature and precipitation and increased cover of anthropogenic land uses, especially in the long term (Fig. 3). The cover of agriculturally improved grasslands, forest, urban and arable land mostly increased at the expense of semi-natural grasslands (a broad category including the land-cover classes defined as meadows, pastures, lowland and upland heathlands and open wetland habitats)[38] (Methods,

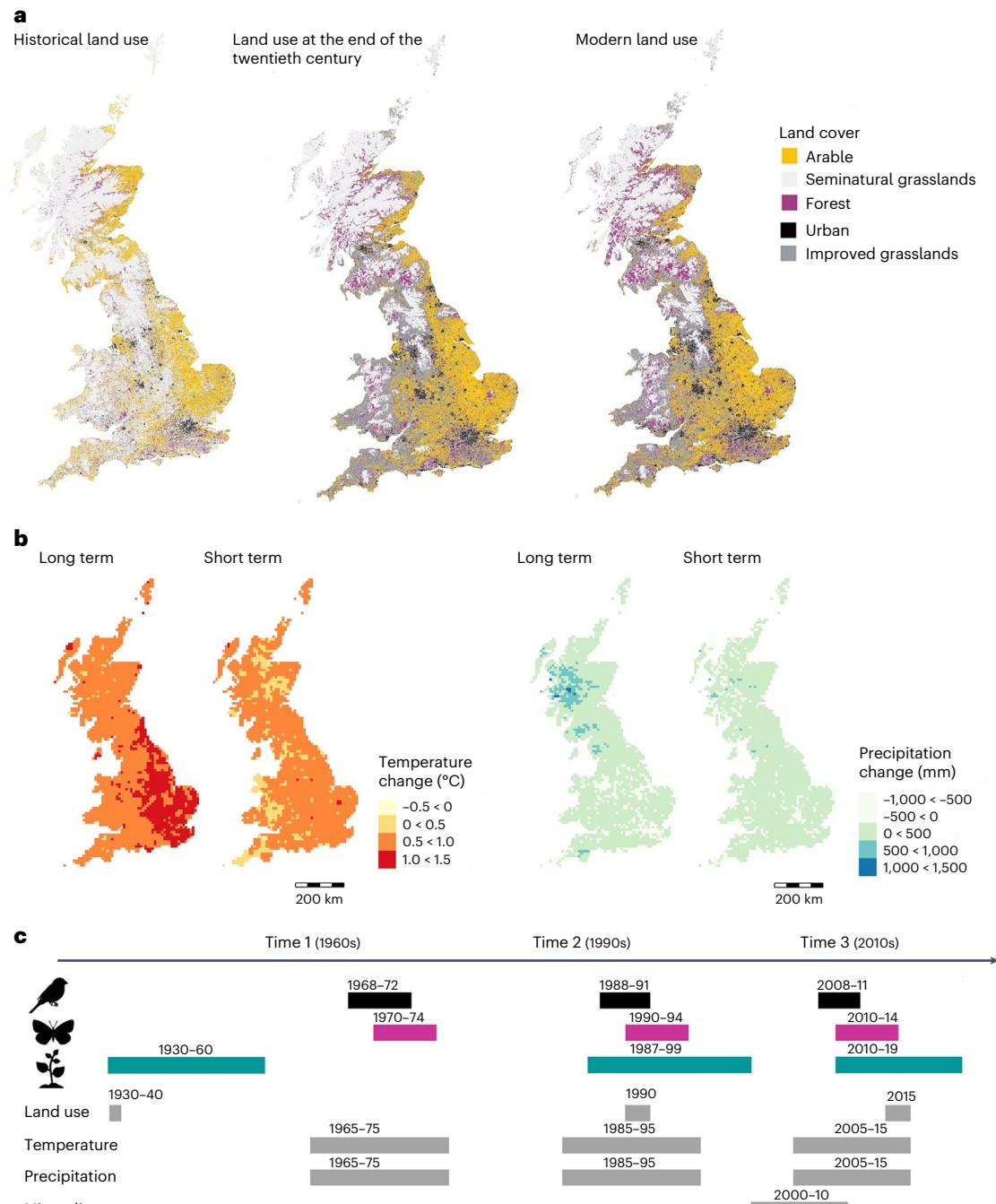

**Fig. 1 | Overview of the change of land use and climate in Great Britain across three time periods and the timeline of the environmental and species datasets used in this study. a**, Land use in Great Britain in the 1930s to 1940s, at the end of the twentieth century (1990) and in the modern period (2015), showing an overall increase of anthropogenic land cover (urban, arable and improved grasslands) mostly at the expense of decreasing coverage of semi-natural grasslands (Supplementary Fig. 1). **b**, Climate change in Great Britain showing changes of temperature (average annual mean temperature) (left) and changes of precipitation (average annual sum of daily precipitation) (right) over two different time periods: long and short terms (from 1960s to 2010s and 1990s to 2010s, respectively). **c**, Timeline of the collected datasets for birds, butterflies and vascular plants, land use and climate data in Great Britain showing the temporal coverage of each one of the datasets and their correspondence with the three time periods of study (time 1, 1960s; time 2, 1990s; time 3, 2010s). Panel **a** adapted from ref. 38, Springer Nature Limited. Data credits for panel **a**: left, Land Use Survey of Great Britain, copyright Giles N. Clark; centre, ref. 78; right, ref. 79 (data for maps at centre and right owned by the UK Centre for Ecology & Hydrology, database right/copyright UKCEH).

Fig. 1 and Supplementary Fig. 1). Results were robust to cross-validation (Methods and Supplementary Fig. 2) but there was a higher uncertainty in results for species richness, especially for plants (Fig. 3b).

Our findings also provide support for the effects of climate–land-use interactions on biodiversity change. For species richness and biotic homogenization, we found that interactions were mostly associated with community change over the long term but there were no clear patterns between or within taxa in terms of the direction of interactions (synergistic or antagonistic) (Fig. 3a). For CTI, interactions were found at both the long (50+ years) and short term (20 years). In the long term, increased cover of arable land and improved grasslands lessened the effect of climate warming on increasing butterfly CTI

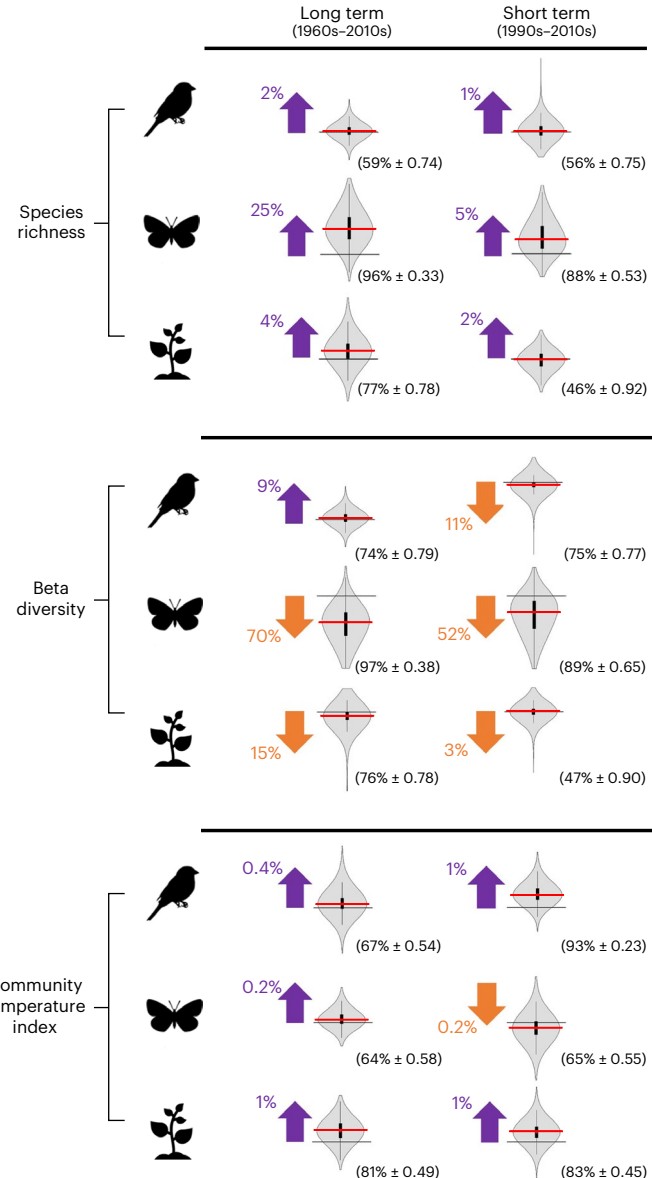

**Fig. 2 | Biodiversity change over time in Great Britain.** Estimated changes in species richness, beta diversity (as a measure of biotic homogenization) and CTI for three taxa in Great Britain at the long (1960s to 2010s) and short (1990s to 2010s) temporal scales. Arrows indicate increase or decrease in biodiversity between time periods, based on equations (1), (2) and (3) (Methods). Coloured numbers provide the estimated proportion of relative biodiversity increase or decrease (significant in all cases and s.d. <0.035; Supplementary Table 4). The violin plots (density curves with boxplots) capture the density distribution of the estimated biodiversity change across grid cells, based on equation (4) (Methods). Solid horizontal lines crossing the violin plots indicate the point where biodiversity change is zero. Red horizontal lines in box plot show the median values of the estimated change. Black bars display the interquartile range (IQR) (first and third quartile). Lower and upper black lines stretching from the black bars identify the first quartile −1.5× IQR and the third quartile +1.5× IQR. Black values in parenthesis give the estimated mean probability (in percentage) of increase or decrease (matching the arrow direction) ± s.e., across grid cells. For example, models estimate a significant average increase of 2% in bird species richness across Great Britain between the 1960s and the 2010s, with an estimated probability of 58–60% for a grid cell to have increased in richness during this time. Number of grid cells analysed as follows: birds 2,670 across all analyses; butterflies species richness and beta diversity 2,013 long term and 2,022 short term, CTI 996 long term and 1,222 short term; plants species richness and beta diversity 2,666 for both long and short terms, CTI 2,351 long term and 2,406 short term.

(antagonistic interaction). In the short term, however, anthropogenic land-use change amplified the effect of climate in driving increases in butterfly and bird CTI (synergistic interaction) (Fig. 3a) and Supplementary Fig. 3).

## The effect of baseline conditions

Baseline conditions of land use and climate were associated with biodiversity changes, especially on the short temporal scale (Fig. 3a). In the long term, community change was associated with both the baseline and the changes in environmental conditions that occurred, exceptions being CTI and biotic homogenization in birds. In the short term, biodiversity change was less related to changes in land use and climate and more related to baseline environmental conditions (that is, the group of explanatory variables for environmental baseline conditions had at least two more associations at 95% credible interval with biodiversity change, than the group of variables for environmental change), except for plant richness and biotic homogenization. Baseline cover of semi-natural grasslands appeared particularly important, with grid cells that originally contained more semi-natural grassland cover exhibiting lower increases in species richness and lower levels of biotic homogenization (lower decreases in beta diversity) (Fig. 3a). However, despite this apparent stability, these grid cells did experience high levels of turnover in terms of the percentage of species both gained and lost over time (Supplementary Table 1). Wetter and colder baseline conditions also promoted stability in some taxa. In that respect, grid cells with higher baseline annual precipitation were associated with lower rates of increased richness and biotic homogenization in bird communities. Also, historically cooler grid cells were associated with lower rates of increased richness and biotic homogenization in butterflies and experienced less climate-associated community change at the short term (lower decreases in butterfly CTI and lower increases in plant CTI) (Fig. 3a). With the exception of plants, long-term changes in CTI were not related to baseline climatic or land-use conditions. However, in the short term, a higher proportion of improved grasslands and semi-natural grasslands in the initial time period was associated with increases of bird CTI; while higher proportions of arable cover were linked to higher decreases and increases of butterfly and plant CTI, respectively (Fig. 3a).

In addition to baseline environmental conditions, baseline levels of species richness and beta diversity were among the variables with the greatest effects on biodiversity change over time. These variables contributed more than any others to improved model fit, leading to an increase in the marginal $R^2$ value (which considers only the variance captured by the fixed effects) of roughly 0.4 (Supplementary Fig. 4). All associations between biodiversity metrics and their baseline values were negative, meaning that higher baseline levels of species richness and beta diversity were associated with lower increases of species richness and greater decreases of beta diversity (higher biotic homogenization), respectively (Fig. 3a). Additionally, higher baseline levels of species richness and lower baseline beta diversity values were associated with more stable communities over time; that is, they were associated with lower percentages of both gained and extirpated species (Supplementary Table 1). Microclimate also influenced community reorganization. Our results show that a higher microclimatic heterogeneity was associated with larger increases of plant and butterfly species richness at both long and short temporal scales, while for birds it was associated with lower increases of CTI (Fig. 3a). Overall, the conditional $R^2$ values of our change models (equation (4); Methods) were considerably higher than the marginal $R^2$ (especially for CTI models) (Supplementary Table 2 and Supplementary Fig. 4), indicating that more variation in biodiversity change was captured by the spatial effects.

## Local contribution to beta diversity

Across all three taxa and time periods (1960s, 1990s and 2010s), our models (equation (5); Methods) indicated that higher values of relative

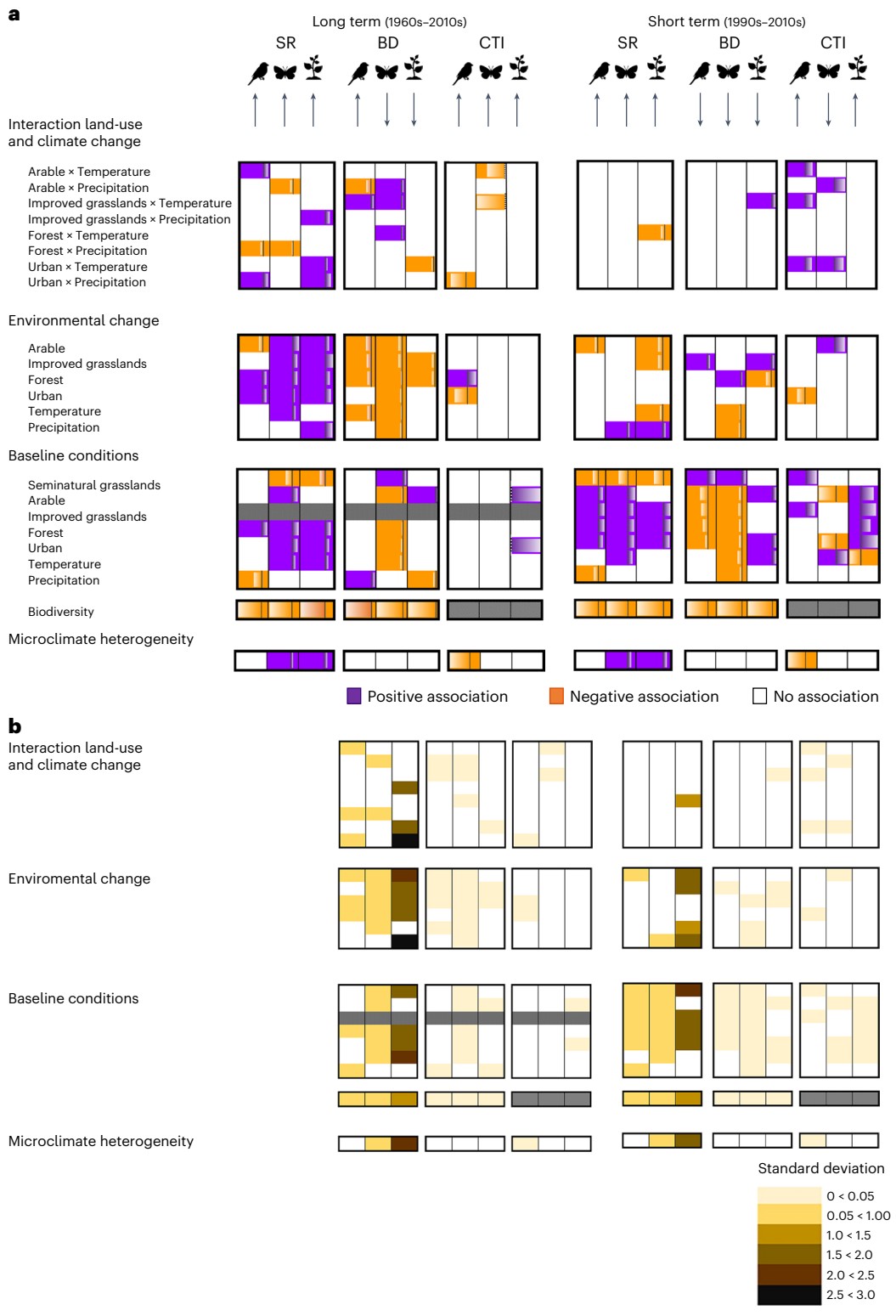

**Fig. 3 | Effect of land use and climate change on biodiversity at two temporal scales. a,b,** Environmental parameter estimates (mean values) (**a**) and associated uncertainty (standard deviation) (**b**) for the association analysis between changes in species richness (SR), beta diversity (BD) and CTI of three different taxon communities in Great Britain (response variable) and the changes of land use and climate, the interaction of land-use change and climate change, baseline conditions of land use and climate, baseline biodiversity (for SR and BD) and microclimatic heterogeneity (explanatory variables) over the long (1960s to 2010s) and short (1990s to 2010s) temporal scales. Coloured blocks in **a** represent the direction of the parameter estimate of each explanatory variable over the response (purple is positive, orange is negative), with gradient bars emanating from dashed zero lines within each box indicating the size of the fixed effect (Supplementary Table 2). Estimates are standardized within each taxon, biodiversity metric and temporal scale—based on equation (4) (Methods). Grey blocks indicate that (1) there was no improved grasslands category in the 1960s land-cover dataset and (2) we did not apply baseline-CTI in the models corresponding to this biodiversity metric. White blocks mean that no associations were found (that is, the 95% posterior distribution of the estimated mean of the coefficient included zero). For ease of interpretation, we have included the direction of the overall estimated biodiversity change at the top of the figure. Note that the compacted uncertainty table in **b** mirrors the table for the parameter estimates in **a**. Microclimate represents only the modern period.

grid cell contribution to national beta diversity (LCBD) were associated with a higher proportion of semi-natural grassland. That is, the communities of birds, butterflies and plants in grid squares that contained more semi-natural grassland cover were sufficiently different to other grid squares to have more positive effects on overall beta diversity in Britain. Semi-natural grasslands had the largest positive effect size of all predictor variables within each taxon and time period, while increased cover of forest and improved grasslands was associated with lower LCBD (Fig. 4 and Supplementary Table 3). Moreover, semi-natural grassland cover was the only variable to exhibit an increasing positive effect size over time, suggesting that the presence and maintenance of this habitat has become increasingly important for the contribution of each particular grid cell biodiversity in Great Britain (Fig. 4). For birds and plants, warmer areas were also associated with higher LCBD across all three time periods, whereas for all three taxa, microclimatic heterogeneity within a grid cell was associated with lower LCBD in the 1990s and the 2010s. Cross-validation analysis showed our models to be robust at estimating fixed effects (Methods and Supplementary Fig. 5).

## Discussion

Our study confirms that changes in climate and land use are associated with community reorganization and biodiversity change across several taxa for both long (50+ years) and short (20 years) temporal scales. Our analyses of British birds, butterflies and plants showed how anthropogenic land conversion, warming temperatures and wetter conditions led to increased richness within 10 km grid cell resolution and to decreases in beta diversity, resulting in an overall biotic homogenization and warmer-adapted communities at the national level.

Given the role of both climate and land-use change in determining biodiversity change over time[20,22,29], we reveal that climate–land-use interactions affect the magnitude of change in richness and biotic homogenization, especially in the long term. Together with the relatively large importance of baseline environmental conditions for determining community change over the short term compared to the long term, our results support the idea that short-term biodiversity change could reflect an inertia (a continuation of ongoing change) derived from past environmental changes. This means that ecological communities continue to reorganize at pace, despite the relatively low levels of climate and land-use change that occur over shorter periods of time. Increased land-use intensification, especially in terms of increased arable and improved grassland cover[39,40], is likely to have contributed to these short-term community shifts to some extent[13,29,41]. However, it is also possible that the continuing increase in species richness and biotic homogenization were triggered by environmental changes that had already happened, as time-lagged responses to environmental change have been demonstrated previously in all three of our study taxa[42–44]. In any case, we found that trends of increased richness and biotic homogenization were weaker in regions that contained higher levels of semi-natural grasslands (including pastures, meadows, lowland and upland heathlands and open wetlands) to start with, both in the long and short terms, signalling the importance of such habitats for biodiversity. In addition, our results indicated an increasing contribution of this habitat to nationwide beta diversity over time. This supports existing knowledge that the unique communities supported by semi-natural grassland habitats are of high biodiversity value[14,45] and shows that the losses in grassland cover that have occurred during our study period have only increased the importance to conservation of the grasslands that remain.

The broad biodiversity trends reported here align with previous findings of local and landscape-level change[8,28,46–48]. Nonetheless, to attribute increased species richness to anthropogenic land-use changes can be surprising and controversial at a time of global biodiversity loss[4]. In that respect, it is important to remember that large changes in the landscape had already occurred before the creation of the historical maps in the 1930s and 1940s and comparisons with pre-agricultural conditions would probably reveal that biodiversity had already been severely degraded. Thus, many species occurring in Great Britain at the start of our study period may have already been those associated with anthropogenic land uses and therefore benefitted from the continued expansion of agricultural and urban habitats[29,45]. Widespread species, including non-natives, have been observed to expand and fill their distributions in recent decades[49,50]. It is not unexpected that climate change can drive increased species richness in some cases; indeed, there is ample evidence of many taxa expanding their ranges over time in Great Britain[51]. However, time-lagged extirpations both in relation to land use or climatic change[52] could also mean that declines in species richness in response to environmental changes are yet to manifest.

The three taxa investigated here showed broadly similar trends over time and responses to environmental change, although the magnitude of such changes varied somewhat. Most strikingly, increases in butterfly richness and decreases in butterfly beta diversity were stronger than for birds and plants (Fig. 2). This is likely to be at least partly due to the lower national-level richness of butterflies meaning that changes in the same number of species as the other taxa will exhibit a larger percentage change, although this mathematical effect does not reduce the potential seriousness of these large changes that could result in corresponding impacts on any ecosystem functions carried out by butterflies. Additionally, fast generational turnover in butterflies means that they often respond relatively quickly to ongoing environmental change, especially habitat destruction[45]. This potentially strengthens the idea that we are observing lagged biodiversity responses to environmental changes taking place in Great Britain, at least in plants and birds. There were also some important exceptions to the direction of community change across taxa and their drivers, detailed below.

Consistent with other studies, we find that reorganization of ecological communities of birds, butterflies and plants reflects a community-level adaptation to warmer climates[6,17,18,25]. Moreover, the interactive effect of warming climate and increasing anthropogenic land use resulted in higher increase of warmer-adapted communities of birds and butterflies in the short term. This is to be expected because temperature plays a fundamental role in biological processes[53]. However, we found that, although butterfly CTI has increased between the 1960s and the 2010s, butterfly communities in the modern period are associated with cooler conditions than they were in the 1990s, despite continued climate warming. That butterfly CTI appears to have increased over the long term but not the short term (with a lower decrease of CTI in those areas with higher baseline temperature) could be related to the difficulty of butterflies in tracking warming climate (climatic debt)[6]. CTI decrease in the short term might also be driven by losses in species that are associated with arable and semi-natural grasslands that have experienced intensification and degradation, respectively[45]. These habitat changes have been particularly concentrated in the warmer south and east of Britain (Fig. 1), where warmer microclimates in these habitat types[54] could also favour warm-adapted species (compare Supplementary Fig. 6).

Birds, which have the highest dispersal capacity of our three taxa, exhibited a long-term increase in beta diversity, sticking out among the otherwise consistent trend of biotic homogenization. This higher mobility means that birds are both: (1) less constrained by local climates and (2) better able to track shifting isotherms albeit moderated by broad changes in land use[55], resulting in increased richness and beta diversity in the long term but increased biotic homogenization in the short term. Our results are also consistent with previous findings where increasing forest cover facilitated bird responses to warming climates[55] and where human-associated species have driven CTI increases in plants[56], providing evidence that changes in land use together with baseline environmental conditions can mediate community responses to climate change. On the other hand, increased precipitation was also shown to be related to reorganization of ecological communities,

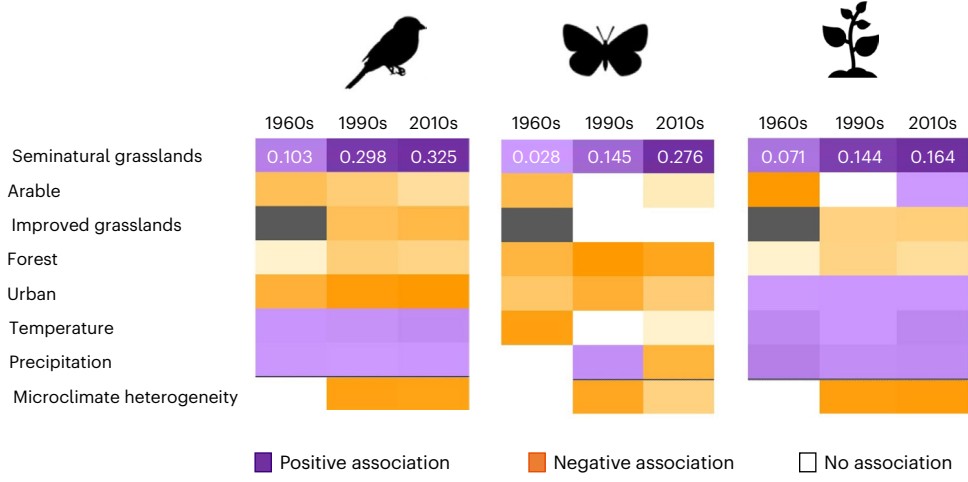

**Fig. 4 | Effect of land use and climate on LCBD over time.** Parameter estimates (mean values) for the association analysis between LCBD and land use and climate conditions in Great Britain, for three different taxon communities and three different time periods (1960s, 1990s and 2010s) based on equation (5) (Methods and Supplementary Table 3). Coloured blocks represent the values of the parameter estimates on a colour gradient, with orange indicating negative values and purple indicating positive values and on a scale from low positive or negative (light shades) to highly positive or negative (dark shades) within each taxon. Grey blocks indicate that there was no improved grasslands category in the 1960s land-cover dataset. White blocks mean that no associations were found. Parameter estimates corresponding to semi-natural grassland demonstrate their increased contribution to national-level biodiversity across time for all three taxa. All parameter estimates have an associated s.d. <1. Microclimate represents only the modern period.

with increasing richness and decreasing diversity of butterflies and plants. Increased annual precipitation has not always been found to benefit species that are associated with moist conditions[56]. Instead, higher variability in the timing of precipitation has resulted in more intense periods of drought, despite increased average rainfall[57]. This has been suggested to drive shifts towards communities tolerating drier climates[26]. We also found that microclimatic heterogeneity affected community reorganization on both long and short temporal scales. However, our results only provided evidence of microclimatic refugia for birds (that is, higher heterogeneity reducing the increase of CTI over time), while microclimatic heterogeneity was associated with increased richness for butterflies and plants and lower grid cell contribution to nationwide diversity across taxa. This was slightly surprising, given that the strong focus of microclimate in ecology on the reduction of extinction risk in communities experiencing environmental change[22–24,26,27]. Instead, our results indicate that microclimatic heterogeneity can also moderate community reorganization in the other direction, not by toning down the extirpation of declining species, but by potentially increasing niche opportunities for colonizing species[24].

We use a nationwide dataset of historical land use, climate and biodiversity information to quantify broad changes in biodiversity and community composition, attributing these changes to anthropogenic drivers of global change. Exercises such as these give important insights into human impacts on biodiversity at relatively large spatiotemporal resolutions but at the same time are inevitably quite simplistic generalizations of subtle changes that often occur at smaller scales. One of our main findings was that climate and land-use change interact to affect richness and biotic homogenization, mostly in the long term. However, whether the combined effect of a changing climate interacting with increased anthropogenic land use resulted in antagonistic or synergistic interactions varied across taxa. This suggests that community responses to interacting climate and land-use change probably mask a large variety of species-specific ecological requirements and shows that it is also important to consider species-specific responses to environmental changes, especially in a biodiversity conservation context. Species are also likely to respond to land-use change at a finer resolution than our 10 km grid cells. However, historical species observations are too sparse to be collated at a smaller scale. In a related point, despite accounting for uneven observer effort as much as possible, it is still not known exactly how representative historical observation data are and how any inconsistencies might affect our results. Nonetheless, although sometimes controversial, our findings (particularly regarding increased species richness) are broadly ecologically rational and follow previous findings based on more structured time-series data.

## Conclusions

High rates of climate change and land conversion are expected to continue through the twenty-first century. Despite being among the most degraded habitats in Great Britain and elsewhere in Europe[10,38], semi-natural grasslands were shown here to be associated with lower rates of biodiversity change over time, which together with these large-scale habitat losses resulted in doubling their contribution to national-level beta biodiversity since the mid-twentieth century. Therefore, we stress the need for protection, conservation and restoration of natural and semi-natural habitats to avoid more biodiversity loss in the future. We also showed that despite the significant effects of climate change and land conversion, these variables only explained a relatively low proportion of biodiversity and CTI change in our models. As such, other factors not addressed here, such as land-use intensification[29,41], topography[55], species' functional traits[58,59] or epigenetic mechanisms[60], are also likely to drive community change to some extent. In addition to meeting national and international goals in relation to preventing catastrophic habitat loss and climate change, halting biodiversity loss will also require local approaches to conservation that consider individual species and community structure, rather than just broad measures of species richness or abundance.

## Methods

### Bird, butterfly and plant data

We retrieved a total of 3,715,724 species occurrence records describing British communities of birds, butterflies and plants in the $n = 2,670$, 10 km square grid cells of the British National Grid. For birds, these data correspond to the atlases of breeding birds for 1968–1972[61]; 1988–1991[62] and 2008–2011[63]. For plants, to the atlases of 1930–1960[64]; 1987–1999[65] and 2000–2019[66] (for which we retrieved the data corresponding to the period 2010–2019). For butterflies, we used the periods 1970–1974; 1990–1994 and 2010–2014 from the national recording scheme Butterflies for the New Millennium, to match the other two taxa. For simplicity,

each one of these time periods is referred hereafter as 1960s, 1990s and 2010s.

For each taxon, we included species (including some taxonomic aggregations and subspecies for plants) that represented stable taxonomic concepts across time from a biological recording perspective, that is, recorded consistently throughout the study period (Supplementary Note 'Taxonomic filtering of observation data'). In total, 250 species of bird, 55 species of butterfly and 1,587 species of plant were retained for analysis.

#### Recorder effort

All datasets refer to presence-only data with uneven and unknown recorder effort, across time and space. Uneven recorder effort among grid cells may potentially lead to biased results[67,68]. To tackle this issue, we estimated for each time period and taxon, the recorder effort in each focal 10 km grid cell using the Frescalo approach[31]. Frescalo is a well-established method to model the data collection process of biological recording schemes[69,70]. It was developed to estimate unknown recording effort, species occurrence and temporal trends of data aggregated in time periods, such as when comparing atlases. Frescalo calculates standardized local frequencies of species within each grid cell using a neighbourhood around each focal grid cell. This neighbourhood is first based on geographic distance and then weighted according to landscape similarity to the focal grid cell (that is, based on the proportions of each land-use category in the 10 km grid cells, see below). Because Frescalo was designed for handling biological records in Great Britain at a 10 km resolution, we used the default settings to define the neighbourhood of each focal point (number of neighbours to include after ranking by distance, 200; and by landscape similarity, 100). For the butterfly and plant datasets, we calculated the recorder effort for each grid cell and time period, using the Frescalo implementation in the R package sparta[71]. Target values of $\varphi$ (weighted mean frequency of a well-recorded grid cell[31]) were set at 0.92 for both the butterfly and plant datasets, suggested automatically, on the basis of the properties of the data. Recorder effort was then computed as $1/\alpha A$, where $\alpha$ represents the sampling effort multiplier (the number of searches required in a grid cell to achieve a standard value of $\varphi$ for the neighbourhood[31,71]). For the bird dataset, another Frescalo-derived measure of recorder effort was already provided for each time period as the percentage of common benchmark species detected in each grid cell[31,72]. Metrics of recorder effort were incorporated in the corresponding taxon models described below.

#### Species richness, beta diversity and community temperature index

For each taxon, time period and grid cell, we calculated three metrics of biodiversity: species richness, beta diversity and CTI. Species richness refers to the number of species observed in a grid cell. Beta diversity was calculated as the community dissimilarity of each focal grid cell in relation to the eight surrounding grid cells, using the function beta.pair available in the R package betapart[73], based on pairwise between grid cell computations of the Sørensen dissimilarity index. CTI is an indicator of the relative community composition of warm- and cold-adapted species, used to quantify how communities are responding to temperature change[6,74]. CTI was computed as the average of the species temperature index (defined as the average temperature experienced by a species across its European range) in each grid cell. Species temperature indices for birds, butterflies and plants were taken from available datasets[6,75,76], covering all species retained for analysis (see above) except for 6 bird species and 80 plant species.

#### Land use, climate and microclimatic heterogeneity

Land-cover data for Great Britain corresponding to the 1960s species time period was extracted from the Land Utilisation Survey of Great Britain that took place in the 1930s to 1940s, producing maps at the 1:63,360 scale[77]. Land-cover data for the 1990s and 2010s time periods were extracted from the 1990 Land-Cover Map[78] and 2015 Land-Cover Map[79] (LCM) at 25 m spatial resolution. Digitized versions of the historical maps were taken from refs. 38,80, who classified the maps into five land-cover categories: arable, semi-natural grasslands (broad category including the land-cover classes defined as meadows, pastures, lowland and upland heathlands and open wetlands in the LCM 1990 and 2015), agriculturally improved grasslands, forest, urban and surface water at a 25 m resolution, providing proportions of each land-use category in the 10 km grid cells. Categories of the LCM maps were aggregated to match the five categories of the historical maps, with agriculturally improved grasslands, which had negligible cover until the late 1960s, added as an additional category (Fig. 1 and Supplementary Fig. 1). Although the historical land-use data predate some of the species' data by several decades, most of the major changes in land use, such as the widespread 'agricultural improvement' of pastures, occurred from the end of the 1960s[81]. Annual mean temperature and total annual precipitation were downloaded from the Met Office, UK[82], at 5 km spatial resolution and aggregated at 10 km to give the mean values within 1965–1975, 1985–1995 and 2005–2015, matching the three time periods of the biological data (Fig. 1 and Supplementary Fig. 1). To derive a proxy for grid cell level heterogeneity in temperature microclimates generated by topography, we used the averaged standard deviation of solar index values (2000–2010) at 10 km spatial resolution from the digital elevation model with horizontal resolution of 3 arcsec (~90 m) obtained from the Shuttle Radar Topography Mission[24]. Although the microclimate data used here represent only the modern time period, we consider these data adequate for use in all three time periods because there is very little difference in insolation regime over the duration of our study.

#### Statistical analysis

All statistical analyses were conducted with R v.4.1.1 (ref. 83). We ran all models using the integrated nested Laplace approximation (INLA) for Bayesian inference[33] and R-INLA (www.r-inla.org) for model execution[84,85]. INLA is a method that approximates Bayesian inference for latent Gaussian models such as the spatial and generalized linear models used in this study, where stochastic structures in the data need to be captured.

#### Differences in biodiversity between time periods

To investigate the difference (proportion of increase or decrease) in species richness, beta diversity and CTI between the 1960s and the 2010s (long term) and between the 1990s and the 2010s (short term), we fitted, for each taxon separately, temporal scale and biodiversity metric, one GLMM with the biodiversity metric acting as a response variable. For species richness, we fitted a Poisson GLMM (birds) and a negative binomial GLMM (butterflies and plants) with natural logarithm link function. For beta diversity and CTI we used a logit link function and a Gamma GLMM with natural logarithm link function, respectively. For each temporal scale, we only include in the analysis those grid cells that were recorded in both time periods (1960s and 2010s for the long term and 1990s and 2010s for the short term), which includes all 2,670 grid cells for the bird dataset; 2,666 for the plant data; and 2,013 and 2,022 for butterflies in the long and short terms, respectively. Then, for each model, we created a categorical variable $T$ with two labels: '1960s' and '2010s' for the long-term models; '1990s' and '2010s' for the short-term models. Labels '1960s' and '1990s' were used as reference level. In that way, the estimated $T$ coefficients would inform about the proportion of increase or decrease on biodiversity from the historical time period (1960s and the 1990s) to the modern time (2010s). To control for uneven recorder effort in the models for richness and beta diversity, we included the estimated recorder effort for the relevant time period as a variable in the logarithmic and natural scale, respectively[86] (removing the part of variation of the response captured by the recorder effort). For the CTI models, we did

not use the estimated recorder effort per se because CTI depends on the identity of the species configuring the community and a higher or lower recorder effort does not necessarily imply a higher or lower CTI. Therefore, we used a slightly different method and, as for British Lepidoptera[87] and for British invertebrates[20], setting a species richness grid cell threshold. We excluded from the analysis all grid cells containing <25% of the number of species recorded in the 'richest' grid cell during the period with the lowest estimated recorder effort, that is, 1960s (Supplementary Fig. 7). This allowed us to only include relatively well-recorded grid cells, with a minimum number of species needed for a meaningful estimation of CTI. For butterflies, the threshold was set at 10 species (that is, 25% of 39 species) leading to a dataset for the CTI analysis of 996 grid cells for the long term and 1,222 for the short term. For plants, the threshold was set at 214 species (that is, 25% of 854 species), leading to a CTI dataset of 2,351 and 2,406 grid cells for the long and short terms, respectively. For birds, the estimated recorder effort was classed as well surveyed and with similar coverage across years for all three atlases[72]. Thus, we did not conduct any grid cell cutoff for the bird data. To account for spatial autocorrelation, as ignoring it could lead to underestimating the uncertainty of the model predictions[88], we included a spatially structured random effect on grid (hereafter, $\gamma$). Parameter $\gamma$ has a variance–covariance structure that depends on a neighbourhood structure. We defined this spatial dependency matrix structure to the eight surrounding grid cells of each focal grid. We then assumed a complete spatial dependency between the focal grid and the eight surrounding grid cells, that is, intrinsic conditional autoregressive model structure (iCAR)[89] (Supplementary Note 'The iCAR model structure'). Nonetheless, for the Poisson and negative binomial GLMMs (species richness models) we wanted to have better control of the overdispersion. Hence, we included the spatial random effect on grid following a Leroux model structure[90]. The Leroux model is a generalization of the iCAR model, where the conditional distribution of the spatially structured random effects ($\gamma$, now referred to as $\upsilon$) is specified as in the iCAR model above but it also incorporates an exchangeable structure for the spatially unstructured residual (hereafter, $\nu$; where $\nu \approx$ Gaussian (0, $\sigma^2_\nu$)[90]. See Supplementary Note 'The iCAR model structure' for assessment of spatial dependency in the models. Our models for species richness are expressed as $SR_i \approx$ Poisson ($\mu_\wedge SR_i$) (for birds) and $SR_i \approx$ negative binomial ($\mu_\wedge SR_i, \phi$) (for butterflies and plants) with model equation

$$\log(\mu_\wedge SR_i) = \text{Intercept} + \eta \times T_i + \Omega \times \log(E_i) + \upsilon_i + \nu_i \quad (1)$$

For beta diversity the model is expressed as $BD_i \approx$ Beta ($\mu_\wedge BD_i, \varnothing$) with model equation

$$\text{logit}(\mu_\wedge BD_i) = \text{Intercept} + \kappa \times T_i + \Psi \times E_i \quad (2)$$

For community temperature index the model is expressed as $CTI_i \approx$ Gamma ($\mu_\wedge CTI_i, \theta$) with model equation

$$\log(\mu_\wedge CTI_i) = \text{Intercept} + \tau \times T_i + \gamma_i \quad (3)$$

The subscript $i$ refers to each one of the grid cells. SR, BD and CTI are the observed species richness, beta diversity and CTI with estimated means referred as $\mu_\wedge$ and the variances being: variance(SR) = $\mu_\wedge SR$ (for birds) and variance(SR) = $\mu_\wedge SR + (\mu_\wedge SR)^2/\phi$ (for butterflies and plants); variance (BD) = ($\mu_\wedge BD \times (1 - \mu_\wedge BD))/(\varnothing + 1)$; and variance(CTI) = $(\mu_\wedge CTI)^2/\theta$. Hyperparameter $\phi$ (variation-type parameter) describes the size of the negative binomial observations (1/overdispersion), while $\varnothing$ and $\theta$ are the precision hyperparameters of the Beta and Gamma distributions, respectively. Parameter $E$ refers to the estimated recorder effort and $\Omega$ and $\psi$ are the coefficients of $E$. The categorical variable is $T$ and $\eta$, $\kappa$ and $\tau$ are the coefficients (regression-type parameters) of $T$. The proportion of increase in

richness, biotic homogenization and CTI will be given by calculating exp ($T$ coefficient) and the proportion of decrease by 1 − exp ($T$ coefficient) (Fig. 2).

Even though we accounted for uneven and unknown recorder effort when evaluating differences in CTI between time periods, using a minimum species richness grid cell threshold cutoff, we could not discard the possibility that the surprising result of decreasing butterfly CTI over the short term was an artefact of recording bias. The CTI decrease could have been driven by under-recording in the 1990–1994 period in those areas of western Britain which show strong drops in CTI (compare Supplementary Figs. 6 and 7). To test this possibility, we replicated the analysis using the period 1995–1999 instead, as it had higher recorder effort (Supplementary Fig. 7). We found that the estimated average change of CTI also decreased over this period (−0.001 °C). However, in this case the 95% credible interval around the estimated change ranged from −0.003 to +0.000 °C, meaning that the possibility of no change at all or an increase in CTI over all Great Britain (probability of CTI increase = 0.64%, derived from the posterior) could not be discarded.

## Modelling associations between changes in biodiversity and changes of land use and climate

To investigate the associations between baseline and interacting effects of land-use and climate change on biodiversity change, we first calculated for each taxon, temporal scale and grid cell, the observed changes in biodiversity (species richness, beta diversity—as measure of biotic homogenization—and CTI), land use, climate and estimated recorder effort (2010s to 1960s for the long term; 2010s to 1990s for the short term). Then, for each biodiversity metric, taxon and temporal scale, we fitted one spatially explicit linear mixed-effects model with Gaussian distribution. The change in biodiversity acted as a response variable. As explanatory variables, we used change in temperature and precipitation; change in proportion cover of each land-use category; two-way interactions between each climate and land-use change variable; microclimatic heterogeneity; and baseline conditions of land use, climate and biodiversity (species richness and beta diversity; Supplementary Note 'Dealing with collinearity'). To control for uneven recorder effort in the species richness and beta diversity models, we included both the change in estimated recorder effort between time periods (to match the response) and the estimated recorder effort in the initial time period of change (as recorder effort was always found to be highest in 2010s; Supplementary Fig. 7) both in its natural scale[86]. In the CTI models, for butterflies and plants we applied the species richness grid cell threshold as described in the previous section and for all three taxa we included the change in estimated recorder effort (but not the initial value). All explanatory variables were standardized (mean = 0, s.d. = 1) to interpret their relative effect over the response. To account for spatial autocorrelation, we included a spatially structured random effect on grid ($\gamma$) modelled following an iCAR model structure as described in previous section. The model structure is expressed as:

$$C_i \approx \text{Gaussian} (\mu_\wedge C_i, \sigma^2)$$
$$\mu_\wedge C_i = \text{Intercept} + \mathbf{X}_i \times \boldsymbol{\beta} + \mathbf{E}_i \times \boldsymbol{\delta} + \gamma_i \quad (4)$$

$C$ is the change in biodiversity (species richness, beta diversity or CTI), with estimated mean = $\mu_\wedge C$ and residual variance (or error) = $\sigma^2$. $\mathbf{X}$ is the vector of explanatory variables and $\boldsymbol{\beta}$ the vector containing the coefficients of interest (Fig. 3). $\mathbf{E}$ is the vector containing estimated recorder effort and change of estimated recorder effort and $\boldsymbol{\delta}$ the vector of their coefficients. The posterior distribution of $\mu_\wedge C$ will inform about the probability of each grid cell to have experienced an increase or decrease of biodiversity and the uncertainty around it. To obtain this information, we subtracted the posterior distribution of $\mu_\wedge C$, calculated the average probability for the area under the curve

to be >0 (that is, estimated probability of biodiversity increase) ± s.e. (both in percentage) across all grid cells and captured the density distribution of $\mu_\wedge C$ through violin plots (Fig. 2). Because beta diversity is by definition spatially autocorrelated, we did not incorporate $\gamma$ in these models to avoid overfitting (Supplementary Note 'Assessing spatial dependency'). To make sure that the different structures of the models 1–3 (quantifying biodiversity change) and 4 (attributing change to environmental predictors) were statistically compatible, we compared predicted values of change for each taxon, time period and community metric; Supplementary Note.

When running spatial models, much of the variation might be captured by the spatially structured random effect, while little might be due to the fixed effects of interest. To assess how much variation of the response was explained by our models and captured by the fixed effects, we calculated conditional and marginal $R^2$ values. $R^2$ conditional considers the variance captured by both the fixed and the spatial random effects. $R^2$ marginal considers only the variance captured by the fixed effects[91]. To assess how much variation of the response was explained by each group of variables: changes in climate and land use, climate–land-use change interactions, baseline conditions of climate and land use, baseline biodiversity and microclimatic heterogeneity, we implemented an additional analysis. This analysis consisted of running the models in consecutive steps. First, we ran a model that only contained the spatially structured random effect + estimated recorder effort at the beginning of the period of change + change of estimated recorder effort. Then, we added a group of variables at a time, starting with changes in climate and land use and followed (in order) by interaction of climate–land-use change, baseline conditions of climate and land use, microclimatic heterogeneity and baseline biodiversity. At each step, $R^2$ conditional and marginal were calculated and the contribution of each group of explanatory variables, could be assessed (Supplementary Fig. 4). Recall that models for beta diversity did not contain spatial dependency and that models for CTI did not include baseline biodiversity.

### Revealing habitat characteristics associated with grid cell contribution to national beta diversity
To unravel which habitat characteristics were associated with grid cell contribution to national beta diversity, we first calculated for each taxon, grid cell and time period (1960s, 1990s and 2010s), the relative contribution of each grid cell to national beta diversity, that is, LCBD a comparative indicator of the ecological uniqueness of a site for its contribution to the overall beta diversity[32] using the beta.div function available in the R package adespatial[92]. Then, for each taxon and time period, we fitted one Beta GLMM with a logistic link function, to ensure that the fitted values ranged between 0 and 1. LCBD acted as the response variable. Climatic variables, proportion cover of each land-use category and microclimatic heterogeneity acted as explanatory variables (Supplementary Note 'Dealing with collinearity'). To control for uneven recorder effort, we included the estimated recorder effort for each time period in its natural scale[86]. All explanatory variables were standardized. To account for spatial autocorrelation, we included a spatially structured random effect on grid ($\gamma$) modelled following an iCAR model structure as described in the previous section (Supplementary Note 'Assessing spatial dependency').

The model structure is expressed as:

$$L_i \approx \text{Beta}(\mu_\wedge L_i, \varPhi)$$
$$\text{logit}(\mu_\wedge L_i) = \text{Intercept} + \mathbf{X}_i \times \boldsymbol{\beta} + \gamma_i \qquad (5)$$

$L_i$ refers to the LCBD in each grid cell, with estimated mean = $\mu_\wedge L$ and variance($L$) = $(\mu_\wedge L \times (1 - \mu_\wedge L))/(\varPhi + 1)$, where $\varPhi$ is the precision hyperparameter of the Beta distribution. $\mathbf{X}$ is the vector of explanatory variables and $\boldsymbol{\beta}$ the vector of the estimated fixed effects of the explanatory variables. Large values of LCBD indicate that a grid cell

has a strongly different species composition compared to a mean site and therefore makes a relatively large contribution to the total beta diversity across all Great Britain's grid cells. From a conservation perspective, high LCBD indicates: a site of special ecological conditions, with an unusual species composition and high conservation value; a degraded species-poor site but functionally unique species that might be a good candidate for ecological restoration; or the result of invasive species on communities[32].

### Model robustness estimating fixed effects
To evaluate the performance of our models (equations (4) and (5)) on estimating the effect of climate and land use on biodiversity in a consistent manner, we used a tenfold cross-validation technique. Cross-validation is a method for evaluating the ability of a model to be effective when presented with new inputs from the same distribution as the training data. When using the tenfold cross-validation technique, we randomly divided the data into ten subsets of equal size, use nine subsets to fit the model and the remaining subset to validate it. We repeated this procedure so that each one of the ten subsets was used once for validation[93] and assessed the performance of our models on estimating fixed effects by plotting the parameter coefficients of the fitted models against their correspondent from the tenfold cross-validation (Supplementary Figs. 2 and 5).

### Prior choice
Because we used a Bayesian approach, priors are required for the parameters in the models. For model equations (1) to (5), the priors for the fixed parameters and intercept were left as default settings in R-INLA, which follow a Gaussian distribution $N(0, \sigma^2 = 1{,}000)$ and $N(0, \sigma^2 = \infty)$, respectively. Priors for the hyperparameter $\phi$ (equation (1)) and precision hyperparameters $\varnothing$ and $\varPhi$ (equations (2) and (5)) and $\theta$ (equation (3)) were also set at their default settings, that is, log-gamma (1, 0.1). For the spatially structured hyperparameter $\sigma_Y$ (iCAR model structure in equations (3), (4) and (5)) and spatial hyperparameters $\sigma_v$ and $\sigma_v$ (Leroux model structure equation (1)), we used weakly informative penalized complexity prior distributions[94], recommended for spatial models with no available prior information. We provide full description of penalized complexity priors choice in the Supplementary Note .

### Reporting summary
Further information on research design is available in the Nature Portfolio Reporting Summary linked to this article.

## Data availability
BTO Bird Atlas data are available on request from (http://www.bto.org/datasets). The data for the Butterflies for the New Millennium recording scheme are available on request from the Butterfly Conservation (https://ukbms.org/request-data). Plant Atlas data are available on request from the Botanical Society of Britain and Ireland (https://bsbi.org/maps-and-data)[66]. Historical land-use data are available at https://doi.org/10.5878/9wks-qg91 refs. 38,80. Modern land cover data were extracted from the 1990 Land Cover Map (https://catalogue.ceh.ac.uk/documents/3d974cbe-743d-41da-a2e1-f28753f13d1e) and 2015 Land Cover Map (https://catalogue.ceh.ac.uk/documents/bb15e200-9349-403c-bda9-b430093807c7) at 25 m spatial resolution. Climate data were downloaded from the Met Office, UK (https://www.metoffice.gov.uk/research/climate/maps-and-data/data/index). The data for the birds' Species Temperature Index were requested from the authors of ref. 6; for butterflies, Species Temperature Index can be downloaded from GBIF (https://doi.org/10.15468/ug7pft)[75]; and for plants, from *Zenodo* (https://doi.org/10.5281/zenodo.1155850)[76].

## Code availability
The code used to run the analyses is available on request.

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

## Acknowledgements

This work was supported by the Swedish Research Council VR 2020-04276 to A.G.A. We would like to thank the many volunteers and the donors who supported the bird and plant atlases and the Butterflies for the New Millennium recording scheme. We would also like to thank O. L. Pescott for helping to manage the plant dataset and for the guidance on species inclusion and V. Devictor for providing the species temperature indices for birds.

## Author contributions

A.G.A., A.J.S and T.M.-J. conceived the study. R.F., B.M., D.B.R. and K.J.W. provided data. T.M.-J. analysed the data. T.M.-J., A.G.A., A.J.S, R.F., M.J., B.M., D.B.R. and K.J.W. interpreted results. T.M.-J. and A.G.A. led the writing. A.J.S., R.F., M.J., B.M., D.B.R. and K.J.W. contributed to the completion of the manuscript.

## FundingInformation

## Competing interests

The authors declare no competing interests.

## Additional information

**Correspondence and requests for materials** should be addressed to Alistair G. Auffret.

# Reporting Summary

## Statistics

For all statistical analyses, confirm that the following items are present in the figure legend, table legend, main text, or Methods section.

| n/a | Confirmed | |
|---|---|---|
| ☐ | ☒ | The exact sample size (*n*) for each experimental group/condition, given as a discrete number and unit of measurement |
| ☐ | ☒ | A statement on whether measurements were taken from distinct samples or whether the same sample was measured repeatedly |
| ☐ | ☒ | The statistical test(s) used AND whether they are one- or two-sided<br>*Only common tests should be described solely by name; describe more complex techniques in the Methods section.* |
| ☐ | ☒ | A description of all covariates tested |
| ☐ | ☒ | A description of any assumptions or corrections, such as tests of normality and adjustment for multiple comparisons |
| ☐ | ☒ | A full description of the statistical parameters including central tendency (e.g. means) or other basic estimates (e.g. regression coefficient) AND variation (e.g. standard deviation) or associated estimates of uncertainty (e.g. confidence intervals) |
| ☐ | ☒ | For null hypothesis testing, the test statistic (e.g. *F*, *t*, *r*) with confidence intervals, effect sizes, degrees of freedom and *P* value noted<br>*Give P values as exact values whenever suitable.* |
| ☐ | ☒ | For Bayesian analysis, information on the choice of priors and Markov chain Monte Carlo settings |
| ☐ | ☒ | For hierarchical and complex designs, identification of the appropriate level for tests and full reporting of outcomes |
| ☒ | ☐ | Estimates of effect sizes (e.g. Cohen's *d*, Pearson's *r*), indicating how they were calculated |

*Our web collection on statistics for biologists contains articles on many of the points above.*

## Software and code

Policy information about availability of computer code

| Data collection | No code was used for data collection. All data was downloaded from repositories accessible or upon request. |
|---|---|
| Data analysis | R version 4.1.1. was used to conduct the analyses, along with the following packages: Sparta (version 0.2.19); betapart (version 1.5.4); adespatial (version 0.3-14); INLA (version 21.02.23) |

For manuscripts utilizing custom algorithms or software that are central to the research but not yet described in published literature, software must be made available to editors and reviewers. We strongly encourage code deposition in a community repository (e.g. GitHub). See the Nature Portfolio guidelines for submitting code & software for further information.

## Data

Policy information about availability of data

All manuscripts must include a data availability statement. This statement should provide the following information, where applicable:
- Accession codes, unique identifiers, or web links for publicly available datasets
- A description of any restrictions on data availability
- For clinical datasets or third party data, please ensure that the statement adheres to our policy

BTO Bird Atlas data is available on request from (http://www.bto.org/datasets). The data for the Butterflies for the New Millennium recording scheme is available on request from the Butterfly Conservation (https://ukbms.org/request-data). Plant Atlas data is available on request from the Botanical Society of Britain and Ireland (https://bsbi.org/maps-and-data). Historical land-use data are available at http://doi.org/10.5878/9wks-qg91. Modern land cover data was extracted from

the 1990 Land Cover Map (https://catalogue.ceh.ac.uk/documents/3d974cbe-743d-41da-a2e1-f28753f13d1e) and 2015 Land Cover Map (https://catalogue.ceh.ac.uk/documents/cb84ee95-01e4-4d55-a33c-380fe01bc58d) at 25-m spatial resolution. Climate data was downloaded from the Met Office, UK website (https://www.metoffice.gov.uk/research/climate/maps-and-data/data/index). The data for the birds' Species Temperature Index were requested from the authors of https://doi.org/10.1038/nclimate1347; for butterflies, Species Temperature Index can be downloaded from GBIF (https://doi.org/10.15468/ug7pft); and for plants, are available from http://doi.org/10.5281/zenodo.1155850

## Human research participants

Policy information about studies involving human research participants and Sex and Gender in Research.

| | |
|---|---|
| Reporting on sex and gender | N/A |
| Population characteristics | N/A |
| Recruitment | N/A |
| Ethics oversight | N/A |

Note that full information on the approval of the study protocol must also be provided in the manuscript.

# Field-specific reporting

Please select the one below that is the best fit for your research. If you are not sure, read the appropriate sections before making your selection.

☐ Life sciences  ☐ Behavioural & social sciences  ☒ Ecological, evolutionary & environmental sciences

For a reference copy of the document with all sections, see nature.com/documents/nr-reporting-summary-flat.pdf

# Ecological, evolutionary & environmental sciences study design

All studies must disclose on these points even when the disclosure is negative.

| | |
|---|---|
| Study description | In this study, we use historical and modern datasets of land-use, climate and species observations from national atlas and monitoring schemes, to investigate how baseline and interacting effects of land-use and climate change drive biodiversity changes in British birds, butterflies, and plants over 50+ years (long-term, approx. 1960s-2010s) and 20 years (short-term, 1990-2010s) time-periods. The main dataset consists of 3,715,724 species occurrence records describing the British communities of breeding birds, butterflies, and plants, in the n = 2,670 10-km square grid-cells of the British National Grid. As biodiversity metrics, we use species richness, beta diversity and community temperature index. Our analyses are based on spatially-explicit generalized linear mixed models and spatially-explicit linear mixed models. We use integrated nested Laplace approximation for Bayesian inference. This method approximates Bayesian inference for latent Gaussian models such as the spatial generalized and mixed-effects linear models used in this study, where latent structures in the data need to be captured. Our models include controls for spatial autocorrelation, variation in recorder effort in space and over time, deal with confounding collinearity effects, and control for the effect of microclimatic heterogeneity in moderating climate-driven effects on biodiversity, and for baseline biodiversity conditions. |
| Research sample | Our study targets the communities of British breeding birds, butterflies and plants at three different time periods (i.e., 1960s, 1990s and 2010s). Specifically, we focus on local-assemblage at the 10-km square grid-cell resolution across Great Britain (i.e., 2,670 grid-cells). We use available datasets. For breeding birds, these data correspond to the atlases from 1968-72; 1988-91 and 2008-11. For plants, to the atlases of 1930-60; 1987-99 and 2000-19 (for which we retrieved the data referring to the period 2010-2019). For butterflies, we use the periods 1970-1974; 1990-94 and 2010-14 from the national recording scheme, Butterflies for the New Millennium, to match the other two taxa. We retrieved a total of 3,715,724 species occurrence records describing the British communities of breeding birds, butterflies, and plants. These available datasets provided us with species-occurrence data for all three taxons, across 2,670 10-km resolution grid-cells over all Great Britain. For each 10-km resolution grid-cell, taxon and time period (1960s, 1990s and 2010s), we calculated three measures of biodiversity: species richness, beta diversity and community temperature index. To calculate the community temperature index metric, we require the species temperature indices. For butterflies, these are available online at "Schweiger, O., Harpke, A., Wiemers, M. & Settele, J. CLIMBER: Climatic niche characteristics of the butterflies in Europe. ZooKeys 367, 65-84 (2014)". For birds and plants, species temperature indices are available upon request at "Devictor V. et al. Differences in the climatic debts of birds and butterflies at a continental scale. Nat. Clim. Change 2: 121-124 (2012)" and "Sparrius, L. B., van den Top, G.G. & van Swaay, C. A. M. An approach to calculate a Species Temperature Index for flora based on open data. Gorteria – Dutch Botanical Archives 40, 073-078 (2018)", respectively. As for environmental data, for each grid-cell, we retrieved 1) land cover data from the available repositories (Land Utilisation Survey of Great Britain, LUSGB, and the 1990 and 2015 1990 Land Cover Maps from the NERC Environmental Information Data Centre); 2) climate data downloadable from the Met Office (UK), available at 5-km spatial resolution and aggregated at 10-km to give the mean values of annual mean temperature and annual total precipitation within years 1965-75, 1985-95 and 2005-15, to match the three time-periods of the biological data; 3) heterogeneity in temperature microclimates available upon request from "Suggitt et al. Extinction risk from climate change is reduced by microclimatic buffering. Nat. Clim. Change 8, 713-717 (2018)." |
| Sampling strategy | For each taxon (i.e., birds, butterflies, and plants), we included native species (including some taxonomic aggregations and subspecies for plants) that represented stable taxonomic concepts across time from a biological recording perspective (i.e., they have |

been recorded consistently throughout the period of the study). A further description is specified in Methods. In total, 250 species of birds, 55 species of butterflies and 1,587 species of plants were retained for analysis.

**Data collection**

All data used in this study is freely available from on-line repositories or upon request from authors and organisations, and are described in Methods. Datasets were collated by the authors as specified in Methods.

**Timing and spatial scale**

This study concerns three time periods, i.e., 1960s; 1990s and 2010s. Observations of breeding birds, butterflies and plants, land-use and climate data are available for each time period and across the 2,670 10-km square grid-cells of the British National Grid included in this study.

**Data exclusions**

First, we excluded from the analysis those species that did not fulfil the criteria for study inclusion described above (in Sampling strategy) and specified in Methods. Second, because we are using atlas and presence only monitoring data, and we are interested in changes between time periods, for each taxon (birds, butterflies, and plants), we only included grid-cells recorded on both time periods, i.e., 1960s and 1990s for the long-term analysis; 1990s and 2010s for the short-term analysis. This covered all 2,670 grid-cells for the bird dataset; 2,666 grid-cells for the plant data; and 2,013 and 2,022 grid-cells for butterflies at the long- and short-term, respectively.

**Reproducibility**

All data used in this study is freely available from on-line repositories or upon request from authors and organisations. The results are fully reproducible by following the modelling description in the Methods section.

**Randomization**

Randomization is not applicable in this study. All data used in this study is already collected and available upon request. However, we had to control for 1) uneven and unknown recorder effort of the monitoring and atlas data collection, to avoid biased results, and 2) spatial dependency, as ignoring it could lead to underestimating the uncertainty of the model predictions. To control for uneven and unknown recorder effort, we estimated for each time-period and taxon, the recorder effort in each focal 10-km grid-cell using the Frescalo approach (further describe in Methods). For the community temperature index analysis, we used a species-threshold cut off similarly to "Macgregor, C.J. et al. Climate-induced phenology shifts linked to range expansions in species with multiple reproductive cycles per year. Nat Commun 10, 4455 (2019)" and "Platts, P.J. et al. Habitat availability explains variation in climate-driven range shifts across multiple taxonomic groups. Sci Rep 9, 15039 (2019)". To account for spatial dependency, we used a Leroux model (which allows the structured part of the spatial residuals to be part of the parameter space, detaching it from the unstructured spatial random effect) and an Intrinsic Conditional Auto-Regressive model (i.e., a random effect with spatial dependent structure - iCAR model) as specified in Methods. We defined the spatial dependency matrix to the eight surrounding grid-cells of each focal grid-cell for both Leroux and iCAR models.

**Blinding**

Not applicable in the analysis of this study. This study is based on records of birds, butterflies and plants as well as on land-use and climate data that has already been collected and is available on online repositories or upon request.

Did the study involve field work? ☐ Yes ☒ No

# Reporting for specific materials, systems and methods

We require information from authors about some types of materials, experimental systems and methods used in many studies. Here, indicate whether each material, system or method listed is relevant to your study. If you are not sure if a list item applies to your research, read the appropriate section before selecting a response.

## Materials & experimental systems

| n/a | Involved in the study |
|-----|----------------------|
| ☒ | Antibodies |
| ☒ | Eukaryotic cell lines |
| ☒ | Palaeontology and archaeology |
| ☒ | Animals and other organisms |
| ☒ | Clinical data |
| ☒ | Dual use research of concern |

## Methods

| n/a | Involved in the study |
|-----|----------------------|
| ☒ | ChIP-seq |
| ☒ | Flow cytometry |
| ☒ | MRI-based neuroimaging |

