## [Peer Review File · Nature Ecology & Evolution]

Peer Review Information

Journal: Nature Ecology & Evolution

Manuscript Title: Anthropogenic climate and land-use change drive short and long-term biodiversity shifts across taxa

Corresponding author name(s): Alistair G. Auffret

Editorial Notes:

Reviewer Comments & Decisions:

Decision Letter, initial version:

16th January 2023

Dear Dr Montràs-Janer,

Your Article entitled "Anthropogenic climate and land-use change drive short- and long-term biodiversity shifts across taxa" has now been seen by 3 reviewers, whose comments are attached. In the light of their advice, we have decided that we cannot offer to publish your manuscript in Nature Ecology & Evolution.

From the reports, you will see that while they find your work of some potential interest, the reviewers raise concerns about the advance your findings represent over earlier work and the strength of the novel conclusions that can be drawn at this stage.

In particular, although all three reviewers agree that scale of the data are valuable and impressive, they all raise major concerns over different aspects of the analysis -- in particular, Referees #1 and #2 question the robustness of the models and treatment of uncertainty, while Referee #3 feels that in the absence of a deeper exploration of the links between climate and land use interactions, they were not convinced that the approach draws sufficiently novel conclusions to justify publication in this journal.

Having taken into account all the comments, and in light of the expertise of each referee, we feel that these criticisms are sufficiently important as to preclude publication of your work in Nature Ecology & Evolution.

Although we cannot offer to publish your paper in *Nature Ecology & Evolution*, I have consulted with two other journals in the Nature Portfolio, who have both expressed an interest in continuing the review process:

1) I have consulted with Dr. Luke Grinham at Communications Biology, a selective open access journal from Nature Portfolio. **They are very interested in your study and would be delighted to send a suitably revised manuscript back to the original referees, should you transfer it to them.** Specifically, they ask that you address the reviewer comments with regard to technical about the modelling approaches and results. To have your revision sent back to review by the editors at Communications Biology, please use the link to the manuscript transfer service in the footnote below. If you would like to discuss your manuscript with them directly, you can contact them at (luke.grinham@nature.com)

2If you decide to submit your revised manuscript to Communications Biology, please ensure that your revision is accompanied by a detailed point-by-point response to all the reviewer comments and a cover letter summarizing the changes made to the manuscript. Please also include completed [Life Sciences Reporting Summary](https://www.nature.com/documents/nr-reporting-summary.pdf) and [policy checklist](https://www.nature.com/documents/nr-editorial-policy-checklist.pdf) forms and modify your manuscript if needed, to comply with the instructions in these documents.

Communications Biology is a fully open-access journal and an article processing charge will apply to any papers accepted for publication. For further details about Communications Biology, including their editorial policies and information about open access funding, please visit the journal website at <https://www.nature.com/commsbio/>.

2) I have also consulted with my colleagues at *Communications Earth & Environment*, a selective, open-access Nature Research title led by an in-house editorial team that publishes research bringing new insight into a focused area of the Earth, environmental or planetary sciences (www.nature.com/commsenv/).

They have also agreed that they would send a suitably revised manuscript back to the reviewers, if you can address the outstanding issues in full.

Communications Earth & Environment (www.nature.com/commsenv) is a new open access journal that publishes high-quality research of relevance to the community of Earth and environmental scientists and is led by Nature Research professional editors. If you have any questions about the journal, Chief Editor Dr Heike Langenberg leads the editorial team and would be happy to answer them (heike.langenberg@nature.com).

I am sorry that we cannot be more positive on this occasion, but hope that you find the reviewers' comments helpful when preparing your paper for resubmission elsewhere.

[REDACTED]

Reviewer expertise:

Reviewer #1: Climate, land-use, biodiversity

Reviewer #2: Biodiversity change

Reviewer #3: Climate, land-use, biodiversity

Reviewers Comments:

Reviewer #1 (Remarks to the Author):

In this manuscript, the authors combine information on biodiversity records and historical land-use and climate change in Great Britain, to assess whether changes in British birds, plants and butterfly assemblages are affected by land use, climate change and their interaction. The authors investigate changes over two time periods, and also consider the effects of microclimatic heterogeneity and baseline biodiversity conditions on biodiversity indices. I found this study interesting and I believe it contains a lot of results that are relevant for field. I do, however, have major concerns regarding some aspects of the analyses, of the presentation of the work, and of interpretation of the results. I believe some improvements are necessary, and I have made some suggestions below which will hopefully be helpful to the authors.

Major

- A finding of your work, that you highlighted as surprising and for which you provided possible explanations in the discussion (lines 229-235)- was the decrease in butterfly CTI over the short term. However, reading the methods section, it seems to me that the confidence attributed to these results is not high, for two reasons that you highlight: because of possible recording biases (lines 570-580, with CI spanning 0 when replicating the analysis over a different time period, line 578) and also because of possible collinearity issues (lines 670-675). So, I can't help but question whether you should exclude the corresponding models and results from your work, as they come across as possibly not robust. (On the basis of the same arguments you could make the case that you're excluding these models).
- Lines 78 and line 84 – “baseline and interacting effects”: does this mean you considered the main effects of land use, climate, and their interactions, as well as their baseline values? I think what the baseline conditions are needs to be defined in a clear way in the main text, given their importance in your analyses and results. For instance, it isn't clear from the main text (line 84) that baseline biodiversity was considered.
- It is not clear to me whether investigating the effects of microclimatic heterogeneity on biodiversity change was one of the main aims of the study, or whether you just needed to control for microclimatic heterogeneity (lines 93/94), so I suggest clarifying this point.
- I think parts of the Methods need to be made clearer (see points below). I found that some places a bit difficult to follow – in particular, the first point of the paragraph “Dealing with collinearity” line 643. Given that it is an important part of the methods describing how you treated and included the models' explanatory variables, I think it is important for this section to be very clear.
- Although I do appreciate the synthesis effort with figures 2, 3, 4, I would have liked to see some of the effect sizes. I wonder if you could change e.g. figure 2 to display the effects from the models (effect sizes for figure 3 and 5 might be harder present synthetically?)
- I suggest highlighting some of the main limitations of your study in the discussion part.

3Minor

- Line 54 and throughout the manuscript, I recommend hyphenating "land-use" only when used as a compound adjective (land-use change), not hyphenating otherwise.
- Line 27: "land-use change is generally inferred using proxies"- this is unclear to me. Do you mean impacts of land-use change on biodiversity, or land-use change itself? If so, what type of proxies?
- Line 33: maybe specify what these baseline environmental conditions are
- Line 37: "lower rates of biodiversity loss": one of the main results presented in the abstract is increases in species richness, which seems to contradict this statement partly
- I wonder if you could conclude the abstract with a few words highlighting some important implications of your work.
- Lines 65-69 and 224: Also see the work by Williams et al. (2019) on CTI in different land-use types: <https://onlinelibrary.wiley.com/doi/full/10.1111/ecog.04806>
- Line 68: I suggest highlighting that this is due to land use affecting local, microclimatic conditions to clarify the distinction between climate warming and microclimate
- Line 74: what kind of proxies? Maybe add a few examples that can highlight how different your study is from these other studies based on proxies.
- Line 77: somewhere in the main text, I suggest displaying the sample sizes (number of considered species) – (or maybe in one of the figures?)
- Line 85: what is biodiversity resilience here? Do you focus on a specific metric?
- Lines 87 and 88: mixed-"effects" models. You could merge for brevity (e.g. "our analyses of biodiversity change were based on spatially explicit (generalised) linear mixed-effects models" or something along these lines).
- Line 93/94: do you control for these in all the models, or just the models built to investigate point (2)? Regarding baseline biodiversity conditions, I think you've already said you investigate their effects line 83/84?
- Line 100: unclear to me what is meant by "loss of spatial heterogeneity" here
- Line 102: this is not the first occurrence of CTI (first occurs line 83); I think you also need to explain a bit more in the main text what the thermal niche of a species is here (at present, the definition of CTI is a bit vague in the main text – although it is defined clearly in the methods).

4- Line 106-108: what do these results mean?
- Line 112: "were consistent with a response to anthropogenic environmental change" I'm not sure what is meant by that
- Line 122: "strong determinants" & line 189 "driving mechanisms" – I suggest reporting results as correlative, because your study doesn't investigate mechanisms
- Line 124: "and the changes that occurred" here, do you mean changes in environmental conditions?
- Line 136 "enhanced stability": how did you determine that? What is your definition of stability (and also resilience, elsewhere)
- Line 141: "improved and semi-natural grasslands" here, as baseline conditions, right?
- Line 197 – delete the d in "land-use changed"
- Line 209-221 I've enjoyed reading this paragraph of the discussion. I wonder if there is something to be said about invasive species as well? Could observed increases in species richness partly be explained by biological invasions?
- Line 461 – Do you mean that for the 1960 time period you approximate land use from the 1930s survey? Are there any drawbacks from using a 1930 survey for a period of time centred around the 1960s?
- Line 499 – delete the d in used
- Line 513 – I'm not familiar with the Leroux model or the iCAR model to control for spatial autocorrelation. Why were these different approaches used for the species richness model and for the beta-diversity and CTI models (in models 1, 2 and 3)? And why was the iCAR model then used for the 18 other models (line 606), and then also in the last model (line 696)?
- Line 522 add an "s" to Poisson
- Line 536 and where relevant throughout the manuscript, there is a mix of Greek letters spelt out (e.g. phi) and written in the Greek alphabet (e.g. ϕ), I suggest homogenising throughout
- Line 542 & 544 add an s to refer and show
- Line 543: I suggest merging this paragraph with the paragraph on spatial autocorrelation line 513. Also, I don't think the following paragraph describes the "assessment for the inclusion of spatial autocorrelation" as you state? To me that following paragraph describes some fine-tuning aspects of the approach. What was, then, the choice to include spatial autocorrelation based on?
- Line 588 – mixed-"effects"

- Why is recorder effort including differently in the first sets of models (1,2,3), in the second set of 18 models (597-602), and in the last model (line 693)?
- Line 628 "and interpreted the results with caution" I'm not sure what this means for the results
- Paragraph "dealing with multicollinearity", in particular point 1: line 655: I'm not clear with what "semi-natural grassland remained unregressed" mean. In my understanding, you used the variation in arable, forest, and urban covers not explained by semi-natural grassland cover (by using the residuals). So why not also include semi-natural grassland in the models then? Residual covers of arable, forest and urban should not then be collinear with semi-natural grassland cover?
- Line 683- "a species-poor site", with few but functionally unique species?

Reviewer #2 (Remarks to the Author):

please find the attached pdf.

Reviewer #3 (Remarks to the Author):

Anthropogenic climate and land-use change drive short- and long-term biodiversity shifts across taxa

Montràs-Janer et al.

Manuscript number: NATECOLEVOL-221117943-T

In this well-written manuscript, the authors use a large data set on bird, butterfly and plant occurrences in Great Britain to assess, 1) how richness, beta diversity and the community temperature index of these taxa have changed over two different time periods; 2) whether these changes are driven by concurrent changes in environmental variables (climate and land cover), starting conditions at the beginning of the investigated time periods ("baseline" conditions of climate, land cover and biodiversity), by microclimatic heterogeneity or interactions between some of those variables; 3) which of these climate and land cover variables affect the grid-specific contribution to national biodiversity. The results show, that both climate and land-cover change led to increased richness, biotic homogenization and communities with more warm-adapted species. With a large part of the biodiversity change driven by baseline environmental conditions, the authors assume that observed changes may be lag-effects, derived from past changes, such as the removal of semi-natural grasslands shown to contribute strongly to overall biodiversity and linked to lower biodiversity loss across time. Despite this being an exceptionally valuable data set, these findings are not novel, and – unfortunately – the one aspect which would have set this study apart from similar studies was mostly neglected in most parts of the paper: how climate and land use may interact in shaping biodiversity

6patterns.

Still, the authors took great care to preselect and standardise the available data to avoid bias. Considering all provided material, the authors give sufficient information that should allow reproducing the results. And even though I have not used the integrated nested Laplace approximation for Bayesian inference myself, the selection of models and the methods applied to dealing with spatial autocorrelation, recorder effort and collinearity seem to be appropriate. However, the figures in the main paper do not show how the data was distributed, how large effects were and which uncertainty they had. And not all of this information is provided in the supplement either (see individual comments to figures and tables below) and/or the descriptions in the legend are missing. Reading the main text, I had to repeatedly refer to the extended methods section or even the supplement to really understand the research questions, the study design, the statistical methods and even some of the graphs (which are mostly simplified summaries of complicated result tables, but not necessarily easy to understand or interpret without seeing the raw data).

Looking at Figure 1 in the supplement I also wonder how meaningful some of the effects are ecologically, because mean values seem to be rather similar for most of the taxa and biodiversity measures. After all, even small differences may turn out to be "significant" if the sample size is large enough (which it probably is here).

All in all, I can see the effort put into this manuscript and the great potential of the data set used, but I do not think this is fully reflected or utilized in the manuscript yet. In particular, one of the main research goals (climate-land-use interactions) was neither sufficiently presented nor discussed.

General comments

Language:

The authors switch between the terms "land-use vs land use", "climate-change vs climate change", "time-periods vs time periods", "land-cover vs land cover". I personally prefer "land use", "climate change" etc unless used in combination with another noun ("e.g. land-use types"). Whichever way it is done, be consistent!

I was struggling a bit with the terms/phrases "(at the) long-term" and "(at the) short-term" repeatedly used throughout the manuscript. Reading this sounds a bit strange sometimes, as if it wasn't the right word/phrase to use.

Title

The title captures the main findings.

l. 1: „... drives" not „drive"?

Summary paragraph

The summary captures the main aspects of the manuscript, yet also puts a lot of emphasize on climate-land use interactions. The resulting expectations of the reader are not met within the rest of the paper (at least in terms of results and discussion, where potential interactions are hardly mentioned)

l. 30: here and throughout change „land-use" to „land use"? unless used in combination with another noun, e.g. "land-use data"

7- l. 33. Unclear, what "baseline environmental conditions" are (need to read methods for that)
- l. 37: mentioning "heathlands" next to semi-natural grasslands, even though it was only one of the land-use types summarised in the category "semi-natural grasslands" unjustifiably puts a lot of effort on that specific land-cover type

Main body

- l. 54: replace „further change“ with „future change“
- l. 57: "has intensified"
- l. 61: "reorganisation at the community level" is a repetition to l. 55-56 ("reorganization of ecological communities")
- l. 62-62: rephrase the sentence "and how they have interacted is a key unknown in studies of environmental change impacts on biodiversity."
- l. 70: change "climate change effects" to "climate-change effects"
- l. 79-80: I suggest to slightly rephrase "... over 50+ years (long-term, approx. 1960s-2010s) and 20 years (short-term, 1990s-2010s) time-periods (Figure 1)" ◊ "...over two different time periods (Figure 1): 50+ years (long-term, approx. 1960s-2010s) and 20 years (short-term, 1990s-2010s)"; change „time-periods“ to "time periods" here and throughout
- l. 83-84: "2) associations of..." this half-sentence is unclear
- l. 93: the microclimatic data is from time 3, and therefore control for effects of the most recent microclimatic conditions. This should be made more clear
- l. 94: unclear what "baseline biodiversity conditions" means in this context (unless one refers to methods at this point)
- l. 105: "examination" instead of "interrogations" (as the latter sounds like the questioning of a crime suspect)
- l. 106: "most taxa" would be more precise than "all" taxa"; also, the use of "time period" for both the "point in time" (1960s, 1990s, and 2010s) and the "long- and short-term difference between time periods" (1960-2010 and 1990 to 2010, see for instance l. 111) is somewhat confusing.
- l. 113: whether the 1930s are included or not, the time period studied here covers at least 60 years (not 50 years)
- l. 114: again, it is unclear what environmental baseline means in this context
- l. 118: "at the expense of" instead of „at expenses of“
- l. 130: "land-cover classes" instead of "land cover classes"
- l. 159-162: "For butterflies..." this sentence is quite complicated and takes some time wrapping your head around
- l. 167: "associated with"
- l. 196: "a clear continuation of community reorganisation" is quite complicated, rather say "where communities are continuously reorganised"
- l. 197: "change" instead of "changed"
- l. 231 – 242: this is the only section where the names of authors from other papers are mentioned in the text (e.g. Devictor et al.). For consistency, I would remove those.
- l. 239: "resulting in..." change to "resulting in increased richness and beta diversity at the long-term, but decreased beta diversity and no clear increase in richness at the short-term."
- l. 271 – 272: "for example,..." the examples listed here surely determine shifts in community richness, but not all of these were studied by the authors. I personally would prefer to have those listed that were actually found to play a role. Of course, "topography" does relate to microclimatic variability, but

without reading the methods this is not clear
l.274: "to conservation that" remove comma

Methods:

Most of the issues I had with understanding the main text were resolved when reading the methods, as they nicely and in detail describe what has been done. For instance, the Frescalo approach to estimate unknown recording effort to standardize data is a great way to account for recording bias. The authors spent quite some time to explain the models, and adjustments for spatial autocorrelation and collinearity seem appropriate considering the limitations of the data, which are also nicely described in the methods section

l. 393: for plants, the years of the atlas (1930 – 1960) do not match Figure 1 (1930 – 1969)

l. 394: for consistency write "1970-74"

l. 396 – 397: here the authors clearly state that, for simplicity, the time periods of sampling are afterwards referred to as 1960s, 1990s etc., but without reading the methods this is not clear in the main text

l. 405: remove comma after "therefore"

l. 461: here it is stated the land cover that is from the 1930s, in the caption of Fig. 1 it says 1930 – 50 and here 1930s. Clarify and be consistent. How may the fact, that the land use data for the time 1960s is from the 1930s may have affected the results?

l. 463: here and below it says "2015 Land Cover Map", while in Fig. 1 it states 2014-15

l. 475 – 478: while these lines nicely describe how and for which time period microclimatic heterogeneity was calculated, the main text does not provide sufficient information to convey that knowledge, which makes it hard to understand the meaning and potential importance of that variable

l. 493: "recorded in both periods"

l. 499: "we did not use"

l. 567: remove comma after U ("...U did not overfit")

l. 572: remove comma "...short-term analysis was an..."

l. 589 – 593: here, the structure of the model is quite clear and nicely explained. In addition, the authors explain what they consider to be "baseline conditions", i.e. the climate, land use and biodiversity status in the earlier time period considered in the biodiversity change models. This should also be clarified in the main text

l. 628: I can see where the autocorrelation issue with beta diversity is coming from and acknowledge that the authors aim to "interpret with caution". However, solely based on the main text this caution is not transferred to the reader. The same applies to the issue of collinearity.

l. 689: remove comma "...time period a spatial..."

l. 712: remove underline "2021"

References:

The literature covers a good selection of previous literature that support the findings.

l. 352: First author name and title of article are in italics

l. 386: Journal name is not italics

l. 742: Journal name is not italics

l. 744: Full stop missing after article title

Tables and Figures

Fig. 1: Without reading the methods, it is somehow unclear that the maps shown in Figure 1 a) and b) are based on data collected within the time ranges listed below in c). And it is not clear that the changes shown for temperature and precipitation are based on the difference between the averaged values for the time periods listed in c). It would also be nice to mention in the main text (and possible caption) that the terms "1960s", "1990s" cover all the ranges listed in c). In addition: why are the three time points shown for land-use change (a) and not the change across the long-term and short-term time periods, as for climate?

Fig 1 caption: as elsewhere "land-use" vs "land use", "time-periods" vs "time period"; I suggest to use different letter ("c") for the panel about precipitation

l. 856: "increase" not "increased"

l. 857: "at the expense of" instead of "at expenses of"

l. 860: "for breeding birds"

l. 863: "(1930 – 1950)" for consistency; also, that does not match the figure, which shows that land use data for time 1 was from 1930 only

Fig. 2: This figure is quite straightforward and easy to understand, although I personally would have preferred to see the actual mean values and credible intervals using the same colour code as in Supplementary table 1, as it also provide a measure of differences in effect size.

l. 868 - 869: "(i.e., 95% credible interval within the positive or negative range)" I know the 95% credible interval indicates that the effect has a 95% probability of falling within the given range, but this may not be clear to everyone based on the wording.

l. 869: "bird species" instead of "birds species"

Fig. 3: I do not find this graph to be very intuitive to understand

l. 881: "the latter" instead of "the former", as this refers to microclimate? It is also not clear, that the microclimate was derived within the last period time 3

Fig. 4: I do not find this figure informative enough to be included in the main text. I would move it to the supplement. Also, I have some issues with connecting the R-square values for models with and without baseline biodiversity data with lines, because this indicates a change (such as a change in space or time), which clearly is not the case here. Using unconnected symbols would have been better.

Fig. 5: Again, this is a simple illustration of the main findings without providing any detail that allows comparing the effect of different variables without having to consult the supplement. The exception hereby is the effect of semi-natural grasslands, for which mean effect values are shown. Why not for the other variables then? Also, microclimatic heterogeneity here is referred to as "micro buffering temperature" which has not been used elsewhere (adapt that)

Supplementary material

Table 1: I like that table and the comparison with the model without spatial dependency. It is a nice way of showing the magnitude and direction of effects, although the values do seem very small, especially for CTI.

Table 2: Again, this is a neat overview of the main results, which translates into the colour coded Figure 3 presented in the main text. The table is rather complicated, and still some important information (here 95% credible intervals) are missing. The length of the coloured bars do not seem to be comparable across the whole table, which makes the bars less useful. Also, this table includes stats missing from Fig 3: the interactions.

Table 3 + 4 captions: "...based on model equation..."

10Table 5: abbreviation of the time periods in caption (T1, T2, T3) does not match abbreviation in figure (time1, time2, time3). For consistency, the first column listing explanatory variables should not be coloured, as in table 1 and 2. Also, a different term for microclimatic heterogeneity is used in this table (micro buffering). This should be adapted

Table 6: do the numbers in this table really indicate the Spearman correlation coefficient, as indicated in the table caption? In that case I am surprised that correlations with very low correlation coefficients (e.g. -0.049 for effect of forest on short-term loss of bird richness) have p-values below 0.05. Also, I find it hard to visualise the relationships based on this figure

Fig. 1: the plots show the distribution of the data, however, apart from the upper two panels for butterflies it is difficult to see the effect suggested by the table. What is the actual difference in e.g. bird species richness between time 1 and 3? Based on the graph richness seems to be identical. I wonder if, due to the relatively large sample size (e.g. 2,670 grid cells for birds) even very small differences in richness will turn out to be significant (95% credible interval not including zero), yet does that mean the effect is biologically significant? Also, the different aspects of the boxplot/violinplot are not described in the figure caption (e.g. center line, median; box limits, upper and lower quartiles; whiskers, 1.5x interquartile range; points, outliers).

Fig. 2: Some of these panels are quite useful. E.g. it is nice to see the different proportion of each land cover, and their change between time periods (which is negligible in some cases). The meaning of panel b) is not that clear, especially of the third arrow (53%).

Fig. 3: Label taxon-groups (birds, butterflies, plants) with a, b and c. Refer to labels in caption

Fig. 4: this graph makes Figure 4 in the main text redundant. Still, I would not use a line to link the different R-squared values, as the x axis does not represent a continuous variable and therefore an increasing response.

Fig. 6: I think these kinds of scatter plots with trend lines are much easier to understand than tables or even the colour graphs used in the main text. They also nicely show the distribution of data points. I can see that it is tricky to include all of those graphs in the main text, as there are too many results. But the main focus of the paper was on interactions (according to the knowledge gap identified in the introduction), and visualising those would have helped! Also, it ought to be mentioned in the caption that only graphs for explanatory variables with „significant effects“ are shown.

** *Communications Biology* is a selective Nature Portfolio title publishing Open Access research that brings new insight in all areas of the biological sciences. [Additional journal metrics and information can be found here](https://www.nature.com/commsbio/journal-information/journal-impact). Their editors prioritise good author service, fast peer review (in 2021, the median time to decision after first review was 40 days), and are happy to answer any questions you may have commsbio@nature.com. The journal has an Impact Factor of 6.548, a CiteScore of 6.0 and a Scimago quartile ranking of Q1.

Please note that *Communications Biology* is a fully open-access journal and an article

11processing charge will apply to any papers accepted for publication. Our [open access pages](https://www.nature.com/commsbio/about/open-access) contain information about article processing charges, open access funding, and advice and support from Springer Nature.

If you wish to transfer your manuscript to *Communications Biology*, please use our manuscript transfer portal using the link below to initiate the transfer to this journal (or to another journal of your choice in the Nature Research portfolio). If you transfer to Nature-branded journals or to the Communications journals, you will not have to re-supply manuscript metadata and files. This link can only be used once and remains active until used. For more information, please see our [manuscript transfer FAQ](https://www.nature.com/nature-portfolio/for-authors/transfer) page.

**** *Communications Biology* will send your work for peer review if you choose to transfer. *Communications Biology* is a selective Nature Portfolio title publishing Open Access research that brings new insight in all areas of the biological sciences. [Additional journal metrics and information can be found here](https://www.nature.com/commsbio/journal-information/journal-impact). Their editors prioritise good author service, fast peer review (in 2021, the median time to decision after first review was 40 days), and are happy to answer any questions you may have commsbio@nature.com. The journal has an Impact Factor of 6.548, a CiteScore of 6.0 and a Scimago quartile ranking of Q1.**

Please note that *Communications Biology* is a fully open-access journal and an article processing charge will apply to any papers accepted for publication. Our [open access pages](https://www.nature.com/commsbio/about/open-access) contain information about article processing charges, open access funding, and advice and support from Springer Nature.

You may transfer your manuscript to *Communications Biology*, please use our manuscript transfer portal using the link below to initiate the transfer to this journal (or to another journal of your choice in the Nature Research portfolio). If you transfer to Nature-branded journals or to the Communications journals, you will not have to re-supply manuscript metadata and files. This link can only be used once and remains active until used. For more information, please see our [manuscript transfer FAQ](https://www.nature.com/nature-portfolio/for-authors/transfer) page.

****I have spoken with my colleague at Communications Biology, and they have agreed to review your manuscript there. To transfer your manuscript please use our [manuscript transfer portal](https://mts-natecolevol.nature.com/cgi-bin/main.plex?el=A2Cn7HCH2A5BUtp6X4A9ftdRH1rFEcnvbR7krWFQjxPQZ). You will not have to re-supply manuscript metadata and files, unless you wish to make modifications, but please note that this link can only be used once and remains active until used. For more information, please see our [manuscript transfer portal](https://mts-natecolevol.nature.com/cgi-bin/main.plex?el=A2Cn7HCH2A5BUtp6X4A9ftdRH1rFEcnvbR7krWFQjxPQZ).**

12http://www.nature.com/authors/author_resources/transfer_manuscripts.html?WT.mc_id=EMI_NPG_1511_AUTHORTRANSF&WT.ec_id=AUTHOR>manuscript transfer FAQ page.

Note that any decision to opt in to In Review at the original journal is not sent to the receiving journal on transfer. You can opt in to *In Review at receiving journals that support this service by choosing to modify your manuscript on transfer. In Review is available for primary research manuscript types only.*

** For Nature Research general information and news for authors, see <http://npg.nature.com/authors>.

Reviewer #2 attachmentReview report on "Anthropogenic climate and land-use change drive short- and long-term biodiversity shifts across taxa"

The submitted article analyses long- and short-term changes in biodiversity for Birds, butterflies and plants in relation to land-use and climate change. Being very much aware of the tremendous efforts required to gather quality data, extensive spatio-temporal coverage for biodiversity, as well as land use and climate change data, I am hesitant to say but wonder if the present study stands upon reasonable data analyses and modelling grounds, while reading the description in the manuscript. This leads to several concerns regarding the utility and appropriateness of the models presented, which would be better to be clarified.

I might have just misunderstood the documentation because of missing some technical clarity and limited notation in the manuscript. However, I still believe that models are crucial, even though the manuscript adopts some standard models, to capture spatio-temporal trends of biodiversity. Also, it would be great if the authors could improve the manuscript's clarity from my misinterpretation. Below, I provide my comments and concerns that are purely based on what I read and how I interpreted the manuscript description, hoping that this helps the authors.

Two-steps modelling

To evaluate the association of the changes in biodiversity to land-use and climate change, the authors have taken two modelling steps: 1) fit a model to each biodiversity index; and 2) fit its changes to land-use and climate change. I am concerned about how the model uncertainty in the first step is inherited in the second step and wonder why this is not evaluated in the manuscript. This is simply because ignoring the model uncertainty induces over-confidence in the second step. Since the authors formulated the models in Bayesian setting, such a flexible modelling framework, it looks more reasonable to combine these two steps into a single model and evaluate the underlying uncertainty. In particular, in the first step, the models (1)–(3) contain time as an explanatory variable representing temporal trends, so I do not see the point of splitting the modelling steps. Is there any particular reason?

Different model fitting for different time durations

Again to evaluate the association of the changes in biodiversity to land-use and climate change, the authors fit different models between the long-term (2010s-1960s) and the short-term (2010s-1990s) periods. Although the model structure is the same, different parameter estimates mean that there are two models for the same overlapped period (the short term is a subset of the long term). This indicates model inconsistency. I wonder how such a discrepancy between the models would be interpreted and if such a separate model-fitting exercise is necessary.

Recorder effort

In models (1)–(2), the recorder effort is included as an explanatory variable E_i with a coefficient to be estimated. If this is meant to be a bias correction term that reduces biases due to different recording efforts, it would be more natural to set the effort as an offset term that does not possess any coefficient. It would be very much appreciated if the authors could elaborate a little bit more on why the coefficient is needed and its interpretation in the model.

14

Different spatial random structures in Richness

It would be very much appreciated if the authors could elaborate more about u_i and v_i , contrasting Y_i , and why the random spatial structure should be different between Richness and the other two, beta diversity and community temperature index. Would this be simply due to Leroux model? Related to this point, the role of the error term ε_i is a bit unclear. Is this term really considered in model fitting? I suppose the models are expressed in terms of their expectations rather than observations.

Assumed distributions

For each response variable, each model assumes a different probability distribution. Although I suppose the reason for convenience, are there any particular reasons? In particular, as to the beta distribution assumed for beta diversity, is this because the beta diversity takes a value between 0 and 1 as Sorensen index?

L550 and L637

I suppose that the authors meant a prior distribution for fixed parameters and intercept is assumed to be a normal distribution $N(0, \sigma^2)$ rather than a random variable N (which is not defined anywhere) follows a normal distribution—I suppose no deed for “ \sim ”.

Decision Letter, first revision:

10th March 2023

Dear Dr Montràs-Janer,

Thank you for your letter asking us to reconsider our decision on your Article entitled "Anthropogenic climate and land-use change drive short- and long-term biodiversity shifts across taxa". After careful consideration we have decided that we would be willing to consider a revised version of your manuscript.

Along with your revised manuscript, you should also submit a separate point-by-point response to all of the concerns raised by the reviewers, in each case describing what changes have been made to the manuscript or, alternatively, if no action has been taken, providing a compelling argument for why that is the case. If we feel that a substantial attempt has been made to address the reviewers' comments, this response will be sent back to the reviewers - along with the revised manuscript - so that they can judge whether their concerns have been addressed satisfactorily or otherwise.

I should stress, however, that we would be reluctant to trouble our reviewers again unless we thought that their comments had been addressed in full.

- ensure it complies with our format requirements for Articles as set out in our guide to authors at www.nature.com/natecolevol/authors/index.html

- state in a cover note the length of the text, methods and legends; the number of references and the number of display items.

Please ensure that all correspondence is marked with your Nature Ecology & Evolution reference number in the subject line.

Please use the following link to submit your revised manuscript:

[REDACTED]

I would appreciate it if you could tell me if you think you will be able to submit a revised manuscript, and also the likely timescale.

I look forward to hearing from you soon.

[REDACTED]

Author Rebuttal, first revision:

16REVIEWERS COMMENTS.

Many thanks for the thoughtful and constructive comments. Before our point-by-point response, we first summarize the main changes made to the manuscript, which we group into three main categories:

[1] Lack of clarity in the methods (all referees)

We facilitate better understanding of our analytical methods by bringing aspects of the methods to the main text (adding a paragraph at the end of the introduction), bringing clarity to the methods section incorporating answers to doubts, concerns, and misinterpretations, and further developing explanation of the analytical approach incorporating Supplementary Methods sections.

[2] Limitations of modelling and uncertainty of results (referees 2 & 3)

We include a model limitations paragraph in the discussion. Furthermore, we now assess model consistency estimating fixed effects using 10-fold cross-validation, as well as assess and display the uncertainty around the model estimates in text and Figures (i.e., standard deviation, 95% credible intervals, and providing violin plots describing the distribution of the data).

[3] Communication of novelty (referee 3).

We emphasize the novelty of our study, from the introduction, through the results, discussion, and Figures. Additionally, we have also improved all Figures, making them more informative, i.e., showing the model estimated fixed effects and uncertainty as well as the variation of the data.

Changes to the manuscript are shown in dark red, and relevant line numbers are given in the answers below.

Reviewer #1 (Remarks to the Author):

In this manuscript, the authors combine information on biodiversity records and historical land-use and climate change in Great Britain, to assess whether changes in British birds, plants and butterfly assemblages are affected by land use, climate change and their interaction. The authors investigate changes over two time periods, and also consider the effects of microclimatic heterogeneity and baseline biodiversity conditions on biodiversity indices. I found this study interesting, and I believe it contains a lot of results that are relevant for the field. I do, however, have major concerns regarding some aspects of the analyses, of the presentation of the work, and of interpretation of the results. I believe some improvements are necessary, and I have made some suggestions below which will hopefully be helpful to the authors.

Major

- A finding of your work, that you highlighted as surprising and for which you provided possible explanations in the discussion (lines 229-235)- was the decrease in butterfly CTI over the short term. However, reading the methods section, it seems to me that the confidence attributed to these results is not high, for two reasons that you highlight: because of possible recording biases (lines 570-580, with CI spanning 0 when replicating the analysis over a different time period, line 578) and also because of possible collinearity issues (lines 670-675). So, I can't help but question whether you should exclude the corresponding models and results from your work, as they come across as possibly not robust. (On the basis of the same arguments you could make the case that you're excluding these models).

Reply: We understand this suggestion. However, a surprising result, even when surrounded by high uncertainty, does not necessarily mean a wrong result or a result that should be just dismissed as it might encompass interesting insides. Considering that our results are resistant to further examination of the data (considering an alternative time period 2, discussed in **lines 494-504**), and that a similar pattern of negative responses of butterflies to climate change was observed in a recent study from Finland (Antão et al. 2022; <https://doi.org/10.1038/s41558-022-01381-x>), we argue that presenting the whole lot of our analysis and results in a completely honest, objective, and transparent way, offers a better contribution to the field than excluding them because they were unexpected. We probably would not discuss excluding the analysis of butterfly CTI if the results had shown an increase in CTI in the short-term, even if the model results were similarly uncertain.

- Lines 78 and line 84 – “baseline and interacting effects”: does this mean you considered the main effects of land use, climate, and their interactions, as well as their baseline values?

18Reply: Yes, we have considered main effects of land use, climate, and their interactions, as well as the baseline values of climate and land use (i.e. those values from the initial period of each analysis of change). We have addressed this lack of clarity by re-writing this whole paragraph. Now, we clearly state what our study is about, (lines 83-87, where old line 78 was found). Then, we state the three main goals of the study, specifying what we consider in each one of them (lines 87-99).

I think what the baseline conditions are needs to be defined in a clear way in the main text, given their importance in your analyses and results.

Reply: True. In this new version of the manuscript, we define that already in line 74, i.e., when we first mention *baseline environmental conditions*.

For instance, it isn't clear from the main text (line 84) that baseline biodiversity was considered.

Reply: True. This is now made clear in line 93 by writing: "We also consider the roles of baseline biodiversity conditions and ..."

- It is not clear to me whether investigating the effects of microclimatic heterogeneity on biodiversity change was one of the main aims of the study, or whether you just needed to control for microclimatic heterogeneity (lines 93/94), so I suggest clarifying this point.

Reply: We agree this whole paragraph was unclear and we have re-written it (lines 83-99). We consider microclimate heterogeneity more as a control variable, but we are of course interested in its effects and therefore report them in the results. We have now rephrased this sentence to more clearly indicate that it is more of a minor consideration, compared to land use and climate change (line 93-95): "We also consider the roles of baseline biodiversity conditions and microclimate heterogeneity (i.e., variation in microclimate temperature), which can influence biodiversity responses to larger-scale environmental changes).

- I think parts of the Methods need to be made clearer (see points below).

Reply: We are aware and appreciate all the listed concerns in that respect. In this new version of the manuscript, we have re-written the whole Methods section to make it more clear and understandable.

I found that some places a bit difficult to follow – in particular, the first point of the paragraph "Dealing with collinearity" line 643. Given that it is an important part of the methods describing how you treated and included the models' explanatory variables, I think it is important for this section to be very clear.

Reply: We agree that "Dealing with collinearity" is an important part of the methods. Therefore, in this new version of the manuscript we have completely re-written and extended. The more detailed description means that we were forced to move it to the Supplementary Methods, due to lack of space. This allowed us to keep with the flow of the manuscript as well as giving this section the amount of

space that we consider it requires for the reader to fully comprehend how we treated and included the explanatory variables in the model.

- Although I do appreciate the synthesis effort with figures 2, 3, 4, I would have liked to see some of the effect sizes. I wonder if you could change e.g. figure 2 to display the effects from the models (effect sizes for figure 3 and 5 might be harder present synthetically?)

Reply: We thank the reviewer for understanding the difficulty on synthesizing such figures and appreciate the suggestions for improvement. In that respect (and considering as well the comments of reviewer 3), we have worked hard to improve the figures.

Figure 2. We display the effects from the models in way that eases the interpretation of the results, i.e., we have transformed the regression coefficients into actual % of increase (or decrease) in species richness, beta diversity, and CTI between time periods. Additionally, in response to one of reviewer's 3 major concern (i.e., general lack in our study displaying the uncertainty around the estimates and the distribution of the data), we now also provide the uncertainty around these estimates, we have added violin plots describing the density distribution of the estimated biodiversity change across grid-cells (i.e., making clear the variation in biodiversity change across grid-cells), and we have calculated the probability of biodiversity increase (or decrease) across grid-cells (\pm standard error).

Figure 3. We display the effects from the models using colored bars representing the values of the parameter estimates (as in Supplementary Table 2). In response to reviewer 3 (i.e., to lift the novelty of the manuscript and to give the full set of results), we have also included the interactions and incorporated the associated uncertainty around the estimated effects.

Figure 5 (now, Figure 4). We have created a colored gradient to represent the effect sizes and we also give information of the standard deviation around the parameter estimates.

- I suggest highlighting some of the main limitations of your study in the discussion part.

Reply: Good suggestion. We have incorporated a discussion paragraph in that respect, specifically the fact that species and communities often respond to environmental change at finer scales than 10-km grid squares, and that even when correcting for sampling effort, there are still many unknowns (lines 308-325). We also point to the critical discussion of the findings of increased species richness that was present in the original manuscript.

Minor

- Line 54 and throughout the manuscript, I recommend hyphenating "land-use" only when used as a compound adjective (land-use change), not hyphenating otherwise.

20Reply: Changed accordingly.

- Line 27: “land-use change is generally inferred using proxies”- this is unclear to me. Do you mean impacts of land-use change on biodiversity, or land-use change itself? If so, what type of proxies?

Reply: Yes, it is unclear. We have re-written this sentence into (lines 32-34) “Yet, due to the lack of historical land-use data, the potential for land-use change and baseline land-use conditions to interact with a changing climate remains largely unknown”.

- Line 33: maybe specify what these baseline environmental conditions are

Reply: Due to limited space, we have not been able to clarify this in the abstract. However, we agree that this needs to be explained. Consequently, we have made this clear at the first mention of *baseline environmental conditions* in the Introduction (line 74).

- Line 37: “lower rates of biodiversity loss”: one of the main results presented in the abstract is increases in species richness, which seems to contradict this statement partly

Reply: We have now changed this to “lower rates of biodiversity change” (line 44).

- I wonder if you could conclude the abstract with a few words highlighting some important implications of your work.

Reply: Good idea. We have added “Our findings highlight the need to protect and restore natural and semi-natural habitats, alongside a fuller consideration of individual species’ requirements beyond simple measures of species richness in biodiversity management and policy.”

- Lines 65-69 and 224: Also see the work by Williams et al. (2019) on CTI in different land-use types:

<https://onlinelibrary.wiley.com/doi/full/10.1111/ecog.04806>

Reply: We have now added this reference.

- Line 68: I suggest highlighting that this is due to land use affecting local, microclimatic conditions to clarify the distinction between climate warming and microclimate

Reply: Changed accordingly.

- Line 74: what kind of proxies? Maybe add a few examples that can highlight how different your study is from these other studies based on proxies.

Reply: We have re-written this paragraph for clarification. Now lines 75-79 say “Yet, due to a lack of historical land-use data covering the timescales over which anthropogenic climate change and its effects have manifested, investigations of climate-land-use interactions are often limited to space-for-time

substitution (comparison of sites across a spatial gradient (e.g.,29), or the use of modern anthropogenic land cover as a measure of landscape change (e.g.,21,22).”

- Line 77: somewhere in the main text, I suggest displaying the sample sizes (number of considered species) – (or maybe in one of the figures?)

Reply: This is a good idea. We have incorporated that in Figure 2.

- Line 85: what is biodiversity resilience here? Do you focus on a specific metric?

Reply: We realized this was a poor choice of wording. Moreover, we also realized that this whole paragraph was not clear in the old version of the manuscript, and we have re-written it. Specifically here, this line has been changed FROM “3) habitat characteristics that determine grid-cell contribution to temporal biodiversity resilience at the national scale, which is key for conservation prioritization and planning” TO “Third, we uncover how climate and land use are associated to local contribution (i.e., grid-cell contribution) to national-level beta diversity over time, which is key for conservation prioritization and planning30” (lines 97-99).

- Lines 87 and 88: mixed-“effects” models. You could merge for brevity (e.g. “our analyses of biodiversity change were based on spatially explicit (generalized) linear mixed-effects models” or something along these lines).

Reply: N/A due to re-writing of this section.

- Line 93/94: do you control for these in all the models, or just the models built to investigate point (2)? Regarding baseline biodiversity conditions, I think you’ve already said you investigate their effects line 83/84?

Reply: Microclimate heterogeneity is included in the models to investigate goals 2 and 3. Baseline biodiversity is included in the models to investigate goal 2. But we realized that from the reviewers’ comments that we needed to incorporate a summary of the Methods in the main text and explain our model approach appropriately. In this new version of the manuscript, we have done exactly that. In lines 101-137, we have written a comprehensive paragraph where we briefly (but also clearly) explained our model approach and analysis of the data answering this (amongst other) doubts.

- Line 100: unclear to me what is meant by “loss of spatial heterogeneity” here

Reply: I agree, it is unclear, confusing and it does not bring any new information. We have deleted it.

- Line 102: this is not the first occurrence of CTI (first occurs line 83); I think you also need to explain a bit more in the main text what the thermal niche of a species is here (at present, the definition of CTI is a bit vague in the main text – although it is defined clearly in the methods).

22Reply: True, now it says CTI (line 145). We also agree that CTI should be better explained in the introduction and therefore we have added an explanation for Community Temperature Index in old line 83 (now line 89) “Community Temperature Index, hereafter CTI, an indicator of the relative community composition of warm- and cold-adapted species⁶”

- Line 106-108: what do these results mean?

Reply: Yes, that was confusing. We have removed.

- Line 112: “were consistent with a response to anthropogenic environmental change” I’m not sure what is meant by that

Reply: We have clarified this sentence (now lines 150-152) “Despite variation across taxa, biodiversity metrics and temporal scales, some broad patterns emerged across biodiversity changes in birds, butterflies, and plants in response to the anthropogenic climate and land-use changes that have taken place during the last 60 years.”

- Line 122: “strong determinants” & line 189 “driving mechanisms” – I suggest reporting results as correlative, because your study doesn’t investigate mechanism

Reply: True. We have changed accordingly throughout the manuscript and now the results are reported in terms of “are associated to”

- Line 124: “and the changes that occurred” here, do you mean changes in environmental conditions?

Reply: Yes. We have clarified this sentence, now it says “and the changes in environmental conditions that occurred” (line 172).

- Line 136 “enhanced stability”: how did you determine that? What is your definition of stability (and also resilience, elsewhere)

Reply: In this new version of the manuscript, we do not use the expression “enhanced stability” or the word “resilience”. We realized that it was a poor choice of words. We do use “stability” though (line 184) and we agree that we need to explain what we mean. In that respect, we have re-written lines 184-188 to clarify.

- Line 141: “improved and semi-natural grasslands” here, as baseline conditions, right?

Reply: Yes, but this paragraph has been re-written and this is N/A.

- Line 197 – delete the d in “land-use changed”

Reply: Changed accordingly.

- Line 209-221 I've enjoyed reading this paragraph of the discussion. I wonder if there is something to be said about invasive species as well? Could observed increases in species richness partly be explained by biological invasions?

Reply: Thank you. We mention non-native species, and species that are associated with human activity (line 262) but would rather not get into the invasive species debate here, due to problems and debates regarding their classification.

- Line 461 – Do you mean that for the 1960 time period you approximate land use from the 1930s survey?

Reply: Yes, this is the data that is available. We have added some lines in that respect (lines 409-411): “Although the historical land use data predates some of the species’ data by several decades, most of the major changes in land use, such as the widespread ‘agricultural improvement’ of pastures, occurred from the end of the 1960s⁸⁰”.

Are there any drawbacks from using a 1930 survey for a period of time centered around the 1960s?

Reply: Implications are discussed in the Discussion section (lines 256-261).

- Line 499 – delete the d in used

Reply: Changed accordingly.

- Line 513 – I'm not familiar with the Leroux model or the iCAR model to control for spatial autocorrelation. Why were these different approaches used for the species richness model and for the beta-diversity and CTI models (in models 1, 2 and 3)? And why was the iCAR model then used for the 18 other models (line 606), and then also in the last model (line 696)?

Reply: The short answer is that in the species richness model (eq. 1, where the response follows a Poisson and Negative Binomial distribution) we want to control for overdispersion. The Leroux model allows us to do that, but not the iCAR model. The iCAR model is simply a random effect on grid, but this random effect is spatially-structured, i.e., the grid depends on a neighbourhood structure (the surrounding eight grid-cells). The Leroux model is a generalization of the iCAR model, where the conditional distribution of the spatially-structured random effects on grid, is specified as in the iCAR model above, but it also incorporates an exchangeable structure for the spatially unstructured residual (i.e., the model residuals or the noise). A Gaussian model (model eq. 4) already incorporates an error term on grid (this model noise), and when we fit a Beta or Gamma GLMM (eq. 2 and 5, and eq. 3 respectively), we do not need to control for overdispersion. This is why we incorporate the spatial dependency using an iCAR model structure for Gaussian LMM, Beta and Gamma GLMM, but we incorporate the spatial dependency using a Leroux model structure for Poisson and Negative Binomial GLMM. We have explained that now (lines 458-470) including a description of the iCAR model structure in Supplementary Methods.

- Line 522 add an “s” to Poisson

Reply: Changed accordingly.

- Line 536 and where relevant throughout the manuscript, there is a mix of Greek letters spelt out (e.g. phi) and written in the Greek alphabet (e.g. ϕ), I suggest homogenizing throughout

Reply: Changed accordingly. Nonetheless, we would rather keep the use of the word “phi” in the Frescalo paragraph for two reasons, (1) to match the nomenclature used in R-package Sparta, and (2) to emphasize the difference between the Frescalo “phi” and the hyperparameter ϕ that describes the size of the Negative Binomial observations (1/overdispersion) in model equation 1 (of course, ϕ is the Greek letter phi but is never mentioned as such and always shown as symbol).

- Line 542 & 544 add an s to refer and show

Reply: Changed accordingly.

- Line 543: I suggest merging this paragraph with the paragraph on spatial autocorrelation line 513.

Reply: N/A due to re-writing of this section.

Also, I don’t think the following paragraph describes the “assessment for the inclusion of spatial autocorrelation” as you state? To me that following paragraph describes some fine-tuning aspects of the approach.

Reply: Agree, this paragraph was not describing the “assessment for the inclusion of spatial autocorrelation” it was vague, and therefore it has been deleted.

What was, then, the choice to include spatial autocorrelation based on?

Reply: In short, we first ran the models without including the spatially-structured random effect on grid. Then, we extract the Pearson residuals, and we plot these residuals against distance. These plots are called sample variograms and describe the spatial continuity of the data (hence, informing about spatial patterns present in the model residuals). If the plots display a relationship between residuals and distance, this will indicate that a spatial structure is present in the residuals and therefore, the model will need to include the spatially-structured random effect on grid.

In this new version of the manuscript, we describe the assessment of spatial dependency (reference included) in a paragraph located in Supplementary Methods (due to limited space in the main text). We clearly refer to this paragraph in lines 470-471 (for model eq. 1-3); lines 540-542 (for model eq. 4) and line 575 (for model eq. 5).

- Line 588 – mixed-“effects”

Reply: Changed accordingly throughout the manuscript.

- Why is recorder effort including differently in the first sets of models (1,2,3), in the second set of 18

25models (597-602), and in the last model (line 693)?

Reply: Note that in all model equations 1, 2 and 5, the recorder effort is included as an explanatory variable, as done in Zuur, A. F. & Ieno, E. N. *Beginner's guide to spatial, temporal and spatio-temporal ecological data analysis with R-INLA. Vol. I: GAM and Zero-Inflated Models*, Highland Statistics Ltd., 2018 – pages 444-445. This is because our main interest lies on the regression coefficients of the (other) environmental explanatory variables, but to correctly estimate these regression coefficients, we need to remove the effect of the recorder effort from the equation (or in other words, remove the part of variation of the response captured by the recorder effort). To accomplish that, we add the recorder effort as an explanatory variable (following the case presented by Zuur & Ieno, 2018 – pages 444-445). The same applies to model equation 4. But here, the response variable is *change of biodiversity*. Therefore, to avoid problems with the intercept, we also need to add the *change in recorder effort* in each grid-cell. As for all models referring to CTI (i.e., model equation 3 and CTI models corresponding to equation 4), we did not use the estimated recorder effort per se because CTI depends on the identity of the species configuring the community, and a higher or lower recorder effort does not necessarily imply a higher or lower CTI. Now, we explained that clearly already in the Introduction (lines 105-137) and in Methods (lines 442-458, 517-523, 572).

- Line 628 “and interpreted the results with caution” I’m not sure what this means for the results

Reply: True, it doesn't really mean anything. After full re-write of the Methods section, this sentence is no longer there.

- Paragraph “dealing with multicollinearity”, in particular point 1: line 655: I’m not clear with what “semi-natural grassland remained unregressed” mean. In my understanding, you used the variation in arable, forest, and urban covers not explained by semi-natural grassland cover (by using the residuals). So why not also include semi-natural grassland in the models then? Residual covers of arable, forest and urban should not then be collinear with semi-natural grassland cover?

Reply: We have been aware that this whole section “Dealing with collinearity” was poorly explained. In this new version of the manuscript, we have addressed that by re-writing the whole section. Now it is extended and moved to the Supplementary Methods, due to limited space. We have clarified how we dealt with collinearity, providing a comprehensive description of the methodology and the approaches that we have used. In short, (1) you have understood correctly: “we used the variation in arable, forest, and urban covers not explained by semi-natural grassland cover (by using the residuals)”, (2) the residual covers of arable, forest and urban were not collinear with semi-natural grassland cover, and (3) we did include semi-natural grasslands in the models - but I see how all of that could be easily misinterpreted in the old version of the manuscript.

- Line 683- "a species-poor site", with few but functionally unique species?

Reply: Exactly. We have added this little piece of additional information that has been suggested (now line 587).

Reviewer #2 (Remarks to the Author):

Review report on "Anthropogenic climate and land-use change drive short- and long-term biodiversity shifts across taxa"

The submitted article analyses long- and short-term changes in biodiversity for Birds, butterflies and plants in relation to land-use and climate change. Being very much aware of the tremendous efforts required to gather quality data, extensive spatio-temporal coverage for biodiversity, as well as land use and climate change data, I am hesitant to say but wonder if the present study stands upon reasonable data analyses and modelling grounds, while reading the description in the manuscript. This leads to several concerns regarding the utility and appropriateness of the models presented, which would be better to be clarified. I might have just misunderstood the documentation because of missing some technical clarity and limited notation in the manuscript. However, I still believe that models are crucial, even though the manuscript adopts some standard models, to capture spatio-temporal trends of biodiversity. Also, it would be great if the authors could improve the manuscript's clarity from my misinterpretation. Below, I provide my comments and concerns that are purely based on what I read and how I interpreted the manuscript description, hoping that this helps the authors.

Two-steps modelling

To evaluate the association of the changes in biodiversity to land-use and climate change, the authors have taken two modelling steps: 1) fit a model to each biodiversity index; and 2) fit its changes to land-use and climate change. I am concerned about how the model uncertainty in the first step is inherited in the second step and wonder why this is not evaluated in the manuscript. This is simply because ignoring the model uncertainty induces over-confidence in the second step. Since the authors formulated the models in Bayesian setting, such a flexible modelling framework, it looks more reasonable to combine these two steps into a single model and evaluate the underlying uncertainty. In particular, in the first step, the models (1)–(3) contain time as an explanatory variable representing temporal trends, so I do not see the point of splitting the modelling steps. Is there any particular reason?

Reply: We are sorry to hear that such unfortunate misunderstanding has occurred. Please note that we did not run a two-step modelling but two independent modelling approaches, each one of which seeks to answer two different questions. In this new version of the manuscript, we have added a summary

27Methods paragraph in the introduction section (lines 101-137) and re-write the Methods section to clarify this misunderstanding to happen again in the future. We are afraid this misunderstanding may have jeopardized (in that case) the evaluation of the modelling approach, but we are confident that the changes we have made in this new version of the manuscript would amend these concerns. On the other hand, we are happy to see that our approach seems to have been appropriately understood by reviewers 1 and 3.

In any case, we want to answer the question. The reason we did not run one unique model to investigate goals number 1 and 2 at once is because goals number 1 and 2 investigate two different things, each one of them requiring its own independent and differentiated modelling approach. While **goal number 1** seeks to investigate whether there has been a “significant” increase or decrease in taxonomic richness, biotic homogenization, and community adaptation to warmer climates between time periods; **goal number 2** seeks to investigate how “observed” changes on biodiversity relate to concurrent changes in climate and land use, climate-land-use change interactions, and baseline conditions of climate and land use (and how much of this biodiversity change is explained by environmental changes). Note that the response variable in goals number 1 and 2 are different. In **goal number 1**, the response variable is the species richness, the beta diversity and the CTI calculated in each grid-cell (for each time period and taxon). In **goal number 2**, the response variable is the change in species richness, beta diversity and CTI calculated in each grid-cell, for each temporal scale and taxon (i.e., richness, beta diversity and CTI in 2010s minus richness, beta diversity and CTI in 1960s for the long-term; and richness, beta diversity and CTI in 2010s minus richness, beta diversity and CTI in 1990s for the short-term). Note as well that **goal number 1** is basically a t-test, where the need to account for uneven recorder effort across grid-cells and spatial dependency, made us build a model instead of running a simple t-test. The answer we get here is the proportion of increase (or decrease) in richness, beta diversity and CTI between 1960s and 2010s, and between the 1990s and the 2010s across Great Britain. If we were to find no “significant” biodiversity increase (or decrease) in goal number 1, then goal number 2 would be practically irrelevant, and the analysis could stop here. One single hierarchical model would be appropriate if we were to investigate a process consisting of two related mechanisms. The point is, we are not investigating two related mechanisms but seeking answers to two different questions. We have now emphasized that and made it clearer in this new version of the manuscript.

Different model fitting for different time durations

Again to evaluate the association of the changes in biodiversity to land-use and climate change, the authors fit different models between the long-term (2010s-1960s) and the short-term (2010s-1990s) periods. Although the model structure is the same, different parameter estimates mean that there are two models for the same overlapped period (the short term is a subset of the long term). This indicates

model inconsistency. I wonder how such a discrepancy between the models would be interpreted and if such a separate model-fitting exercise is necessary.

Reply: One of the goals (or the main goal) of our study is to evaluate the association between changes in biodiversity and changes of land-use and climate across taxa and temporal scales (lines 120-129). To unravel these associations we fit (for each taxon and biodiversity metric) one model to investigate associations at the long-term (2010s-1960s) and one model to investigate the associations at the short-term (2010s-1990s), because each time period has different values of both response variables and predictor variables. A similar approach was also taken by Antão et al. (2022; <https://doi.org/10.1038/s41558-022-01381-x>), who split their data into even smaller decadal time periods, as well as splitting their study region into three blocks. As such, we are confident that this modelling approach was appropriate to address the goals of the study, and hope that our improved manuscript makes both these goals, and the methods chosen to approach them, more clear to the reader.

Recorder effort

In models (1)–(2), the recorder effort is included as an explanatory variable E_i with a coefficient to be estimated. If this is meant to be a bias correction term that reduces biases due to different recording efforts, it would be more natural to set the effort as an offset term that does not possess any coefficient. It would be very much appreciated if the authors could elaborate a little bit more on why the coefficient is needed and its interpretation in the model.

Reply: Yes, in model equations 1 and 2, we control for uneven recorder effort by adding the estimated recorder effort as an explanatory variable in the model, as done in Zuur, A. F. & Ieno, E. N. *Beginner's guide to spatial, temporal and spatio-temporal ecological data analysis with R-INLA. Vol. I: GAM and Zero-Inflated Models*, Highland Statistics Ltd., 2018 – pages 444-445. The reason why we control for uneven recorder effort in that way is because our main interest lies on the regression coefficients of the categorical variable T (which will give us the proportion increase or decrease in richness and beta diversity, see last paragraph below), but to correctly estimate these regression coefficients, we need to remove the effect of the recorder effort from the equation (or in other words, “remove” the part of variation of the response captured by the recorder effort). To accomplish that, we incorporate the recorder effort as an explanatory variable in the model (following the case presented by Zuur & Ieno, 2018 – pages 444-445). This is one way of controlling uneven recorder effort.

If we were to include recorder effort as an offset term, what we are doing is to assume that *doubling the recorder effort results in doubling species richness and beta diversity* (as explained in Zuur & Ieno, 2018), and we are not confident that this assumption holds for this data. Nonetheless, we did not want to dismiss the reviewer's comment. On the contrary, we wanted to test it. Therefore, we ran the models

eq. 1 and 2 again, this time including the recorder effort as an offset. Our approach was to include an offset variable with the form, $\log(\text{recorder effort})^\beta$, where β is the linear relationship between the recorder effort and species richness (in eq. 1) and beta diversity (in eq. 2) (as described in Plue et al., 2019; <https://doi.org/10.1111/geb.13201>). In this way, we estimate the linear relationship between the response and the recorder effort first, and then we include the offset term taking into account the actual curve-relationship (i.e., relaxing the doubtful assumption above). Almost all the results of this alternative method were consistent with all our results presented in the study (where we included recorder effort as an explanatory variable). We found only one exception, butterfly richness in the long and short-term (i.e., only 2 out of 12): while including the recorder effort as an explanatory variable in the model resulted in a proportion increase of butterfly richness of 25% and 5% at the long and short-term (Figure 2 in main text), including the $\log(\text{recorder effort})$ as an offset resulted in a proportion decrease of 6% at both temporal scales.

Note that by including the effort as an explanatory variable or an offset, we are not really modelling the same thing. By including the effort as an explanatory variable what we are asking to the model is the *proportion of temporal change in richness for one unit of effort*; while by including the effort as an offset what we are asking to the model is the *proportion of temporal change in richness per unit of effort* which is not the same. In any case, these results could suggest that while butterfly richness has increased across grid-cells, in modern times is also required a higher recorder effort to observe the same number of species per grid-cell. Considering the expansion range of the more common species and the alarming decrease in butterfly abundance, this interpretation could make ecological sense. But why do we only get different results for butterfly richness? Are these results supported by an ecological meaning or are they the result of a mathematical operation? This led us to a very interesting and important discussion among co-authors: what does it mean to include an offset in our model and how our model behaves when we include an offset. We need to keep in mind that any recording scheme had a lower recording effort in the past (in our study, we illustrate that in Supplementary Figure 7). Hence, as time goes by, the recorder effort increases (especially for butterflies) and just that, just the fact that the recorder effort increases across time, results in that: using recorder effort as an offset will fix the intercept and will force a negative result (which is exactly the result we got for butterfly richness). Following these concerns, we decided that our initial approach is valid and have therefore not added the alternative offset approach to the manuscript. However, we do now shortly describe our approach already in Introduction (lines 116-118), and in Methods (lines 440-445 and lines 491-492), including also reference to Zuur & Ieno, 2018. Nonetheless, we are grateful for this comment that made us think hard about our data and modelling approach.

Different spatial random structures in Richness

30It would be very much appreciated if the authors could elaborate more about u_i and v_i , contrasting Y_i , and why the random spatial structure should be different between Richness and the other two, beta diversity and community temperature index. Would this be simply due to Leroux model?

Reply: The short answer is that in the species richness model (eq. 1, where the response follows a Poisson and Negative Binomial distribution) we want to control for overdispersion. The Leroux model allows us to do that, but not the iCAR model. The iCAR model is simply a random effect on grid, but this random effect is spatially-structured, i.e., the grid depends on a neighbourhood structure (the surrounding eight grid-cells). The Leroux model is a generalization of the iCAR model, where the conditional distribution of the spatially-structured random effects on grid, is specified as in the iCAR model above, but it also incorporates an exchangeable structure for the spatially unstructured residual (i.e., the model residuals or the noise). A Gaussian model (model eq. 4) already incorporates an error term on grid (this model noise), and when we fit a Beta or Gamma GLMM (eq. 2 and 5, and eq. 3 respectively), we do not need to control for overdispersion. This is why we incorporate the spatial dependency using an iCAR model structure for Gaussian LMM, Beta and Gamma GLMM, but we incorporate the spatial dependency using a Leroux model structure for Poisson and Negative Binomial GLMM. We have explained that now (lines 458-471) including a description of the iCAR model structure in Supplementary Methods.

Related to this point, the role of the error term ε_i is a bit unclear. Is this term really considered in model fitting?

Reply: This was a mistake on our part. See from explanation above that to write ε_i is incorrect. We removed it.

I suppose the models are expressed in terms of their expectations rather than observations.

Reply: Yes, the models are expressed in terms of their expectations.

Assumed distributions

For each response variable, each model assumes a different probability distribution. Although I suppose the reason for convenience, are there any particular reasons? In particular, as to the beta distribution assumed for beta diversity, is this because the beta diversity takes a value between 0 and 1 as Sorensen index?

Reply: Yes, the reason each model assumes a different distribution is the following. Species richness is a count variable and follows a Poisson distribution (or Negative Binomial in case of overdispersion) and therefore is modelled as such. Beta diversity and LCBD are modelled using a Beta distribution because – as the reviewer points out – they both range between 0 and 1. CTI is a continuous and positive (no negative values and no zeros) variable that follows a Gamma distribution. For the change in species

richness, change in beta diversity and change in CTI, these are continuous variables ranging from negative to positive values and thus a Gaussian distribution is the appropriate here.

L550 and L637

I suppose that the authors meant a prior distribution for fixed parameters and intercept is assumed to be a normal distribution $N(0, \sigma^2)$ rather than a random variable N (which is not defined anywhere) follows a normal distribution—I suppose no need for “~”.

Reply: True, thanks. Now changed accordingly (line 604).

Reviewer #3 (Remarks to the Author):

Anthropogenic climate and land-use change drive short- and long-term biodiversity shifts across taxa

Montràs-Janer et al.

Manuscript number: NATECOLEVOL-221117943-T

In this well-written manuscript, the authors use a large data set on bird, butterfly and plant occurrences in Great Britain to assess, 1) how richness, beta diversity and the community temperature index of these taxa have changed over two different time periods; 2) whether these changes are driven by concurrent changes in environmental variables (climate and land cover), starting conditions at the beginning of the investigated time periods (“baseline” conditions of climate, land cover and biodiversity), by microclimatic heterogeneity or interactions between some of those variables; 3) which of these climate and land cover variables affect the grid-specific contribution to national biodiversity. The results show, that both climate and land-cover change led to increased richness, biotic homogenization and communities with more warm-adapted species. With a large part of the biodiversity change driven by baseline environmental conditions, the authors assume that observed changes may be lag-effects, derived from past changes, such as the removal of semi-natural grasslands shown to contribute strongly to overall biodiversity and linked to lower biodiversity loss across time. Despite this being an exceptionally valuable data set, these findings are not novel, and – unfortunately – the one aspect which would have set this study apart from similar studies was mostly neglected in most parts of the paper: how climate and land use may interact in shaping biodiversity patterns.

Reply: Many thanks for your review. We are strongly convinced that our study is novel, being to our knowledge the first national-scale analysis of the community effects of both changes in climate and land use over an appropriately long time period where wholesale changes in both variables have occurred. As

32such, even if some of community trends in Figure 2 have been previously identified (although we think that many would argue there are some controversial results there given the current 'biodiversity change' debate), we do not think that detracts from our focus on untangling the effects of land-use change, climate change and baseline environmental conditions on these observed patterns. We agree that perhaps the most novel aspect of the study is the possibility to investigate interactions between climate and land use change, and their effects on biodiversity. However, we concede that this aspect was somewhat neglected in the original manuscript version. In this new version of the manuscript, we have highlighted the climate-land-use interactions already in the introductory paragraph (lines 33, 42), in the Introduction (lines 67, 75-81, 91-97), in the Results (lines 160-168, and giving it more prominence in Figure 3) and in the Discussion (starting in line 234). Moreover, now we provide a Figure (in the Supplementary due to the large size of it) showing the associations between the interacting effects of land-use and climate change and biodiversity change (Supplementary Figure 3).

Still, the authors took great care to preselect and standardize the available data to avoid bias. Considering all provided material, the authors give sufficient information that should allow reproducing the results. And even though I have not used the integrated nested Laplace approximation for Bayesian inference myself, the selection of models and the methods applied to dealing with spatial autocorrelation, recorder effort and collinearity seem to be appropriate. However, the figures in the main paper do not show how the data was distributed, how large effects were and which uncertainty they had. And not all of this information is provided in the supplement either (see individual comments to figures and tables below) and/or the descriptions in the legend are missing. Reading the main text, I had to repeatedly refer to the extended methods section or even the supplement to really understand the research questions, the study design, the statistical methods and even some of the graphs (which are mostly simplified summaries of complicated result tables, but not necessarily easy to understand or interpret without seeing the raw data).

Reply: We really appreciated the reviewer likes, supports, and agrees with our model approaches. We also understand that the methodology needs to be explained in a more comprehensive and clear way. In that respect, we have brought more of the Methods to the main text, in the form of a summary method paragraph at the end of the introduction section (lines 101-137). Additionally, we have carefully rewritten the whole "Statistical analysis" section (in the Methods) and extended some parts in Supplementary Methods, with the intention to incorporate answers and clarifications to the reviewers' comments. Figures and Tables have also been modified according to reviewers' comments. As stated in a response to reviewer 1, we now also provide the uncertainty around estimates of community change in Figure 2, where we have also added violin plots describing the density distribution of the estimated biodiversity change across grid-cells (i.e., making clear the variation in biodiversity change across grid-cells). Finally, we also include the probability of biodiversity increase (or decrease) across grid-cells

(\pm standard error). We also provide the uncertainty around the model estimated fixed effects in Figures 3 and 4. To further convince the readers of the robustness of our models estimating fixed effects, we applied 10-fold cross-validation to model equations 4 and 5 (described in Methods, lines 590-599). This exercise demonstrated that our model estimates were robust. The results are presented in Supplementary Figures 2 and 5 and are presented in lines 157-158 and lines 222-223.

Looking at Figure 1 in the supplement I also wonder how meaningful some of the effects are ecologically, because mean values seem to be rather similar for most of the taxa and biodiversity measures. After all, even small differences may turn out to be “significant” if the sample size is large enough (which it probably is here).

Reply: Yes, this is tricky to overcome because it relates to the data itself. Nonetheless, now we have extended Figure 2 (in main text, that Supplementary Figure 1 refers to) to display not only increase or decrease of species richness, beta diversity and CTI between time periods, but also, the estimated probability of increase (or decrease) \pm standard error, across grid-cells and provide violin plots capturing the density distribution of the estimated biodiversity change (in accordance with previous comment). In this way, we intend to give the reader a better full picture of the uncertainty and variation around the temporal biodiversity change across Great Britain.

All in all, I can see the effort put into this manuscript and the great potential of the data set used, but I do not think this is fully reflected or utilized in the manuscript yet. In particular, one of the main research goals (climate-land-use interactions) was neither sufficiently presented nor discussed.

Reply: We have been aware of that and took notice of all reviewers concerns to address it. In this new version of the manuscript, we have not only used our Bayesian approach in a deeper manner to display uncertainties and data distribution, but we have also put much effort in highlighting the climate-land-use interactions. We have done that already in the Introduction, then in the Results and finally, in the Discussion section. Moreover, now we provide a Figure (Supplementary Figure 3, due to its large size could not be incorporated as a main Figure) showing the associations between the interacting effects of land-use and climate change and biodiversity change.

General comments

Language:

The authors switch between the terms “land-use vs land use”, “climate-change vs climate change”, “time-periods vs time periods”, “land-cover vs land cover”. I personally prefer “land use”, “climate change” etc unless used in combination with another noun (“e.g. land-use types”). Whichever way it is done, be consistent!

Reply: Change accordingly throughout the manuscript and in line with reviewer’s 1 comment above.

I was struggling a bit with the terms/phrases “(at the) long-term” and “(at the) short-term” repeatedly used throughout the manuscript. Reading this sounds a bit strange sometimes, as if it wasn’t the right word/phrase to use.

Reply: We use long-term and short-term to shorten to describe our time periods, where long = 50+ years, 1930s/60s-2010, and short = 20 years, 1990-2010. We have checked through the manuscript and made sure that in all sections, both long and short-term are defined either in terms of time periods of data, or length of time.

Title

The title captures the main findings.

I. 1: „... drives“ not „drive“?

Reply: Changed accordingly.

Summary paragraph

The summary captures the main aspects of the manuscript, yet also puts a lot of emphasize on climate-land use interactions. The resulting expectations of the reader are not met within the rest of the paper (at least in terms of results and discussion, where potential interactions are hardly mentioned)

I. 30: here and throughout change „land-use“ to „land use“? unless used in combination with another noun, e.g. “land-use data”

Reply: Changed accordingly. As mentioned above, we also give the interactions of climate and land use more emphasis throughout the rest of the manuscript.

I. 33. Unclear, what “baseline environmental conditions” are (need to read methods for that)

Reply: We agree that this needs to be explained. However, due to limited space, we have not been able to clarify this in the abstract, but we have made this clear at the first mention of *baseline environmental conditions* in the Introduction (line 74).

I. 37: mentioning “heathlands” next to semi-natural grasslands, even though it was only one of the land-use types summarized in the category “semi-natural grasslands” unjustifiably puts a lot of effort on that specific land-cover type

Reply: We have changed that to “Semi-natural grasslands (meadows, pastures, lowland and upland heathlands and open wetlands)” throughout the manuscript.

Main body

I. 54: replace „further change“ with „future change“

Reply: Changed accordingly.

I. 57: “has intensified”

Reply: Changed accordingly.

I. 61: “reorganisation at the community level” is a repetition to I. 55-56 (“reorganization of ecological communities)

Reply: True. We have re-written old line 61 (now line 66) to “Over the same period, the global climate has warmed by approximately 1°C¹⁶, which has been associated with shifts in species’ ranges to higher latitudes^{6,17,18}”.

I. 62-62: rephrase the sentence “and how they have interacted is a key unknown in studies of environmental change impacts on biodiversity.”

Reply: N/A due to re-writing of this section. This part of the sentence has been deleted.

I. 70: change “climate change effects” to “climate-change effects”

Reply: Changed accordingly.

I. 79-80: I suggest to slightly rephrase “... over 50+ years (long-term, approx. 1960s-2010s) and 20 years (short-term, 1990s-2010s) time-periods (Figure 1)” ◊ „...over two different time periods (Figure 1): 50+ years (long-term, approx. 1960s-2010s) and 20 years (short-term, 1990s-2010s)”; change „time-periods” to “time periods” here and throughout

Reply: Changed accordingly.

I. 83-84: “2) associations of...” this half-sentence is unclear

Reply: True. This sentence was part of the “goals paragraph”. Now, this whole paragraph has been re-written, and the sentence is completed. Please, see lines 91-97

I. 93: the microclimatic data is from time 3, and therefore control for effects of the most recent microclimatic conditions. This should be made more clear

Reply: Clarification has been added in the new version of the manuscript (line 95-97).

I. 94: unclear what “baseline biodiversity conditions” means in this context (unless one refers to methods at this point)

Reply: True. This is now made clear in line 93. Moreover, now we define what baseline conditions are already in line 74, when we first use this concept.

I. 105: “examination” instead of “interrogations” (as the latter sounds like the questioning of a crime suspect)

36

Reply: Changed accordingly.

I. 106: “most taxa” would be more precise than “all” taxa”; also, the use of “time period” for both the “point in time” (1960s, 1990s, and 2010s) and the “long- and short-term difference between time periods” (1960-2010 and 1990 to 2010, see for instance

Reply: N/A due to re-writing of this section. This part of the sentence has been deleted.

I. 111) is somewhat confusing.

Reply: We have rephrased (lines 150-152) to “Despite variation across taxa, biodiversity metrics and temporal scales, some broad patterns emerged across biodiversity changes in birds, butterflies, and plants in response to the anthropogenic climate and land-use changes that have taken place during the last 60 years.”

I. 113: whether the 1930s are included or not, the time period studied here covers at least 60 years (not 50 years)

Reply: Changed accordingly.

I. 114: again, it is unclear what environmental baseline means in this context

Reply: Now, this should be clear as: (1) we have already defined what environmental baseline means in line 74; (2) we have clarified this specific objective in lines 91-97; and (3) we have provided a summary of the Methods at the end of the introduction, where this specific model (and context) is described in lines 120-129.

I. 118: “at the expense of” instead of „at expenses of”

Reply: Changed accordingly.

I. 130: “land-cover classes” instead of “land cover classes”

Reply: Changed accordingly.

I. 159-162: “For butterflies...” this sentence is quite complicated and takes some time wrapping your head around

Reply: N/A due to re-writing of this paragraph.

I. 167: “associated with”

Reply: N/A due to re-writing of this paragraph.

I. 196: “a clear continuation of community reorganisation” is quite complicated, rather say “where communities are continuously reorganised”

Reply: Changed accordingly.

I. 197: “change” instead of “changed”

Reply: Changed accordingly.

I. 231 – 242: this is the only section where the names of authors from other papers are mentioned in the text (e.g. Devictor et al.). For consistency, I would remove those.

Reply: Changed accordingly.

I. 239: “resulting in...” change to “resulting in increased richness and beta diversity at the long-term, but decreased beta diversity and no clear increase in richness at the short-term.”

Reply: Changed accordingly.

I. 271 – 272: “for example,…” the examples listed here surely determine shifts in community richness, but not all of these were studied by the authors. I personally would prefer to have those listed that were actually found to play a role. Of course, “topography” does relate to microclimatic variability, but without reading the methods this is not clear

Reply: Please, note that just before citing the examples in lines 336-337, we say, “meaning that other factors could also be important” (in line 335). Our intention here is no other but citing other factors that could be relevant for determining shifts in community richness, biotic homogenization, and CTI (regardless their effect have already been studied or not).

I.274: “to conservation that” remove comma

Reply: Changed accordingly.

Methods:

Most of the issues I had with understanding the main text were resolved when reading the methods, as they nicely and in detail describe what has been done. For instance, the Frescalo approach to estimate unknown recording effort to standardize data is a great way to account for recording bias. The authors spent quite some time to explain the models, and adjustments for spatial autocorrelation and collinearity seem appropriate considering the limitations of the data, which are also nicely described in the methods section.

Reply: We are glad to hear that you find the Methods section to nicely describe our analysis and modelling approach. We understand, however, that the methods need to be explained somewhere in

the introduction for the reader to understand the main text without problems. To improve the manuscript, now we have added a summary Methods paragraph at the end of the introduction (lines 101-137) which should address this issue.

I. 393: for plants, the years of the atlas (1930 – 1960) do not match Figure 1 (1930 – 1969)

Reply: True. My mistake. Fixed accordingly.

I. 394: for consistency write “1970-74”

Reply: Changed accordingly.

I. 396 – 397: here the authors clearly state that, for simplicity, the time periods of sampling are afterwards referred to as 1960s, 1990s etc., but without reading the methods this is not clear in the main text

Reply: True. Now this should be clear in the main text as it is already incorporated in the summary Methods paragraph, specifically in lines 102-107.

I. 405: remove comma after “therefore”

Reply: Changed accordingly.

I. 461: here it is stated the land cover that is from the 1930s, in the caption of Fig. 1 it says 1930 – 50 and here 1930s. Clarify and be consistent.

Reply: True. We have arranged Figure 1 and the legend for consistency.

How may the fact, that the land use data for the time 1960s is from the 1930s may have affected the results?

Reply: Implications are discussed in the Discussion section (lines 256-261). But we have also added (lines 409-411) in Methods to make clear that “Although the historical land use data predates some of the species’ data by several decades, most of the major changes in land use, such as the widespread ‘agricultural improvement’ of pastures, occurred from the end of the 1960s⁸⁰”.

I. 463: here and below it says “2015 Land Cover Map”, while in Fig. 1 it states 2014-15

Reply: True. We have arranged Figure 1 for consistency.

I. 475 – 478: while these lines nicely describe how and for which time period microclimatic heterogeneity was calculated, the main text does not provide sufficient information to convey that knowledge, which makes it hard to understand the meaning and potential importance of that variable

Reply: True. Now, we have added lines 95-97 (in the introduction section) to make that clear from the beginning.

I. 493: “recorded in both periods”

Reply: Changed accordingly.

I. 499: “we did not use”

Reply: Changed accordingly.

I. 567: remove comma after U (“...U did not overfit”)

Reply: Changed accordingly.

I. 572: remove comma “...short-term analysis was an...”

Reply: Changed accordingly.

I. 589 – 593: here, the structure of the model is quite clear and nicely explained. In addition, the authors explain what they consider to be “baseline conditions”, i.e. the climate, land use and biodiversity status in the earlier time period considered in the biodiversity change models. This should also be clarified in the main text

Reply: We are glad to hear you liked the way we explain and describe the model structure here. Now we have incorporated a definition of what baseline conditions are already in **line 74**, when we first use this concept.

I. 628: I can see where the autocorrelation issue with beta diversity is coming from and acknowledge that the authors aim to “interpret with caution”. However, solely based on the main text this caution is not transferred to the reader. The same applies to the issue of collinearity.

Reply: True. Now, we have removed this part of the sentence and instead clearly state this issue in the Methods.

I. 689: remove comma “...time period a spatial...”

Reply: Changed accordingly.

I. 712: remove underline “2021”

Reply: Changed accordingly.

References:

The literature covers a good selection of previous literature that support the findings.

I. 352: First author name and title of article are in italics

Reply: Changed accordingly.

I. 386: Journal name is not italics

Reply: Changed accordingly.

I. 742: Journal name is not italics

Reply: Changed accordingly.

I. 744: Full stop missing after article title

Reply: Changed accordingly.

Tables and Figures

Fig. 1: Without reading the methods, it is somehow unclear that the maps shown in Figure 1 a) and b) are based on data collected within the time ranges listed below in c). And it is not clear that the changes shown for temperature and precipitation are based on the difference between the averaged values for the time periods listed in c).

Reply: To address these issues and clarify, we have now re-written the legend in Figure 1.

It would also be nice to mention in the main text (and possible caption) that the terms “1960s”, “1990s” cover all the ranges listed in c).

Reply: We have clarified this in the legend, and try to specify it in Methods: **lines 346-352** for the biological data, and in paragraph starting in **line 398** for the land use, climate and microclimate data.

In addition: why are the three time points shown for land-use change (a) and not the change across the long-term and short-term time periods, as for climate?

Reply: The reason is to ease visualization and be able to put all this information in one single figure.

Note that if we were to show the change of each land use category, we would need one map for each category and temporal scale, while if we show land use at the three time points (as we do), it only takes three maps for the reader to easily visualize how the land use have change in Great Britain across the three time periods.

Fig 1 caption: as elsewhere “land-use” vs “land use”, “time-periods” vs “time period”; I suggest to use different letter (“c”) for the panel about precipitation

Reply: Changed accordingly.

I. 856: “increase” not “increased”

Reply: Changed accordingly.

I. 857: “at the expense of” instead of “at expenses of”

Reply: Changed accordingly.

I. 860: “for breeding birds”

Reply: Changed accordingly.

I. 863: “(1930 – 1950)” for consistency; also, that does not match the figure, which shows that land use data for time 1 was from 1930 only

Reply: True. Changed accordingly.

Fig. 2: This figure is quite straightforward and easy to understand, although I personally would have preferred to see the actual mean values and credible intervals using the same colour code as in Supplementary table 1, as it also provides a measure of differences in effect size.

Reply: In that respect, we have improved this figure by (1) displaying the effects from the models in a transforming the regression coefficients into actual % of increase (or decrease) in species richness, beta diversity, and CTI between time periods (i.e., to facilitate interpretation of the results); (2) we provide the uncertainty around these estimates (in standard deviation instead of 95% CI) in the figure legend to not overcrowd the figure itself and to reduce the number of words in the legend; and (3) we have also added violin plots describing the density distribution of the estimated biodiversity change across grid-cells (to make clear the variation in biodiversity change across grid-cells), and we have calculated the probability of biodiversity increase (or decrease) across grid-cells (\pm standard error) (in answer to introductory comments above).

I. 868 - 869: “(i.e., 95% credible interval within the positive or negative range)” I know the 95% credible interval indicates that the effect has a 95% probability of falling within the given range, but this may not be clear to everyone based on the wording.

Reply: True. However, due to limited wording space and because how the significance is assessed for in a Bayesian approach is not necessary here, we have rephrased to “Coloured numbers next to the arrows provide the estimated proportion of relative biodiversity increase or decrease (significant in all cases and standard deviation lower than 0.035, Supplementary Table 6)”, which should be clear to all readers.

I. 869: “bird species” instead of “birds species”

Reply: Changed accordingly.

Fig. 3: I do not find this graph to be very intuitive to understand

Reply: Following suggestions from reviewer 1 and previous major comments from reviewer 3, we have improved this figure as follows. Now, we display the effects from the models using colored bars representing the values of the parameter estimates (as in Supplementary Table 2); we lift the novelty of the manuscript by including the interactions; and we incorporated the associated uncertainty around the estimated effects.

I. 881: “the latter” instead of “the former”, as this refers to microclimate?

Reply: N/A due to improvement and re-writing of the figure legend

It is also not clear, that the microclimate was derived within the last period time 3

Reply: We have addressed this issue and made that clear.

Fig. 4: I do not find this figure informative enough to be included in the main text. I would move it to the supplement.

Reply: Done. This figure has been removed from the main text.

Also, I have some issues with connecting the R-square values for models with and without baseline biodiversity data with lines, because this indicates a change (such as a change in space or time), which clearly is not the case here. Using unconnected symbols would have been better.

Reply: True. We have fixed the Supplementary Figure 4 according to the reviewer suggestions, i.e., removing connecting lines and replacing them by symbols.

Fig. 5: Again, this is a simple illustration of the main findings without providing any detail that allows comparing the effect of different variables without having to consult the supplement.

Reply: We have improved this figure following suggestions of both reviewer 1 and 3. Figure 5 (now Figure 4) has been improved by: (1) creating a colored gradient to display the effect of the different variables without overcrowding the figure with numbers; and (2) incorporating the standard deviation around the parameter estimates in the legend.

The exception hereby is the effect of semi-natural grasslands, for which mean effect values are shown. Why not for the other variables then?

Reply: Parameter estimates corresponding to semi-natural grassland are given in table because this land-cover category was the only one that showed an increased contribution to national-level biodiversity across time for all three taxa. Now, we have clarified that in the legend (lines 987-989).

Also, microclimatic heterogeneity here is referred to as “micro buffering temperature” which has not been used elsewhere (adapt that)

Reply: Changed accordingly. Now, it says “microclimate heterogeneity”, here and everywhere else throughout the paper.

Supplementary material

Table 1: I like that table and the comparison with the model without spatial dependency. It is a nice way of showing the magnitude and direction of effects, although the values do seem very small, especially for CTI.

Reply: In this new version of the manuscript, this is Supplementary Table 6 (not 1) and we have simplified it to strictly displayed the model results from equations 1,2 and 3 (as we found no differences between the models with and without spatial dependency).

Table 2: Again, this is a neat overview of the main results, which translates into the colour coded Figure 3 presented in the main text. The table is rather complicated, and still some important information (here 95% credible intervals) are missing. The length of the coloured bars do not seem to be comparable across the whole table, which makes the bars less useful. Also, this table includes stats missing from Fig 3: the interactions.

Reply: We are aware of the complexity of the table as it pretends to capture the whole set of (many) results. The purpose of this table is (1) to give the actual value of the estimated fixed effects; and (2) to show the R-squared values (marginal and conditional) of each model. Please, note that in this new version of the manuscript, we have improved the main Figure 3 by including this missing information. Now, Figure 3 includes both the interactions and the standard deviation around the estimated fixed effects, providing this important information already in the main text. Please, note that the reason why the fixed effects cannot be compared across the whole table is because the explanatory variables had to be standardized within each model (i.e., within each taxon, temporal scale, and biodiversity metric).

Table 3 + 4 captions: "...based on model equation..."

Reply: Changed accordingly.

Table 5: abbreviation of the time periods in caption (T1, T2, T3) does not match abbreviation in figure (time1, time2, time3).

Reply: Now, caption for the time periods (both in supplementary table and main figure) says 1960s, 1990s and 2010s (this is now Supplementary Table 3).

For consistency, the first column listing explanatory variables should not be coloured, as in table 1 and 2.

Reply: Changed accordingly.

Also, a different term for microclimatic heterogeneity is used in this table (micro buffering). This should be adapted

Reply: Changed accordingly.

Table 6: do the numbers in this table really indicate the Spearman correlation coefficient, as indicated in the table caption?

Reply: Yes, it is Spearman correlation coefficient. Note this is now Supplementary Table 1 and only displays the baseline conditions that are discussed in the main text (i.e., land-cover categories others than semi-natural habitats have been removed).

In that case I am surprised that correlations with very low correlation coefficients (e.g. -0.049 for effect of forest on short-term loss of bird richness) have p-values below 0.05.

Reply: We can only show the results as they appear. In our experience, correlation calculations such as these are almost always significant, regardless of the coefficient, even with data sets smaller than this.

Also, I find it hard to visualize the relationships based on this figure

Reply: True. Now, we have added “Note that higher baseline levels of species richness and lower beta diversity are associated with more stable communities over time, i.e., lower percentages of both species gained and lost” in the legend to help interpretation.

Fig. 1: the plots show the distribution of the data, however, apart from the upper two panels for butterflies it is difficult to see the effect suggested by the table. What is the actual difference in e.g. bird species richness between time 1 and 3? Based on the graph richness seems to be identical. I wonder if, due to the relatively large sample size (e.g. 2,670 grid cells for birds) even very small differences in richness will turn out to be significant (95% credible interval not including zero), yet does that mean the effect is biologically significant? Also, the different aspects of the boxplot/violin plot are not described in the figure caption (e.g. center line, median; box limits, upper and lower quartiles; whiskers, 1.5x interquartile range; points, outliers).

Reply: This figure has been removed. After the improvement of main Figure 2, we considered that this figure did not provide any additional not substantial information.

Fig. 2: Some of these panels are quite useful. E.g. it is nice to see the different proportion of each land cover, and their change between time periods (which is negligible in some cases). The meaning of panel b) is not that clear, especially of the third arrow (53%).

Reply: Old panel (b) has now been removed (now Supplementary Figure 1).

Fig. 3: Label taxon-groups (birds, butterflies, plants) with a, b and c. Refer to labels in caption

Reply: Addressed accordingly (now Supplementary Figure 7).

Fig. 4: this graph makes Figure 4 in the main text redundant. Still, I would not use a line to link the different R-squared values, as the x axis does not represent a continuous variable and therefore an increasing response.

Reply: True. Main Figure 4 has been removed and we have fixed the Supplementary Figure 4 according to the reviewer suggestions, i.e., removing connecting lines and replacing them by symbols.

Fig. 6: I think these kinds of scatter plots with trend lines are much easier to understand than tables or even the colour graphs used in the main text. They also nicely show the distribution of data points. I can see that it is tricky to include all of those graphs in the main text, as there are too many results.

But the main focus of the paper was on interactions (according to the knowledge gap identified in the introduction) and visualizing those would have helped!

Reply: True. Now, we have added a new Supplementary Figure (i.e., **Supplementary Figure 3**) that displays only the significant interactions. However, because of the lack of strong and consistent patterns regarding the statistical interactions and their effects on community change, we did not feel that we could select any particular panels for 'promotion' to the main text.

Decision Letter, second revision:

30th June 2023

Dear Dr Montràs-Janer,

Thanks for your patience during the re-review of your manuscript entitled "Anthropogenic climate and land-use change drives short and long-term biodiversity shifts across taxa". I can confirm that it has now been seen by the same 3 reviewers, whose comments are attached.

As you can see, although Referees #1 and #3 are largely satisfied with the changes, Referee #2 continues to raise concerns over the modelling which will need to be addressed before we can offer publication in Nature Ecology & Evolution. We will therefore need to see your responses to the criticisms raised, along with a revised manuscript, before we can reach a final decision regarding publication.

We therefore invite you to revise your manuscript taking into account all reviewer and editor comments. Please highlight all changes in the manuscript text file [OPTIONAL: in Microsoft Word format].

* If you have not done so already please begin to revise your manuscript so that it conforms to our Article format instructions at <http://www.nature.com/natecolevol/info/final-submission>. Refer also to

46any guidelines provided in this letter.

[REDACTED]

Nature Ecology & Evolution is committed to improving transparency in authorship. As part of our efforts in this direction, we are now requesting that all authors identified as 'corresponding author' on published papers create and link their Open Researcher and Contributor Identifier (ORCID) with their account on the Manuscript Tracking System (MTS), prior to acceptance. ORCID helps the scientific community achieve unambiguous attribution of all scholarly contributions. You can create and link your ORCID from the home page of the MTS by clicking on 'Modify my Springer Nature account'. For more information please visit www.springernature.com/orcid.

[REDACTED]

Reviewers' comments:

Reviewer #1 (Remarks to the Author):

I have now read a revised submission of "Anthropogenic climate and land-use change drives short and long-term biodiversity shifts across taxa". I find the manuscript much clearer overall; in particular, I think that the re-writing of the introduction has clarified the main objectives and framework, and I think having added a more detailed description/overview of the methodology at the end of the

47introduction is helpful. I would also like to thank the authors for their detailed response to previous comments. That said, I think the manuscript needs to be improved further. I don't have any additional comment on the technical aspects of the work. However, the presentation and framing of the work could, in my opinion, be improved. I hope my comments can contribute to clarifying aspects of this work and further strengthen the manuscript.

- Results/Discussion: The results show important differences in relative changes in biodiversity metrics among taxa, with the strongest relative effects for butterflies (e.g., 25% increase in species richness and 70% decrease in beta-diversity over the long term for butterflies). Other relative changes are, in comparison, much lower. This is something that has not been highlighted in the results section, and there is currently no comment in the discussion about these differences in terms of relative changes. This is something that I think is important and should be highlighted/discussed. What are the implications? (Line 235 of the discussion mentions 'magnitude of change' but there is no further detail).

- Framing of the work: I wonder if you could articulate some explicit hypotheses/expectations when presenting the major aims, to avoid framing the work as entirely exploratory. I think this could help structuring the results section around some key points and guide the reader through a lot of results (I find the results section is currently quite descriptive).

- Language: I suggest using 'associated with' consistently throughout the manuscript rather than jumping between 'associated to' and 'associated with'. More generally, I suggest paying attention to prepositions throughout the manuscript (e.g., change in rather than change of, etc., as the authors currently use different, and I think sometimes inaccurate wording).

- Title: should 'drives' be 'drive'? (climate and land-use change?)

- Lines 33/34 I think should be rephrased to specify what land use-climate interactions affect (e.g., biodiversity; and same for lines 79-81).

- Line 38/39 – I thought you found an opposite response for butterfly CTI over the short-term. Maybe add 'overall'.

- Line 99 – for extra clarity I suggest highlighting why beta diversity is important for conservation prioritization and planning.

- Line 144-145 But not butterfly communities over the short-term, so maybe rephrase with 'except for' or something along those lines.

- Line 335 – unclear to me what you mean by spatially structure here.

- Line 351 'is' rather than 'will be'

- Line 978 – effect 'on' rather than 'to'

- Figure 2 – I like this new version of the figure, I think it nicely summarizes a lot of the information. I do have a few concerns/thoughts about it though:

- o I suggest swapping “change species richness” for ‘species-richness change’, and so forth
- o Line 903 – I think “change in” is more correct than “change of” here. And line 906 – increase or decrease “in”. Line 910 – replace by based “on”. Please pay attention to this throughout the manuscript and SI.

- o I like how you display the relative change in the metrics in this revision. Why not also add the confidence interval around this relative change as you did for the estimated mean probability of increase or decrease?

- o Line 911 – the white dots are quite subtle on this figure. Maybe replace by a cross in a contrasting color? Or another horizontal line in a different color?

- o Line 918 – to have ‘increases’

- o Line 919 observed “in”

- Figure 3 – I appreciate the authors’ efforts to include uncertainty around the estimations, notably adding in panel (b). My main comment on this figure is that the color gradients representing the strength of the associations in panel (a) are not explained enough in the legend: what does a longer bar mean? What is meant by “strength”? Do these gradients reflect the value of the fixed effects? This makes the figure difficult to interpret without referring to the SI.

- Figure 4 lacks color gradient scales. What associations are positive, what are negative? This can’t be read from the figure currently.

Reviewer #2 (Remarks to the Author):

Please find the attached pdf.

Reviewer #3 (Remarks to the Author):

Dear authors,

I was happy see the effort you put into revising the manuscript, which covers a very nice data set and aims to answer some important questions about the drivers of biodiversity change. All in all, you did a great job dealing with all the reviewer comments.

I personally still prefer the more detailed figures from the supplement over the simplified summaries in the main text, but I appreciate the additions to existing graphs that show effect sizes and credible intervals now.

49I do have some (mostly minor) additional suggestions for change:

l. 95 - 97: "Although the microclimate data used..." move to the methods?

l. 97 - 99 and elsewhere:

- the meaning of the phrase "local/grid-cell contribution to national-level beta diversity" is not very clear. Here, the authors assess how climate and land use contributes to creating communities that are different in terms of species composition compared to average sites, which makes them (and the factors related to this difference) particularly relevant for conservation. I am not surprised semi-natural habitats turned out to be important in this context, as e.g. calcareous grasslands harbour a set of very distinct species. With the current terminology used, this important () message is hidden. I suggest clarifying the meaning of the phrase "local/grid-cell contribution to national-level beta diversity" throughout the text and also adjusting the wording in the results section and discussion accordingly.

- "associated with" instead of "associated to" (throughout the manuscript)

l. 129 - 133: see comment above (wording to clarify meaning of "grid-cell contribution to national beta diversity"). Also, was the climate-land-use interaction not modelled here (as in previous models)?

l. 146: "decrease of" instead of "decrease on"

l. 164: "of arable land"

l. 166: "At the short-term, however, ..."

l. 192: "plant CTI, respectively"

l. 197 and elsewhere: "... on the marginal R-square value"

l. 314: "climate interaction with"?

l. 440: "reference label" should not be italics

l. 522: "... in the previous section"

Fig. 1: Nice figure and clearer now with changes, but font size of headings, titles and legends is much too small and almost impossible to read

Fig. 2: I like the additional information provided in this plot now. Unfortunately, the values in brackets are rather small and it is impossible to see the white point which indicates the median. Maybe it is possible to make that larger?

l. 916 - 918: "model estimates..." should not be italics

l. 918 - 919: "b, overall number of species observed in GB in each time period."

Supplementary information:

I suggest to order tables in a way that makes them correspond with the figures in the main text. E.g. Table 6 refers to Fig.2 in the main text, so it should have been moved up in the supplement

Table 2: would have been nice to have the same order in this table as in corresponding fig. 3a (here, interactions are listed at the top)

Table 4 and 5 are not referred to in the main text

*****END*****

Author Rebuttal, second revision:

Reviewer #1 (Remarks to the Author):

I have now read a revised submission of “Anthropogenic climate and land-use change drives short and long-term biodiversity shifts across taxa”. I find the manuscript much clearer overall; in particular, I think that the re-writing of the introduction has clarified the main objectives and framework, and I think having added a more detailed description/overview of the methodology at the end of the introduction is helpful. I would also like to thank the authors for their detailed response to previous comments. That said, I think the manuscript needs to be improved further. I don't have any additional comment on the technical aspects of the work. However, the presentation and framing of the work could, in my opinion, be improved. I hope my comments can contribute to clarifying aspects of this work and further strengthen the manuscript.

Reply: Many thanks for your careful review of our revised paper. We are glad that you were generally pleased with the changes, and thanks to your new comments we are confident that the manuscript is now in much better shape.

- Results/Discussion: The results show important differences in relative changes in biodiversity metrics among taxa, with the strongest relative effects for butterflies (e.g., 25% increase in species richness and 70% decrease in beta-diversity over the long term for butterflies). Other relative changes are, in comparison, much lower. This is something that has not been highlighted in the results section, and there is currently no comment in the discussion about these differences in terms of relative changes. This is something that I think is important and should be highlighted/discussed. What are the implications? (Line 235 of the discussion mentions ‘magnitude of change’ but there is no further detail).

Reply: We have now added a paragraph in the discussion in response to this point (Lines 279-290. We mention both that this result could be down to changes in the less speciose butterfly community having relatively large impacts on percentage change, and that butterflies are also known to respond more quickly to environmental change, which could also result in a larger magnitude of community change.

51This then leads into the existing paragraphs looking more deeply at the individual trends that diverge from the broad patterns (increasing bird beta diversity and reduction in butterfly CTI).

- Framing of the work: I wonder if you could articulate some explicit hypotheses/expectations when presenting the major aims, to avoid framing the work as entirely exploratory. I think this could help structuring the results section around some key points and guide the reader through a lot of results (I find the results section is currently quite descriptive).

Reply: We have not added anything to the introduction following this comment. In our opinion, our dataset and analysis was sufficiently novel to not fully know what to expect, both in terms of the general directions of change (because different studies show a wide range of biodiversity changes across taxa and spatio-temporal scales and space-for-time and time-series approaches) and the drivers of change (because a national-level historical dataset of land use of comparable age and detail do not exist to our knowledge). We think that our introduction (and discussion) and the references used do show that we were aware of the existing literature and its wide range of findings and knowledge gaps, and that this is an equally reasonable way of conceiving and developing a study as outlining specific expectations and hypotheses. In the results section, we have now added a number of subheadings (according to the author instructions), which we think improves the structure.

- Language: I suggest using “associated with” consistently throughout the manuscript rather than jumping between “associated to” and “associated with”. More generally, I suggest paying attention to prepositions throughout the manuscript (e.g., change in rather than change of, etc., as the authors currently use different, and I think sometimes inaccurate wording).

Reply: We have changed this specific issue accordingly (e.g. Lines 129, 177) , and have also carefully checked the whole manuscript for other errors, editing for language and readability.

- Title: should “drives” be “drive”? (climate and land-use change?)

Reply: Yes. Changed accordingly (Line 1).

- Lines 33/34 I think should be rephrased to specify what land use-climate interactions affect (e.g., biodiversity; and same for lines 79-81).

Reply: Changed in both cases (Line 34, 80).

- Line 38/39 – I thought you found an opposite response for butterfly CTI over the short-term. Maybe add 'overall'.

Reply: Changed to 'broadly led' due to the short-term butterfly CTI response and the long term bird beta diversity responses that did not follow the general pattern (Line 37).

- Line 99 – for extra clarity I suggest highlighting why beta diversity is important for conservation prioritization and planning.

Reply. Clarification now added: "...which can aid conservation prioritization and planning both by identifying particular locations that contribute to large-scale biodiversity, as well as finding environmental attributes shared by the most valuable sites" (Lines 96-99).

- Line 144-145 But not butterfly communities over the short-term, so maybe rephrase with "except for" or something along those lines.

Reply: Changed to specify that all taxa exhibited CTI increases in the long-term, rather than breaking up the flow to highlight the exception (which is addressed in the following sentence; Line 147).

- Line 335 – unclear to me what you mean by spatially structure here.

Reply: Here we referred to the low marginal R² values in our model compared to the larger conditional R² values, indicating that spatial effects explain a relatively large amount of variation in our response variables. Rather than get stuck in these kinds of details in our final paragraph, we have now simplified the statement to "We also showed that despite the significant effects of climate change and land conversion, these variables only explained a relatively low proportion of biodiversity and CTI change in our models", before listing a few other potential mechanisms that might affect levels of community change (Lines 358-362)..

- Line 351 'is' rather than 'will be'

Reply: changed accordingly (Line 375).

- Line 978 – effect “on” rather than ‘to’

Reply: changed accordingly (Line 986).

- Figure 2 – I like this new version of the figure, I think it nicely summarizes a lot of the information.

Reply: Thank you. We put a lot of effort into incorporating the additional (and highly relevant) information into the figure while still keeping it simple to interpret the main patterns.

I do have a few concerns/thoughts about it though:

o I suggest swapping “change species richness” for ‘species-richness change’, and so forth

Reply: we have now swapped “change species richness” to “species richness” (and similarly for beta diversity and community temperature index), to avoid overcrowding the figure.

o Line 903 – I think “change in” is more correct than “change of” here.

Reply: changed accordingly.

And line 906 – increase or decrease “in”.

Reply: changed accordingly (Line 930).

Line 910 – replace by based “on”. Please pay attention to this throughout the manuscript and SI.

Reply: changed accordingly throughout the manuscript and SI (Line 934).

o I like how you display the relative change in the metrics in this revision. Why not also add the confidence interval around this relative change as you did for the estimated mean probability of increase or decrease?

Reply: Thank you. We actually tried that at the beginning but the figure got too crowded. Instead, for illustration purposes, we opted to provide the maximum uncertainty of the estimated proportion of relative biodiversity increase or decrease (in the standard deviation) in the figure caption (line 908) and detailed values in Supplementary Table 4.

o Line 911 – the white dots are quite subtle on this figure. Maybe replace by a cross in a contrasting color? Or another horizontal line in a different color?

Reply: Good idea. We have now replaced the white dot with a red dot and a horizontal red solid line, to improve readability.

o Line 918 – to have ‘increases’

Reply: We think that the original ‘probability ... for a grid cell to have increased in richness’ is correct here (Line 942).

o Line 919 observed “in”

Reply: Added, thanks (Line 943).

- Figure 3 – I appreciate the authors’ efforts to include uncertainty around the estimations, notably adding in panel (b). My main comment on this figure is that the color gradients representing the strength of the associations in panel (a) are not explained enough in the legend: what does a longer bar mean? What is meant by “strength”? Do these gradients reflect the value of the fixed effects? This makes the figure difficult to interpret without referring to the SI.

Reply: We have now improved the figure caption to explain that the orange and purple coloured blocks represent the direction of the effect from the model, while the (gradient) shaded bars within the box denote the effect size (previously referred to as ‘strength’; Lines 963-966).

- Figure 4 lacks color gradient scales. What associations are positive, what are negative? This can't be read from the figure currently.

Reply: We have now improved the figure caption to improve readability (Line 990-993).

Reviewer #2 (Remarks to the Author):

Thank you very much for your clarification and additional explanation in the manuscript and your point-to-point responses. I believe that the authors have successfully improved the manuscript's clarity, which readers will greatly appreciate. However, there are still some parts for which more clarification and consideration would be very much appreciated. I address these points below in blue ink for clarity.

Two-steps modelling

I very much appreciated the authors' effort to clarify this point which greatly helped me to understand the authors' view. It is now clear to me that, using the notation in the manuscript, the change in biodiversity (species richness *etc.*) in Eq. 4 is defined as

$$C_i = SR_i(t + 1) - SR_i(t)$$

for species richness, for example, rather than $C_i = \mu^{\hat{SR}_i(t+1)} - \mu^{\hat{SR}_i(t)}$ (please read $(t + 1)$ and t as different two time-points as appropriate to the context). Therefore this is not two-step modelling—I agree. However, my point initially meant a little bit more, which might not have been clear and seems to have caused a miscommunication here, I am afraid. Let me explain, elaborating on this aspect a little more carefully in detail, for your consideration.

Taking species richness as an example here, the model (Eq. 1) describes a trend between two time-points as

$$\begin{aligned}\mu^{SR_i(t+1)} &= \exp(\beta_0 + \eta + \Omega \log(E_i) + u_i + v_i) \\ \mu^{SR_i(t)} &= \exp(\beta_0 + \Omega \log(E_i) + u_i + v_i)\end{aligned}$$

The change in species richness is then expressed as

$$\begin{aligned}\mu^{C_i} &= \mu^{\text{SR}_i(t+1)} - \mu^{\text{SR}_i(t)} \\ &= \exp(\beta_0 + \eta + \Omega \log(E_i) + u_i + v_i) - \exp(\beta_0 + \Omega \log(E_i) + u_i + v_i) \\ &= (e^\eta - 1)e^{\beta_0} E_i^\Omega e^{u_i + v_i},\end{aligned}$$

where η is the parameter specifying the magnitude of species richness change, which can be a function of explanatory variables as $\eta(x_i)$. (Note: if you used the relative difference instead, $\mu^{\text{SR}_i(t+1)}/\mu^{\text{SR}_i(t)}$, the expression becomes simpler.) The above multiplicative expression is essentially the model for the change (difference) in species richness that the model (Eq. 1) suggests. However, the model proposed for biodiversity change (Eq. 4) in the manuscript takes a different additive form as

$$\mu^{C_i} = \alpha_0 + \mathbf{x}_i^\top \boldsymbol{\beta} + \delta E_i + \gamma_i,$$

which does not coincide with the previous expression derived from the model (Eq. 1). In particular, these models (Eqs. 1 and 4) assume different recorder effort and error structures in their form.

A concern here is the fact that the studied models (Eqs. 1 and 4) contradict each other because of the different recorder effort and error structures the authors have introduced. The same logic applies to the other biodiversity indices with different link functions. I see the authors' argument that there are two goals asking different questions; therefore, there are two different response variables and models. Addressing the core question in two steps, checking whether there is any temporal change in biodiversity and then further investigating links with climate and land use, is a sensible approach. However, I am inclined not to let rephrasing twists the mathematical model structure. I am curious to know the authors' thoughts and would like to encourage the authors to reconcile the contradiction since this is such a tremendous effort and great work.

Reply: This is an interesting discussion, and we appreciate the referee's efforts to ensure that our approach is correct and consistent. However, in short, we do not fully agree with the reviewer's conclusion that, because the expression $(\exp^{\text{intercept}} (\exp^\eta - 1) E_i^\Omega \exp^{(u_i + v_i)})$ and our equation 4 are different, then our equations (1 and 4), (2 and 4) and (3 and 4) "contradict each other because of the different

recorder effort and error structures the authors have introduced". We continue to stand by our initial approach. Here, we explain why this is our position.

In our study, we have estimated biodiversity change as defined in equation 4 (μ^C):

$$C_i \sim \text{Gaussian}(\mu^C, \sigma^2)$$
$$\mu^C_i = \text{Intercept} + X_i * \beta + E_i * \delta + \gamma_i$$

We agree with the reviewer that this estimated biodiversity change (μ^C_i) should be equivalent to the difference of an estimated biodiversity at time (t+1) and an estimated biodiversity at time (t) (meaning that the estimated values in both sides of the equation should be similar):

$$\mu^C_i = \exp(\mu^A SR_{i(t+1)}) - \exp(\mu^A SR_{i(t)})$$

(expression suggested by the reviewer - taking species richness SR as example – and where SR is a response variable that follows a Poisson or Negative Binomial distribution. We call this equation A)

Alternatively, one could also model biodiversity change as we do in our equation 4 - change in observed biodiversity modelled as a function of environmental change and its interactions, as described above. Our understanding is that the referee suggests that we for consistency use a similar model structure both for detection of biodiversity change (eq 1-3) and for the association of biodiversity change to environmental variables (eq 4). But we do not see how that can be done. We might still be misunderstanding each other, but in our view equation A and equation 4 should be similar in terms of estimated biodiversity change, and as such are not contradictory. However, we do not understand why our analyses must follow the same structure if they are being used to investigate different aspects of biodiversity change (that is, detecting change - equations 1 to 3 on which equation A is based; and associating change to environmental variables - equation 4).

The goal with equations 1, 2 and 3 (detecting change) was to model each biodiversity metric across GB in two different time periods, incorporating a factor variable (time period) that acted as a response variable to give us the proportion of change overall and whether this change is "significant or not", i.e., in essence a t-test, but using the more advanced mixed modelling approach which allowed us to control for spatial autocorrelation and recorder effort. The response variables in these models depended on the distributions of the data and as such Poisson and Negative Binomial (SR), Beta (Beta diversity) and Gamma (CTI) error structures were used, all of these having a log/logit link function. This would differ from the response in equation 4 (which follows a Gaussian distribution).

If we were to use the equation A approach for the analysis for associating biodiversity change to environmental variables, we would have to have the same data table structure used for equations 1-3, in order that the response variable would be the biodiversity metric (rather than biodiversity change), so that the error structure would follow the distributions of the respective biodiversity metrics (rather than the Gaussian distributions of the change values). We do not think that this is possible. For equations 1-3, the data table contains double the number of rows as the number of grid squares (i.e. each grid square at t and $t+1$). We did initially consider a similar option for our subsequent analysis but discarded it, mainly because a) it would not have allowed us to model the interaction of climate and land-use change; and b) it would have forced us to incorporate land use and climate conditions at time t and $t+1$, which would have very high collinearity. We do not think that our sequential regression approach to reduce collinearity would be appropriate for these variables, and collinearity was not an issue when including change in conditions together with baseline values. Due to these issues, we instead opted for the model structure defined in equation 4 (which answers our research question, i.e., effects of climate and land-use change, interactions, baseline conditions and microclimate heterogeneity on biodiversity change between time periods), where X is the vector of explanatory variables and β the vector containing the coefficients of interest; and E , the vector containing estimated recorder effort and change of estimated recorder effort, and δ the vector of their coefficients.

Regarding recorder effort (noting that this was not included in CTI models equation 3, for reasons explained in the Methods), we are not sure that E_i in the final solved version of equation A is incorporated in an appropriate way. Note that we used the Frescalo algorithm to model recording effort for each grid square for all study periods separately. As such, E_i in time $t+1$ and E_i in time t are different. Therefore, in the final solved equation E_i should refer to the difference between $E_i(t+1) - E_i(t)$, which we are not sure that it does as presented. We also include $E_i(t)$ as a separate predictor in our analysis because the effect of change in recorder effort will depend on the initial effort, and in this respect the referee is correct that equations 1-3 differ from equation 4 (as well as including baseline values of biodiversity, for the reasons we explain in the Main text). To make sure that this was not an issue, we did run the analysis based on equation 4 again with and without baseline E_i . The model predictions did not change.

As a further note regarding spatial terms. Note that the spatial term $u_i + v_i$ (in the developed equation A by the reviewer ($\exp^{\text{intercept}} (\exp^{\eta_i - 1}) E_{i,0} \exp^{(u_i + v_i)})$) and γ_i (in our equation 4) only differ in that, by using the term $u_i + v_i$ we are able to have more control over the overdispersion in the Poisson and Negative binomial distributed responses (i.e., SR, as explained in detail in Methods). So, in that case, this should not be a source of disagreement between any of the equations.

In sum, we are not convinced that our approach (i.e. equations 1 & 4, 2 & 4, 3 & 4) is contradictory, and we do not see what we could feasibly change while still being able to answer our research questions, given the constraints of the modelling methodology and our data. It is possible that we could have approached our research questions in another way, but we feel that our approach is appropriate, well-motivated and performed well in terms of model validation, despite the variation of biodiversity change recorded and estimated among grid cells. Importantly, we are fully open with the reader regarding our modelling approach, and have carefully read the method sections and made changes to improve clarity (for example that sampling effort was different for each grid square and time period in equations 1-2). We hope that this further explanation makes our model approach and the decisions we have taken along the way much clearer.

Different model fitting for different time durations

Thank you very much for the explanation. I see the authors' view.

No additional changes needed

Recorder effort

I very much appreciated the thoughtful comments and additional explanation. Quantifying recorder efforts is a challenging task. I am quite sure readers will appreciate the authors' effort carefully thought through about the recorder effort in this context.

No additional changes needed

Different spatial random structures in Richness

This is great—thank you for the explanation. Related to this point, please double check L.534

Reply: True, we had missed that one, and have now deleted it (Line 560)

Assumed distributions

Thank you for the clarification.

No additional changes needed

L550 and L637

Thank you, great.

No additional changes needed

Reviewer #3 (Remarks to the Author):

Dear authors,

I was happy see the effort you put into revising the manuscript, which covers a very nice data set and aims to answer some important questions about the drivers of biodiversity change. All in all, you did a great job dealing with all the reviewer comments.

I personally still prefer the more detailed figures from the supplement over the simplified summaries in the main text, but I appreciate the additions to existing graphs that show effect sizes and credible intervals now.

Reply: thank you. It is a challenge to summarise such an amount of results in a single figure that fits in the main text.

I do have some (mostly minor) additional suggestions for change:

l. 95 - 97: "Although the microclimate data used..." move to the methods?

Reply: Changed accordingly (Lines 442-444).

l. 97 - 99 and elsewhere:

- the meaning of the phrase "local/grid-cell contribution to national-level beta diversity" is not very clear. Here, the authors assess how climate and land use contributes to creating communities that are different in terms of species composition compared to average sites, which makes them (and the factors related to this difference) particularly relevant for conservation. I am not surprised semi-natural habitats turned out to be important in this context, as e.g. calcareous grasslands harbour a set of very distinct species. With the current terminology used, this important () message is hidden. I suggest clarifying the meaning of the phrase "local/grid-cell contribution to national-level beta

61diversity" throughout the text and also adjusting the wording in the results section and discussion accordingly.

Reply: We have now made changes throughout the manuscript. In the introduction, we are now more specific with what LCBD is (Lines 313-136). In the results, we spell out more clearly what a high association of semi-natural grassland cover for LCBD means (Lines 221-226). In the discussion, we explain in more detail why these results mean that semi-natural grasslands are of high conservation importance (Lines 260-263).

- "associated with" instead of "associated to" (throughout the manuscript)

Reply: changed accordingly.

l. 129 - 133: see comment above (wording to clarify meaning of "grid-cell contribution to national beta diversity"). Also, was the climate-land-use interaction not modelled here (as in previous models)?

Reply: See above for improvements to the LCBD part more specifically. Regarding the modelling, no, the interaction climate-land-use is not included in this model. Here, we aim to investigate how climate and land use (per se, not the interactions) are associated with local contribution (i.e., grid-cell contribution) to national-level beta diversity diversity over time. We think that the interacting effects of climate and land use change on community change is best kept to our main models concerned with understanding the ecology of biodiversity change (i.e. results in Fig 3.). For the LCBD models which we think are more conservation-focussed, we think that it is advantageous to keep to relatively simple models that can be translated into relatively simple messages, e.g. 'protect more grassland', compared to (as a potential example) 'protect more grassland, though this is less important when temperature/precipitation is relatively low/high'.

l. 146: "decrease of" instead of "decrease on"

Reply: changed to 'decrease in' (Line 145).

l. 164: "of arable land"

Reply: changed accordingly (Line 171).

l. 166: "At the short-term, however, ..."

Reply: changed accordingly (Line 172).

l. 192: "plant CTI, respectively"

Reply: changed accordingly (Line 198).

l. 197 and elsewhere: "... on the marginal R-square value"

Reply: changed accordingly throughout the manuscript and SI (Line 202).

l. 314: "climate interaction with"?

Reply Changed accordingly (Line 338).

l. 440: "reference label" should not be italics

Reply: changed accordingly and to 'reference level' (Line 466).

l. 522: "... in the previous section"

Reply: changed accordingly (Line 548).

Fig. 1: Nice figure and clearer now with changes, but font size of headings, titles and legends is much too small and almost impossible to read

Reply: We have done our best to increase the text size in all Figures, which now all are towards the upper end of the specifications given in the author instructions.

Fig. 2: I like the additional information provided in this plot now. Unfortunately, the values in brackets are rather small and it is impossible to see the white point which indicates the median. Maybe it is possible to make that larger?

Reply: We have changed the colour of the point and added a horizontal line for clarity.

l. 916 - 918: "model estimates..." should not be italics

Reply: Italics removed, but 'models estimate' is correct for what we are saying here (Line 940).

l. 918 - 919: "b, overall number of species observed in GB in each time period."

Reply: changed accordingly (Line 943).

Supplementary information:

I suggest to order tables in a way that makes them correspond with the figures in the main text. E.g. Table 6 refers to Fig.2 in the main text, so it should have been moved up in the supplement

Reply: We have reordered the tables as follows. Tables 4 and 5 have been moved to Supplementary Notes '*Penalised Complexity priors*', as they are actually part of this section. The remaining tables (and figures) have been numbered by order of appearance in the main text and then in the figures of the main text.

Table 2: would have been nice to have the same order in this table as in corresponding fig. 3a (here, interactions are listed at the top)

Reply: changed accordingly.

Table 4 and 5 are not referred to in the main text

Reply: True. Now, Tables 4 and 5 are mentioned and included in Supplementary Notes '*Penalised Complexity priors*', as they are actually part of this section.Review report on "Anthropogenic climate and land-use change drive short- and long-term biodiversity shifts across taxa"

Thank you very much for your clarification and additional explanation in the manuscript and your point-to-point responses. I believe that the authors have successfully improved the manuscript's clarity, which readers will greatly appreciate. However, there are still some parts for which more clarification and consideration would be very much appreciated. I address these points below in blue ink for clarity.

Two-steps modelling

To evaluate the association of the changes in biodiversity to land-use and climate change, the authors have taken two modelling steps: 1) fit a model to each biodiversity index; and 2) fit its changes to land-use and climate change. I am concerned about how the model uncertainty in the first step is inherited in the second step and wonder why this is not evaluated in the manuscript. This is simply because ignoring the model uncertainty induces over-confidence in the second step. Since the authors formulated the models in Bayesian setting, such a flexible modelling framework, it looks more reasonable to combine these two steps into a single model and evaluate the underlying uncertainty. In particular, in the first step, the models (1)–(3) contain time as an explanatory variable representing temporal trends, so I do not see the point of splitting the modelling steps. Is there any particular reason?

Reply: We are sorry to hear that such unfortunate misunderstanding has occurred. Please note that we did not run a two-step modelling but two independent modelling approaches, each one of which seeks to answer two different questions. In this new version of the manuscript, we have added a summary Methods paragraph in the introduction section (lines 101-137) and re-write the Methods section to clarify this misunderstanding to happen again in the future. We are afraid this misunderstanding may have jeopardized (in that case) the evaluation of the modelling approach, but we are confident that the changes we have made in this new version of the manuscript would amend these concerns. On the other hand, we are happy to see that our approach seems to have been appropriately understood by reviewers 1 and 3.

In any case, we want to answer the question. The reason we did not run one unique model to investigate goals number 1 and 2 at once is because goals number 1 and 2 investigate two different things, each one of them requiring its own independent and differentiated modelling approach. While goal number 1 seeks to investigate whether there has been a "significant" increase or decrease in taxonomic richness, biotic homogenization, and community adaptation to warmer climates between time periods; goal number 2 seeks to investigate how "observed" changes on biodiversity relate to concurrent changes in climate and land use, climate-land-use change interactions, and baseline conditions of climate and land use (and how much of this biodiversity change is explained by environmental changes). Note that the response variable in goals number 1 and 2 are different. In goal number 1, the response variable is the species richness, the beta diversity and the CTI calculated in each grid-cell (for each time period and taxon). In goal number 2, the response variable is the change in species richness, beta diversity and CTI calculated in each grid-cell, for each temporal scale and taxon (i.e., richness, beta diversity and CTI in 2010s minus richness, beta diversity and CTI in 1960s for the long-term; and richness, beta diversity and CTI in 2010s minus richness, beta diversity and CTI in 1990s for the short-term). Note as well that goal number 1 is basically a t-test, where the need to account for uneven recorder effort across grid-cells and spatial dependency, made us build a model instead of running a simple t-test. The answer we get here is the proportion of increase (or decrease) in richness, beta

56

diversity and CTI between 1960s and 2010s, and between the 1990s and the 2010s across Great Britain. If we were to find no “significant” biodiversity increase (or decrease) in goal number 1, then goal number 2 would be practically irrelevant, and the analysis could stop here. One single hierarchical model would be appropriate if we were to investigate a process consisting of two related mechanisms. The point is, we are not investigating two related mechanisms but seeking answers to two different questions. We have now emphasized that and made it clearer in this new version of the manuscript.

I very much appreciated the authors’ effort to clarify this point which greatly helped me to understand the authors’ view. It is now clear to me that, using the notation in the manuscript, the change in biodiversity (species richness *etc.*) in Eq. 4 is defined as

$$C_i = SR_i(t + 1) - SR_i(t),$$

for species richness, for example, rather than $C_i = \hat{\mu}^{SR_i(t+1)} - \hat{\mu}^{SR_i(t)}$ (please read $(t + 1)$ and t as different two time-points as appropriate to the context). Therefore this is not two-step modelling—I agree. However, my point initially meant a little bit more, which might not have been clear and seems to have caused a miscommunication here, I am afraid. Let me explain, elaborating on this aspect a little more carefully in detail, for your consideration.

Taking species richness as an example here, the model (Eq. 1) describes a trend between two time-points as

$$\begin{aligned} \mu^{SR_i(t+1)} &= \exp(\beta_0 + \eta + \Omega \log(E_i) + u_i + v_i) \\ \mu^{SR_i(t)} &= \exp(\beta_0 + \Omega \log(E_i) + u_i + v_i) \end{aligned}$$

The change in species richness is then expressed as

$$\begin{aligned} \mu^{C_i} &= \mu^{SR_i(t+1)} - \mu^{SR_i(t)} \\ &= \exp(\beta_0 + \eta + \Omega \log(E_i) + u_i + v_i) - \exp(\beta_0 + \Omega \log(E_i) + u_i + v_i) \\ &= (e^\eta - 1)e^{\beta_0} E_i^\Omega e^{u_i + v_i}, \end{aligned}$$

where η is the parameter specifying the magnitude of species richness change, which can be a function of explanatory variables as $\eta(x_i)$. (Note: if you used the relative difference instead, $\mu^{SR_i(t+1)} / \mu^{SR_i(t)}$, the expression becomes simpler.) The above multiplicative expression is essentially the model for the change (difference) in species richness that the model (Eq. 1) suggests. However, the model proposed for biodiversity change (Eq. 4) in the manuscript takes a different additive form as

$$\mu^{C_i} = \alpha_0 + x_i^\top \beta + \delta E_i + \gamma_i,$$

which does not coincide with the previous expression derived from the model (Eq. 1). In particular, these models (Eqs. 1 and 4) assume different recorder effort and error structures in their form.

A concern here is the fact that the studied models (Eqs. 1 and 4) contradict each other because of the different recorder effort and error structures the authors have introduced. The same logic applies to the other biodiversity indices with different link functions. I see the authors’ argument that there are two goals asking different questions; therefore, there are two different response variables and models. Addressing the core question in two steps, checking whether there is any temporal change in biodiversity and then further investigating links with climate and land use, is a sensible approach. However, I am inclined not to let rephrasing twists the mathematical model structure. I am curious to know the authors’ thoughts and would like to encourage the authors to reconcile the contradiction since this is such a tremendous effort and great work.

57

Different model fitting for different time durations

Again to evaluate the association of the changes in biodiversity to land-use and climate change, the authors fit different models between the long-term (2010s-1960s) and the short-term (2010s-1990s) periods. Although the model structure is the same, different parameter estimates mean

that there are two models for the same overlapped period (the short term is a subset of the long term). This indicates model inconsistency. I wonder how such a discrepancy between the models would be interpreted and if such a separate model-fitting exercise is necessary.

Reply: One of the goals (or the main goal) of our study is to evaluate the association between changes in biodiversity and changes of land-use and climate across taxa and temporal scales (lines 120-129). To unravel these associations we fit (for each taxon and biodiversity metric) one model to investigate associations at the long-term (2010s-1960s) and one model to investigate the associations at the short term (2010s-1990s), because each time period has different values of both response variables and predictor variables. A similar approach was also taken by Antão et al. (2022; <https://doi.org/10.1038/s41558-022-01381-x>), who split their data into even smaller decadal time periods, as well as splitting their study region into three blocks. As such, we are confident that this modelling approach was appropriate to address the goals of the study, and hope that our improved manuscript makes both these goals, and the methods chosen to approach them, more clear to the reader.

Thank you very much for the explanation. I see the authors' view.

Recorder effort

In models (1)–(2), the recorder effort is included as an explanatory variable E_i with a coefficient to be estimated. If this is meant to be a bias correction term that reduces biases due to different recording efforts, it would be more natural to set the effort as an offset term that does not possess any coefficient. It would be very much appreciated if the authors could elaborate a little bit more on why the coefficient is needed and its interpretation in the model.

Reply: Yes, in model equations 1 and 2, we control for uneven recorder effort by adding the estimated recorder effort as an explanatory variable in the model, as done in Zuur, A. F. & Ieno, E. N. *Beginner's guide to spatial, temporal and spatio-temporal ecological data analysis with R-INLA. Vol. I: GAM and Zero-Inflated Models*, Highland Statistics Ltd., 2018 – pages 444-445. The reason why we control for uneven recorder effort in that way is because our main interest lies on the regression coefficients of the categorical variable T (which will give us the proportion increase or decrease in richness and beta diversity, see last paragraph below), but to correctly estimate these regression coefficients, we need to remove the effect of the recorder effort from the equation (or in other words, "remove" the part of variation of the response captured by the recorder effort). To accomplish that, we incorporate the recorder effort as an explanatory variable in the model (following the case presented by Zuur & Ieno, 2018 – pages 444-445). This is one way of controlling uneven recorder effort.

If we were to include recorder effort as an offset term, what we are doing is to assume that doubling the recorder effort results in doubling species richness and beta diversity (as explained in Zuur & Ieno, 2018), and we are not confident that this assumption holds for this data. Nonetheless, we did not want to dismiss the reviewer's comment. On the contrary, we wanted to test it. Therefore, we ran the models eq. 1 and 2 again, this time including the recorder effort as an offset. Our approach was to include an offset variable with the form, $\log(\text{recorder effort}) \hat{\beta}$, where β is the linear relationship between the recorder effort and species richness (in eq. 1) and beta diversity (in eq. 2) (as described in Plue et al., 2019; <https://doi.org/10.1111/geb.13201>). In this way, we estimate the linear relationship between the response and the recorder effort first, and then we include the offset term taking into account the actual curverelationship (i.e., relaxing the doubtful assumption above). Almost all the results of this alternative method were consistent with all our results presented in the study (where we included recorder effort as an explanatory variable). We found only one exception, butterfly richness in the long and short-term (i.e., only 2 out of 12): while including the recorder effort as an explanatory variable in the model resulted in a proportion increase of butterfly richness of 25% and 5% at the long and short-term (Figure 2 in main text), including the log (recorder effort) as an offset resulted in a proportion decrease of 6% at both temporal scales. Note that by including the effort as an explanatory variable or an offset, we are not really modelling the same thing. By including the effort as an explanatory variable what we are asking to the model is the proportion of temporal change in richness for

58

one unit of effort; while by including the effort as an offset what we are asking to the model is the proportion of temporal change in richness per unit of effort which is not the same. In any case, these results could suggest that while butterfly richness has increased across grid-cells, in modern times is also required a higher recorder effort to observe the same number of species per grid-cell. Considering the expansion range of the more common species and the alarming decrease in butterfly abundance, this interpretation could make ecological sense. But why do we only get different results for butterfly richness? Are these results supported by an ecological meaning or are they the result of a mathematical operation? This led us to a very interesting and important discussion among co-authors: what does it mean to include an offset in our model and how our model behaves when we include an offset. We need to keep in mind that any recording scheme had a lower recording effort in the past (in our study, we illustrate that in Supplementary Figure 7). Hence, as time goes by, the recorder effort increases (especially for butterflies) and just that, just the fact that the recorder effort increases across time, results in that: using recorder effort as an offset will fix the intercept and will force a negative result (which is exactly the result we got for butterfly richness). Following these concerns, we decided that our initial approach is valid and have therefore not added the alternative offset approach to the manuscript. However, we do now shortly describe our approach already in Introduction (lines 116-118), and in Methods (lines 440-445 and lines 491-492), including also reference to Zuur & Ieno, 2018. Nonetheless, we are grateful for this comment that made us think hard about our data and modelling approach.

I very much appreciated the thoughtful comments and additional explanation. Quantifying recorder efforts is a challenging task. I am quite sure readers will appreciate the authors' effort carefully thought through about the recorder effort in this context.

Different spatial random structures in Richness

It would be very much appreciated if the authors could elaborate more about u_i and v_i , contrasting Y_i , and why the random spatial structure should be different between Richness and the other two, beta diversity and community temperature index. Would this be simply due to Leroux model?

Reply: The short answer is that in the species richness model (eq. 1, where the response follows a Poisson and Negative Binomial distribution) we want to control for overdispersion. The Leroux model allows us to do that, but not the iCAR model. The iCAR model is simply a random effect on grid, but this random effect is spatially-structured, i.e., the grid depends on a neighbourhood structure (the surrounding eight grid-cells). The Leroux model is a generalization of the iCAR model, where the conditional distribution of the spatially-structured random effects on grid, is specified as in the iCAR model above, but it also incorporates an exchangeable structure for the spatially unstructured residual (i.e., the model residuals or the noise). A Gaussian model (model eq. 4) already incorporates an error term on grid (this model noise), and when we fit a Beta or Gamma GLMM (eq. 2 and 5, and eq. 3 respectively), we do not need to control for overdispersion. This is why we incorporate the spatial dependency using an iCAR model structure for Gaussian LMM, Beta and Gamma GLMM, but we incorporate the spatial dependency using a Leroux model structure for Poisson and Negative Binomial GLMM. We have explained that now (lines 458-471) including a description of the iCAR model structure in Supplementary Methods.

This is great—thank you for the explanation.

Related to this point, the role of the error term ε_i is a bit unclear. Is this term really considered in model fitting?

Reply: This was a mistake on our part. See from explanation above that to write ε_i is incorrect. We removed it.

Please double check L.534.

I suppose the models are expressed in terms of their expectations rather than observations.

Reply: Yes, the models are expressed in terms of their expectations.

Thank you for the clarification.

59

Assumed distributions

For each response variable, each model assumes a different probability distribution. Although I suppose the reason for convenience, are there any particular reasons? In particular, as to the beta distribution assumed for beta diversity, is this because the beta diversity takes a value between 0 and 1 as Sorensen index?

Reply: Yes, the reason each model assumes a different distribution is the following. Species richness is a count variable and follows a Poisson distribution (or Negative Binomial in case of overdispersion) and therefore is modelled as such. Beta diversity and LCBD are modelled using a Beta distribution because – as the reviewer points out – they both range between 0 and 1. CTI is a continuous and positive (no negative values and no zeros) variable that follows a Gamma distribution. For the change in species richness, change in beta diversity and change in CTI, these are continuous variables ranging from negative to positive values and thus a Gaussian distribution is the appropriate here.

Thank you for the clarification.

L550 and L637

I suppose that the authors meant a prior distribution for fixed parameters and intercept is assumed to be a normal distribution $N(0, \sigma^2)$ rather than a random variable N (which is not defined anywhere) follows a normal distribution—I suppose no deed for “~”.

Reply: True, thanks. Now changed accordingly (line 604).

Thank you, great.

Decision Letter, third revision:

25th October 2023

Dear Dr. Auffret,

Thank you for submitting your revised manuscript "Anthropogenic climate and land-use change drive short and long-term biodiversity shifts across taxa" (NATECOLEVOL-221117943C). It has now been seen again by the original reviewers and their comments are below. The reviewers find that the paper has improved in revision, and therefore we'll be happy in principle to publish it in Nature Ecology & Evolution, pending minor revisions to satisfy the reviewers' final requests and to comply with our editorial and formatting guidelines.

[REDACTED]

Our ref: NATECOLEVOL-221117943C

3rd November 2023

Dear Dr. Auffret,

Thank you for your patience as we've prepared the guidelines for final submission of your Nature Ecology & Evolution manuscript, "Anthropogenic climate and land-use change drive short and long-term biodiversity shifts across taxa" (NATECOLEVOL-221117943C). Please carefully follow the step-by-step instructions provided in the attached file, and add a response in each row of the table to indicate the changes that you have made. Please also check and comment on any additional marked-up edits we have proposed within the text. Ensuring that each point is addressed will help to ensure that your revised manuscript can be swiftly handed over to our production team.

**We would like to start working on your revised paper, with all of the requested files and forms, as soon as possible (preferably within two weeks). Please get in contact with us immediately if you

71anticipate it taking more than two weeks to submit these revised files.**

In recognition of the time and expertise our reviewers provide to Nature Ecology & Evolution's editorial process, we would like to formally acknowledge their contribution to the external peer review of your manuscript entitled "Anthropogenic climate and land-use change drive short and long-term biodiversity shifts across taxa". For those reviewers who give their assent, we will be publishing their names alongside the published article.

Nature Ecology & Evolution offers a Transparent Peer Review option for new original research manuscripts submitted after December 1st, 2019. As part of this initiative, we encourage our authors to support increased transparency into the peer review process by agreeing to have the reviewer comments, author rebuttal letters, and editorial decision letters published as a Supplementary item. When you submit your final files please clearly state in your cover letter whether or not you would like to participate in this initiative. Please note that failure to state your preference will result in delays in accepting your manuscript for publication.

Cover suggestions

We welcome submissions of artwork for consideration for our cover. For more information, please see our https://www.nature.com/documents/Nature_covers_author_guide.pdf guide for cover artwork.

Nature Ecology & Evolution has now transitioned to a unified Rights Collection system which will allow our Author Services team to quickly and easily collect the rights and permissions required to publish your work. Approximately 10 days after your paper is formally accepted, you will receive an email in providing you with a link to complete the grant of rights. If your paper is eligible for Open Access, our Author Services team will also be in touch regarding any additional information that may be required to arrange payment for your article.

Please note that *Nature Ecology & Evolution* is a Transformative Journal (TJ). Authors may

72publish their research with us through the traditional subscription access route or make their paper immediately open access through payment of an article-processing charge (APC). Authors will not be required to make a final decision about access to their article until it has been accepted. [Find out more about Transformative Journals](https://www.springernature.com/gp/open-research/transformative-journals)

Authors may need to take specific actions to achieve [compliance with funder and institutional open access mandates](https://www.springernature.com/gp/open-research/funding/policy-compliance-faqs). If your research is supported by a funder that requires immediate open access (e.g. according to [Plan S principles](https://www.springernature.com/gp/open-research/plan-s-compliance)) then you should select the gold OA route, and we will direct you to the compliant route where possible. For authors selecting the subscription publication route, the journal's standard licensing terms will need to be accepted, including [editorial-policies/self-archiving-and-license-to-publish](https://www.nature.com/nature-portfolio/editorial-policies/self-archiving-and-license-to-publish). Those licensing terms will supersede any other terms that the author or any third party may assert apply to any version of the manuscript.

[REDACTED]

[REDACTED]

Reviewer #2:

None

Review report on "Anthropogenic climate and land-use change drive short- and long-term biodiversity shifts across taxa"

Thank you very much for your clarification and additional explanation in the manuscript and your point-to-point responses. I provide my comments below in black and keep my previous ones in blue for sake of clarity.

I very much appreciated the authors' effort to clarify this point which greatly helped me to understand the authors' view. It is now clear to me that, using the notation in the manuscript, the change in biodiversity (species richness *etc.*) in Eq. 4 is defined as

$$C_i = \text{SR}_i(t+1) - \text{SR}_i(t),$$

for species richness, for example, rather than $C_i = \hat{\mu}^{\text{SR}_i(t+1)} - \hat{\mu}^{\text{SR}_i(t)}$ (please read $(t+1)$ and t as different two time-points as appropriate to the context). Therefore this is not two-step modelling—I agree. However, my point initially meant a little bit more, which might not have been clear and seems to have caused a miscommunication here, I am afraid. Let me explain, elaborating on this aspect a little more carefully in detail, for your consideration.

Taking species richness as an example here, the model (Eq. 1) describes a trend between two time-points as

$$\begin{aligned} \mu^{\text{SR}_i(t+1)} &= \exp(\beta_0 + \eta + \Omega \log(E_i) + u_i + v_i) \\ \mu^{\text{SR}_i(t)} &= \exp(\beta_0 + \Omega \log(E_i) + u_i + v_i) \end{aligned}$$

The change in species richness is then expressed as

$$\begin{aligned} \mu^{C_i} &= \mu^{\text{SR}_i(t+1)} - \mu^{\text{SR}_i(t)} \\ &= \exp(\beta_0 + \eta + \Omega \log(E_i) + u_i + v_i) - \exp(\beta_0 + \Omega \log(E_i) + u_i + v_i) \\ &= (e^\eta - 1) e^{\beta_0} E_i^\Omega e^{u_i + v_i}, \end{aligned}$$

where η is the parameter specifying the magnitude of species richness change, which can be a function of explanatory variables as $\eta(x_i)$. (Note: if you used the relative difference instead, $\mu^{\text{SR}_i(t+1)} / \mu^{\text{SR}_i(t)}$, the expression becomes simpler.) The above multiplicative expression is essentially the model for the change (difference) in species richness that the model (Eq. 1) suggests. However, the model proposed for biodiversity change (Eq. 4) in the manuscript takes a different additive form as

$$\mu^{C_i} = \alpha_0 + \mathbf{x}_i^\top \boldsymbol{\beta} + \delta E_i + \gamma_i,$$

which does not coincide with the previous expression derived from the model (Eq. 1). In particular, these models (Eqs. 1 and 4) assume different recorder effort and error structures in their form.

A concern here is the fact that the studied models (Eqs. 1 and 4) contradict each other because of the different recorder effort and error structures the authors have introduced. The same logic applies to the other biodiversity indices with different link functions. I see the authors' argument that there are two goals asking different questions; therefore, there are two different response variables and models. Addressing the core question in two steps, checking whether there is any temporal change in biodiversity and then further investigating links with climate and land use, is a sensible approach. However, I am inclined not to let rephrasing twists the mathematical model structure. I am curious to know the authors' thoughts and would like to encourage the authors to reconcile the contradiction since this is such a tremendous effort and great work.

74

Reply: This is an interesting discussion, and we appreciate the referee’s efforts to ensure that our approach is correct and consistent. However, in short, we do not fully agree with the reviewer’s conclusion that, because the expression $(e^{\text{intercept}}(e^{\eta} - 1)E_i^{\Omega}e^{\mu_i+v_i})$ and our equation 4 are different, then our equations (1 and 4), (2 and 4) and (3 and 4) “contradict each other because of the different recorder effort and error structures the authors have introduced”. We continue to stand by our initial approach. Here, we explain why this is our position. In our study, we have estimated biodiversity change as defined in equation 4 (μ^{C_i}):

$$C_i \sim \text{Gaussian}(\mu^{C_i}, \sigma^2) \tag{1}$$

$$\mu^{C_i} = \text{Intercept} + X_i * \beta + E_i * \delta + \gamma_i \tag{2}$$

We agree with the reviewer that this estimated biodiversity change (μ^{C_i}) should be equivalent to the difference of an estimated biodiversity at time $(t + 1)$ and an estimated biodiversity at time (t) (meaning that the estimated values in both sides of the equation should be similar):

$$\mu^{C_i} = \exp(\mu^{SR_i(t+1)}) - \exp(\mu^{SR_i(t)})$$

(expression suggested by the reviewer - taking species richness SR as example – and where SR is a response variable that follows a Poisson or Negative Binomial distribution. We call this equation A)

Alternatively, one could also model biodiversity change as we do in our equation 4 - change in observed biodiversity modelled as a function of environmental change and its interactions, as described above. Our understanding is that the referee suggests that we for consistency use a similar model structure both for detection of biodiversity change (eq 1-3) and for the association of biodiversity change to environmental variables (eq 4). But we do not see how that can be done. We might still be misunderstanding each other, but in our view equation A and equation 4 should be similar in terms of estimated biodiversity change, and as such are not contradictory. However, we do not understand why our analyses must follow the same structure if they are being used to investigate different aspects of biodiversity change (that is, detecting change - equations 1 to 3 on which equation A is based; and associating change to environmental variables - equation 4).

The goal with equations 1, 2 and 3 (detecting change) was to model each biodiversity metric across GB in two different time periods, incorporating a factor variable (time period) that acted as a response variable to give us the proportion of change overall and whether this change is “significant or not”, i.e., in essence a t-test, but using the more advanced mixed modelling approach which allowed us to control for spatial autocorrelation and recorder effort. The response variables in these models depended on the distributions of the data and as such Poisson and Negative Binomial (SR), Beta (Beta diversity) and Gamma (CTI) error structures were used, all of these having a log/logit link function. This would differ from the response in equation 4 (which follows a Gaussian distribution).

If we were to use the equation A approach for the analysis for associating biodiversity change to environmental variables, we would have to have the same data table structure used for equations 1-3, in order that the response variable would be the biodiversity metric (rather than biodiversity change), so that the error structure would follow the distributions of the respective biodiversity metrics (rather than the Gaussian distributions of the change values). We do not think that this is possible. For equations 1-3, the data table contains double the number of rows as the number of grid squares (i.e. each grid square at t and $t+1$). We did initially consider a similar option for our subsequent analysis but discarded it, mainly because a) it would not have allowed us to model the interaction of climate and land-use change; and b) it would have forced us to incorporate land use and climate conditions at time t and $t+1$, which would have very high collinearity. We do not think that our sequential regression approach to reduce collinearity would be appropriate for these variables, and collinearity was not an issue when including change in conditions together with baseline values. Due to these issues, we instead opted for the model structure defined in equation 4 (which answers our research question, i.e., effects of climate and land-use change, interactions, baseline conditions and microclimate heterogeneity on biodiversity change between time periods), where X is the vector of explanatory variables and

β the vector containing the coefficients of interest; and E , the vector containing estimated recorder effort and change of estimated recorder effort, and δ the vector of their coefficients. Regarding recorder effort (noting that this was not included in CTI models equation 3, for reasons explained in the Methods), we are not sure that E_i in the final solved version of equation A is incorporated in an appropriate way. Note that we used the Frescalo algorithm to model recording effort for each grid square for all study periods separately. As such, E_i in time $t + 1$ and E_i in time t are different. Therefore, in the final solved equation E_i should refer to the difference between $E_i(t + 1) - E_i(t)$, which we are not sure that it does as presented. We also include $E_i(t)$ as a separate predictor in our analysis because the effect of change in recorder effort will depend on the initial effort, and in this respect the referee is correct that equations 1-3 differ from equation 4 (as well as including baseline values of biodiversity, for the reasons we explain in the Main text). To make sure that this was not an issue, we did run the analysis based on equation 4 again with and without baseline E_i . The model predictions did not change. As a further note regarding spatial terms. Note that the spatial term $u_i + v_i$ (in the developed equation A by the reviewer ($e^{\text{intercept}}(e^\eta - 1)E_i^\alpha e^{u_i+v_i}$) and γ_i (in our equation 4) only differ in that, by using the term $u_i + v_i$ we are able to have more control over the overdispersion in the Poisson and Negative binomial distributed responses (i.e., SR, as explained in detail in Methods). So, in that case, this should not be a source of disagreement between any of the equations.

In sum, we are not convinced that our approach (i.e. equations 1 & 4, 2 & 4, 3 & 4) is contradictory, and we do not see what we could feasibly change while still being able to answer our research questions, given the constraints of the modelling methodology and our data. It is possible that we could have approached our research questions in another way, but we feel that our approach is appropriate, well-motivated and performed well in terms of model validation, despite the variation of biodiversity change recorded and estimated among grid cells. Importantly, we are fully open with the reader regarding our modelling approach, and have carefully read the method sections and made changes to improve clarity (for example that sampling effort was different for each grid square and time period in equations 1-2). We hope that this further explanation makes our model approach and the decisions we have taken along the way much clearer.

I am glad to see that we have agreed upon the critical point, as the authors have commented: *"We agree with the reviewer that this estimated biodiversity change (μ^{C_i}) should be equivalent to the difference of an estimated biodiversity at time $(t + 1)$ and an estimated biodiversity at time (t) ..."* and further emphasised: *"... but in our view equation A and equation 4 should be similar in terms of estimated biodiversity change, ..."*. This is why I had to raise a question regarding the inconsistency in the model assumptions the authors have set, although I respect the researchers degrees of freedom. Since the authors have introduced different model structures for Eqs. 1 and 4, statistically speaking, it is not trivial for both estimates to become the same (or very similar). This requires at least some evidence to support and demonstrate that it is indeed the case here, as error structures, in general, affect the parameter estimates. I would like to encourage the authors to engage in reconciling this logical gap—I am quite sure that the authors' efforts will strengthen the current manuscript and will be ultimately paid off.

Author Rebuttal, third revision:Response to referees

Comment

I am glad to see that we have agreed upon the critical point, as the authors have commented: "We agree with the reviewer that this estimated biodiversity change (μCi) should be equivalent to the difference of an estimated biodiversity at time ($t + 1$) and an estimated biodiversity at time (t)..." and further emphasised: "... but in our view equation A and equation 4 should be similar in terms of estimated biodiversity change, ...". This is why I had to raise a question regarding the inconsistency in the model assumptions the authors have set, although I respect the researchers degrees of freedom. Since the authors have introduced different model structures for Eqs. 1 and 4, statistically speaking, it is not trivial for both estimates to become the same (or very similar). This requires at least some evidence to support and demonstrate that it is indeed the case here, as error structures, in general, affect the parameter estimates. I would like to encourage the authors to engage in reconciling this logical gap—I am quite sure that the authors' efforts will strengthen the current manuscript and will be ultimately paid off.

Response

We now in the methods alert the reader to this potential issue (Line 566-570), directing them to a new Supplementary Table 5 where we compare metrics of community change predicted from the different models with their different structures. We have also added a supplementary note 'Assessing model compatibility' (supplementary due to word limits in the main Method text), where we describe the approach and discuss the outcomes. We paste this text below. Many thanks for your help in improving our manuscript.

Assessing model compatibility

Our analytical approach involved first testing for average change in each community metric (species richness, beta diversity, CTI) over time (Models 1-3, value of each community metric at each grid-cell and time period on a separate row), and then attributing any observed changes to environmental variables in each grid cell (Model 4, change in community metric in each grid-cell on each row). We believe that this is a sensible and robust approach to both identify the direction and strength of community changes (while also controlling for spatial dependency and sampling effort), and to understand what might be driving these changes. However, this does mean the different models addressing the same community metric contain different error structures. For example, for bird species richness, Model 1 (testing for change) assumed a Poisson distribution in observed species richness, while Model 4 (attributing change) assumed a Gaussian distribution for observed change in species richness. In addition, the different models accounted for observer effort in different ways, with Models 1-2 including observer effort for each grid cell and time period (observer effort not included in Model 3 for CTI, see methods), while Model 4 included change in effort as well as the effort in the earlier time period.

To check for compatibility across these modelling approaches, we used our Models 1-3 to predict community metrics for each grid cell, time period and taxa, and then used a variant of Model 4 in which only observer effort and spatial variables were included to predict change in community metrics for each grid cell and taxa between time periods. We then compared predicted change from Models 1-3 (e.g. predicted value at time 3 minus predicted value at

78

ess
: is

time 1 for long-term change) with predicted change from Model 4. We found that the sign of predicted values (i.e. positive or negative change in species richness, beta diversity or CTI) in each grid cell was broadly consistent across model structures, while the correlation coefficients was also high (Supplementary Table 5). Furthermore, the average difference between model predictions of community change for each metric was low in relation to the range of values of predicted change. This exercise shows that Models 1-3 and Model 4 are compatible in terms of assessing both the direction and magnitude of community change over time using spatial and sampling effort predictors, and therefore the subsequent inclusion of environmental variables for the attribution of change was appropriate. Despite high sign agreement and low relative and absolute differences in predicted values, correlation in CTI across model types was somewhat lower than for species richness and beta diversity. This is understandable: while changes in community composition have commonly been shown to match the general direction of climate warming, rates of community changes are generally only weakly associated with the magnitude of climate warming that has occurred in a specific place^{10,11}. CTI models contained only spatial controls (see Methods), and as such we found that predicted change based only on spatial dependencies were less correlated. Finally, we also note that a main potential issue of the different model structures (Models 1-3 and 4) is related to how the parameter estimates for the predictor variables are interpreted. It was not in our interest to compare the estimates of the spatial and observer-effort variables present across model structures (to each other), while we also do not interpret relative values of parameter estimates of environmental variables from the outputs of Model 4 (e.g. effect of climate vs. effect of land use, or arable vs. urban).

Final Decision Letter:

4th January 2024

Dear Dr Auffret,

We are pleased to inform you that your Article entitled "Anthropogenic climate and land-use change drive short and long-term biodiversity shifts across taxa", has now been accepted for publication in Nature Ecology & Evolution.

Over the next few weeks, your paper will be copyedited to ensure that it conforms to Nature Ecology and Evolution style. Once your paper is typeset, you will receive an email with a link to choose the appropriate publishing options for your paper and our Author Services team will be in touch regarding any additional information that may be required

Due to the importance of these deadlines, we ask you please us know now whether you will be difficult to contact over the next month. If this is the case, we ask you provide us with the contact information (email, phone and fax) of someone who will be able to check the proofs on your behalf, and who will be available to address any last-minute problems . Once your paper has been scheduled for online publication, the Nature press office will be in touch to confirm the details.

Acceptance of your manuscript is conditional on all authors' agreement with our publication policies (see www.nature.com/authors/policies/index.html). In particular your manuscript must not be published elsewhere and there must be no announcement of the work to any media outlet until the publication date (the day on which it is uploaded onto our web site).

Please note that *Nature Ecology & Evolution* is a Transformative Journal (TJ). Authors may publish their research with us through the traditional subscription access route or make their paper immediately open access through payment of an article-processing charge (APC). Authors will not be required to make a final decision about access to their article until it has been accepted. [Find out more about Transformative Journals](https://www.springernature.com/gp/open-research/transformative-journals)

Authors may need to take specific actions to achieve [compliance with funder and institutional open access mandates](https://www.springernature.com/gp/open-research/funding/policy-compliance-faqs). If your research is supported by a funder that requires immediate open access (e.g. according to [Plan S principles](https://www.springernature.com/gp/open-research/plan-s-compliance)) then you should select the gold OA route, and we will direct you to the compliant route where possible. For authors selecting the subscription publication route, the journal's standard licensing

80terms will need to be accepted, including <https://www.nature.com/nature-portfolio/editorial-policies/self-archiving-and-license-to-publish>. Those licensing terms will supersede any other terms that the author or any third party may assert apply to any version of the manuscript.

We welcome the submission of potential cover material (including a short caption of around 40 words) related to your manuscript; suggestions should be sent to Nature Ecology & Evolution as electronic files (the image should be 300 dpi at 210 x 297 mm in either TIFF or JPEG format). Please note that such pictures should be selected more for their aesthetic appeal than for their scientific content, and that colour images work better than black and white or grayscale images. Please do not try to design a cover with the Nature Ecology & Evolution logo etc., and please do not submit composites of images related to your work. I am sure you will understand that we cannot make any promise as to whether any of your suggestions might be selected for the cover of the journal.

You can generate the link yourself when you receive your article DOI by entering it here: <http://authors.springernature.com/share>.

[REDACTED]

P.S. Click on the following link if you would like to recommend Nature Ecology & Evolution to your librarian <http://www.nature.com/subscriptions/recommend.html#forms>** Visit the Springer Nature Editorial and Publishing website at http://editorial-jobs.springernature.com?utm_source=ejp_NEcoE_email&utm_medium=ejp_NEcoE_email&utm_campaign=ejp_NEcoE for more information about our career opportunities. If you have any questions please click [here](mailto:editorial.publishing.jobs@springernature.com). **